# Electron-donable heterojunctions with synergetic Ru-Cu pair sites for biocatalytic microenvironment modulations in inflammatory mandible defects

Mingru Bai[1,6], Ting Wang[2,6], Zhenyu Xing [2], Haoju Huang[2], Xizheng Wu [2], Mohsen Adeli [3], Mao Wang[2], Xianglong Han [1,4], Ling Ye [1,5] ✉ & Chong Cheng [1,2] ✉

The clinical treatments of maxillofacial bone defects pose significant challenges due to complex microenvironments, including severe inflammation, high levels of reactive oxygen species (ROS), and potential bacterial infection. Herein, we propose the de novo design of an efficient, versatile, and precise electron-donable heterojunction with synergetic Ru-Cu pair sites (Ru-Cu/EDHJ) for superior biocatalytic regeneration of inflammatory mandible defects and pH-controlled antibacterial therapies. Our studies demonstrate that the unique structure of Ru-Cu/EDHJ enhances the electron density of Ru atoms and optimizes the binding strength of oxygen species, thus improving enzyme-like catalytic performance. Strikingly, this biocompatible Ru-Cu/EDHJ can efficiently switch between ROS scavenging in neutral media and ROS generation in acidic media, thus simultaneously exhibiting superior repair functions and bioadaptive antibacterial properties in treating mandible defects in male mice. We believe synthesizing such biocatalytic heterojunctions with exceptional enzyme-like capabilities will offer a promising pathway for engineering ROS biocatalytic materials to treat trauma, tumors, or infection-caused maxillofacial bone defects.

The maxillofacial bone defects after the trauma, deformity, tumor, or infection may cause severe dysfunctions and dramatically decrease the life quality of patients[1-3]. According to clinical data, there are about 2 million patients suffer from maxillofacial bone defects worldwide[4]. Due to their irregular shapes and large volumes, the tissue reconstructions of mandible defects remain a great challenge in clinical treatments[5]. Several strategies have been proposed recently, including bone grafting and heterogeneous material implants[2,6,7], but the clinical treatment effects are very limited due to the complex microenvironments, including severe inflammatory conditions, high ROS levels, and potential bacterial infections[8-10]. The regeneration of bone tissues in mandible defects is principally through the process of intramembranous osteogenesis, in which stem cells differentiate initially into osteoblasts and afterward grow into complex bone structures[11,12]. It is of critical importance to provide an appropriate microenvironment for the infiltration and differentiation of stem cells in inflammatory

[1]State Key Laboratory of Oral Diseases, National Center for Stomatology, National Clinical Research Center for Oral Diseases, West China Hospital of Stomatology, Sichuan University, Chengdu, China. [2]College of Polymer Science and Engineering, State Key Laboratory of Polymer Materials Engineering, Sichuan University, Chengdu, China. [3]Institute of Chemistry and Biochemistry, Free University of Berlin, Berlin, Germany. [4]Department of Orthodontics, West China Hospital of Stomatology, Sichuan University, Chengdu, China. [5]Department of Cariology and Endodontics, West China Hospital of Stomatology, Sichuan University, Chengdu, China. [6]These authors contributed equally: Mingru Bai, Ting Wang. ✉e-mail: yeling@scu.edu.cn; chong.cheng@scu.edu.cn

mandible defect[13,14]. However, therapeutic efficacy based on endogenous stem cells is always threatened by the accumulated reactive oxygen species (ROS) and poor oxygen supply[15]. Especially in inflammatory defects, high ROS-level microenvironments could aggravate localized tissue injury and lead to chronic inflammation[14,16,17]. Therefore, clearance of excessive ROS to provide an appropriate microenvironment for endogenous stem cells has become an intriguing approach for promoting bone tissue regeneration in the treatments of inflammatory mandible defects[18].

In the human body, natural antioxidases, like catalase (CAT), superoxide dismutase (SOD), and glutathione peroxidase (GPx), are responsible for maintaining the balance of oxidative stress in normal tissue[19–22]. Nevertheless, the antioxidase-based therapy also exhibits several intrinsic drawbacks, such as short circulating half-life, low cell permeability, and antigenicity, which restricts their direct applications in treating inflammatory injured sites[22,23]. Chemists and material scientists are now endeavoring to imitate natural antioxidases to create new biocatalytic materials for various antioxidant therapeutic purposes[23–27]. Recently, polyphenol nanoparticles[28–30], nanocarbons[31,32], metal oxides[22,33,34], etc., have been intensively developed into antioxidant nanostructures for ROS clearance and tissue regeneration therapies, especially creating metal oxides for biocatalytic therapeutics. Among diverse metal oxides, cerium oxide ($CeO_2$) has been reported to exhibit excellent biostability, biocompatibility, and a certain degree of antioxidase-like properties and can be fabricated into bone scaffolds with high mechanical strength[35–40]. However, the biocatalytic ROS scavenging activities of currently reported $CeO_2$ materials are very low, the elimination ROS types are also limited, and especially, they lack intelligent and bioadaptive catalytic properties to meet the complex needs of repair microenvironments[36,41].

Recently, our team has disclosed that the Ru coordination with porphyrin networks on nanocarbon surfaces exhibits superior antioxidase-like activities[15], which can mimic the natural enzyme system to overcome the multielectron reactions of oxygen species via accelerating electron transfer for efficient redox balance. However, this biocatalytic system relies on conductive frameworks and carbon substrates; utilizing metal oxides to design Ru-based artificial antioxidase will be challenging due to the lack of sufficient electron donors[18,42]. The catalytic atoms in metal oxides-based artificial antioxidases are usually coordinated with electronegative O elements, which may lead to low ROS scavenging efficiencies and require high dosages[43,44]. Research has shown that creating unique metal oxide nanocrystallines[45–50] or heterojunctions[48,51–53] can provide strong electronic metal-support interactions (EMSIs) in catalytic systems, which can tune the electronic structures of catalytic centers by providing necessary electrons and facilitating rapid electron transfer; while this has not been explored for the metal oxides-based artificial enzyme systems. As an essential trace element in humans, Cu ions play critical roles in many natural enzymes[54]; meanwhile, Cu species tend to lose electrons and show high electrical conductivity[55]. Thus, the copper oxide-based nanomaterials or heterojunctions have aroused extensive interest in both energy-conversion-related catalytic fields and biocatalytic artificial enzyme systems[22,56–58], which may offer a new opportunity to design Ru coordination centers on metal oxides with superior antioxidase-like activities.

Here, inspired by the natural antioxidases and EMSIs catalytic system, we propose the de novo design of an efficient, versatile, and precise electron-donable heterojunction with synergetic Ru-Cu pair sites (Ru-Cu/EDHJ) for superior biocatalytic regeneration of inflammatory mandible defects and pH-controlled antibacterial therapies. As depicted in Fig. 1, our research is driven by two primary purposes: (1) the unique structure of Ru-Cu/EDHJ is favorable for enhancing the electron density of Ru atoms and optimizing the binding strength of oxygen species on Ru sites, thus enhancing the enzyme-like catalytic performances (Fig. 1a), (2) the biocompatible Ru-Cu/EDHJ can

efficiently switch the ROS scavenging (neutral media) or ROS generation (acidic media) capabilities via pH-controlled pathway (Fig. 1b), thus simultaneously exhibiting superior repair function and bioadaptive antibacterial properties in treating mandible defects (Fig. 1c). Remarkably, our experimental and theoretical studies demonstrate that the Ru-Cu coordination structure enhances the electron density of Ru atoms, thus empowering the Ru-Cu/EDHJ with excellent and versatile antioxidase-like activities and pH-controlled ROS catalytic properties. For instance, for the CAT-like catalytic elimination of $H_2O_2$, the Ru-Cu/EDHJ exhibits an exceptional $V_{max}$ of $60.24\ \mu m\ s^{-1}$, $K_m$ of 79.37 mM, and turnover number of $8.09\ s^{-1}$, largely surpassing the state-of-the-art biocatalysts. Accordingly, the antioxidase-like Ru-Cu/EDHJ can maintain stem cell survival, proliferation, and osteogenic differentiation in high ROS-level conditions by preventing oxidative stress-induced DNA damage and cell apoptosis, maintaining the metabolism and osteogenesis functions, and reducing inflammatory responses. Moreover, the Ru-Cu/EDHJ also displays excellent pH-controlled antibacterial properties via peroxidase (POD)-like catalytic ROS production in acidic environments, which allows the Ru-Cu/EDHJ owns bioadaptive anti-infection activity in treating bacteria-induced refractory mandibular diseases. Strikingly, the synthesized Ru-Cu/EDHJ exhibits superior and unexpected biocatalytic performances, which offers a promising pathway to create ROS biocatalytic materials for treating trauma, tumor, or infection-caused maxillofacial bone defects.

## Results
### Synthesis and characterization of biocatalytic heterojunctions
The electron-donable heterojunctions (EDHJ) were synthesized by a two-step hydrothermal process (Supplementary Fig. 1). In brief, the $CeO_2$@$Cu_2Cl(OH)_3$ precursor (the $CeO_2$ matrix supported $Cu_2Cl(OH)_3$ nanoclusters) is synthesized first, the abundant -OH groups on the precursor can act as functional nanoglues to immobilize Ru atoms. By changing the reaction temperatures (120 °C and 180 °C), two Ru atoms loaded heterojunctions ($CeO_2$@$Cu_2O$ (high electron-donation) and $CeO_2$@CuO (low electron-donation)) can be obtained (Fig. 2a). On the $CeO_2$@$Cu_2O$, there are abundant Cu atoms on $Cu_2O$ surface to bind with Ru species; thus, the obtained artificial enzyme is named as Ru-Cu/EDHJ. On the $CeO_2$@CuO, most Ru species are bound with surface O atoms of CuO due to its unique crystalline structure, thus naming Ru-O/EDHJ. For comparison, the $CeO_2$ nanoparticles and atomic Ru supported by bare $CeO_2$ (denoted as $CeO_2$-$Ru_{SA}$) are also prepared as described in Methods (Supplementary Figs. 2–4). In this work, both $CeO_2$@$Cu_2O$ and $CeO_2$@CuO are recognized as electron-donable substrates; in order to distinguish them, if not specifically stated, the EDHJ group refers to $CeO_2$@$Cu_2O$.

The crystalline structure of Ru-Cu/EDHJ is clarified by the powder X-ray diffraction (XRD); the peaks (28.55°, 33.08°, 47.48°, 56.33°) can be assigned to (111), (200), (220), (311) planes of $CeO_2$ (JCPDS 34-0394), and peaks (36.42°, 42.30°, 61.34°) can be ascribed to (111), (200), (220) planes of $Cu_2O$ (JCPDS 05-0667, Fig. 2b). Similarly, the XRD peaks of Ru-O/EDHJ confirm the two-phase heterojunction structure ($CeO_2$ and CuO, Supplementary Fig. 3). The scanning electron microscope (SEM) and transmission electron microscopy (TEM) images demonstrate that Ru-Cu/EDHJ and Ru-O/EDHJ present spherical structures with an average size of 65.33 nm and 58.46 nm, respectively (Supplementary Figs. 4–6).

Figure 2c, d depict the proposed Ru binding sites on heterojunctions and atomic structure difference of Ru-Cu/EDHJ and Ru-O/EDHJ, which will be systematically verified in the following sections. The aberration-corrected high-angle annular dark-field scanning TEM (HAADF-STEM) images are collected to investigate the atom distributions. For the Ru-Cu/EDHJ, it is revealed that $Cu_2O$ clusters with a size of ~5 nm (Fig. 2e) are supported on the surface of $CeO_2$ (Fig. 2f). Then, atomic-resolution images are analyzed as shown in Fig. 2g, it is found

**a**  Structure design of the biocatalysts

**b**  pH-controlled ROS biocatalysis of Ru-Cu/EDHJ

**c**  Superior mandible defects repair and bioadaptive antibacterial properties

**Fig. 1 | Structure design and biocatalytic properties of Ru-Cu/EDHJ. a** Structural and biocatalytic advantages of Ru-Cu/EDHJ when compared with Ru-O/EDHJ. **b** Graphical illustration for pH-controlled ROS biocatalysis of Ru-Cu/EDHJ. **c** The superior mandible defects repair and bioadaptive antibacterial properties of Ru-Cu/EDHJ.

that individual Ru atoms are homogenously dispersed on $Cu_2O$ (indicated by yellow circles). For the Ru-O/EDHJ, CuO clusters are also supported on $CeO_2$, and abundant atomic Ru can be found on the CuO, which is similar to the Ru-Cu/EDHJ (Fig. 2i–k). It should be noted that the Ru atoms may also deposit onto the $CeO_2$ surfaces, similar to the reference sample ($CeO_2$-$Ru_{SA}$, Supplementary Figs. 7–9). Further evidence is given by atomic-resolution energy-dispersive X-ray spectroscopy mapping (Fig. 2h, l), proving the atomic distribution of Ru atoms on the heterojunctions.

The surface chemical compositions and electronic states are conducted by X-ray photoelectron spectroscopy (XPS)[59–61]. The XPS survey spectra confirm the presence of O, Ce, Cu, and Ru in Ru-Cu/EDHJ (Supplementary Figs. 10–14 and Table 1). In the high-resolution XPS, the peaks located at 484.42 eV and 462.22 eV are assigned to Ru $3p_{1/2}$ and Ru $3p_{3/2}$, suggesting the $Ru^{n+}$ oxidation state in Ru-Cu/EDHJ (Fig. 2m)[62]. Similarly, Ru atoms in Ru-O/EDHJ and $CeO_2$-$Ru_{SA}$ also possess a positive valence state (Fig. 2m and Supplementary Fig. 13). Notably, the Ru $3p$ peaks of Ru-Cu/EDHJ downshift about 1.0 eV compared to other materials, thus suggesting that there are pronounced electron transfer from substrate to Ru atoms in Ru-Cu/EDHJ and form the electron-rich Ru sites for efficient ROS catalysis. Meanwhile, for the Cu $2p$ of Ru-Cu/EDHJ, the typical peaks at 931.76 eV can be indexed to $Cu^+ 2p_{3/2}$, corresponding to the crystal structure of $Cu_2O$ (Fig. 2n and

Supplementary Fig. 12). While the Cu $2p$ of Ru-O/EDHJ exhibits higher binding energy at 933.29 eV, proving that Ru-O/EDHJ contains a large amount of $Cu^{2+}$ species. Moreover, the auger peak in Cu LMM indicates that there are no metallic $Cu^0$ species in Ru-Cu/EDHJ. The Ce $3d$ XPS spectra show that Ru-Cu/EDHJ and Ru-O/EDHJ have similar valence states of Ce (Fig. 2o).

Thereafter, the X-ray absorption near-edge structure (XANES) spectroscopy and extended X-ray absorption fine structure (EXAFS) spectroscopy are applied to disclose the precise atomic coordination environments of Ru centers[63–65]. The Ru $K$-edge XANES proves that the absorption edge position of Ru-Cu/EDHJ is between $RuO_2$ and Ru foil, which is also much lower than that of Ru-O/EDHJ, thus confirming the cationic Ru centers in Ru-Cu/EDHJ has higher electron density than Ru in Ru-O/EDHJ (Fig. 3a)[66,67]. Then, we simulate these structures via the density functional theory (DFT) method to give the theoretical evidence. The Bader charge analysis demonstrates that the Ru site in Ru-Cu/EDHJ loses the fewest electrons (−0.75 |e|), exhibiting a much higher electron density than that in Ru-O/EDHJ (Fig. 3b and Supplementary Fig. 15).

Furthermore, we studied the fine coordination structures of Ru sites to reveal the charge interaction between Ru and coordinated Cu atoms. In the Fourier transform EXAFS spectra, the Ru-Cu/EDHJ shows two peaks at ~1.55 Å and ~2.12 Å, which can be indexed to the Ru-O

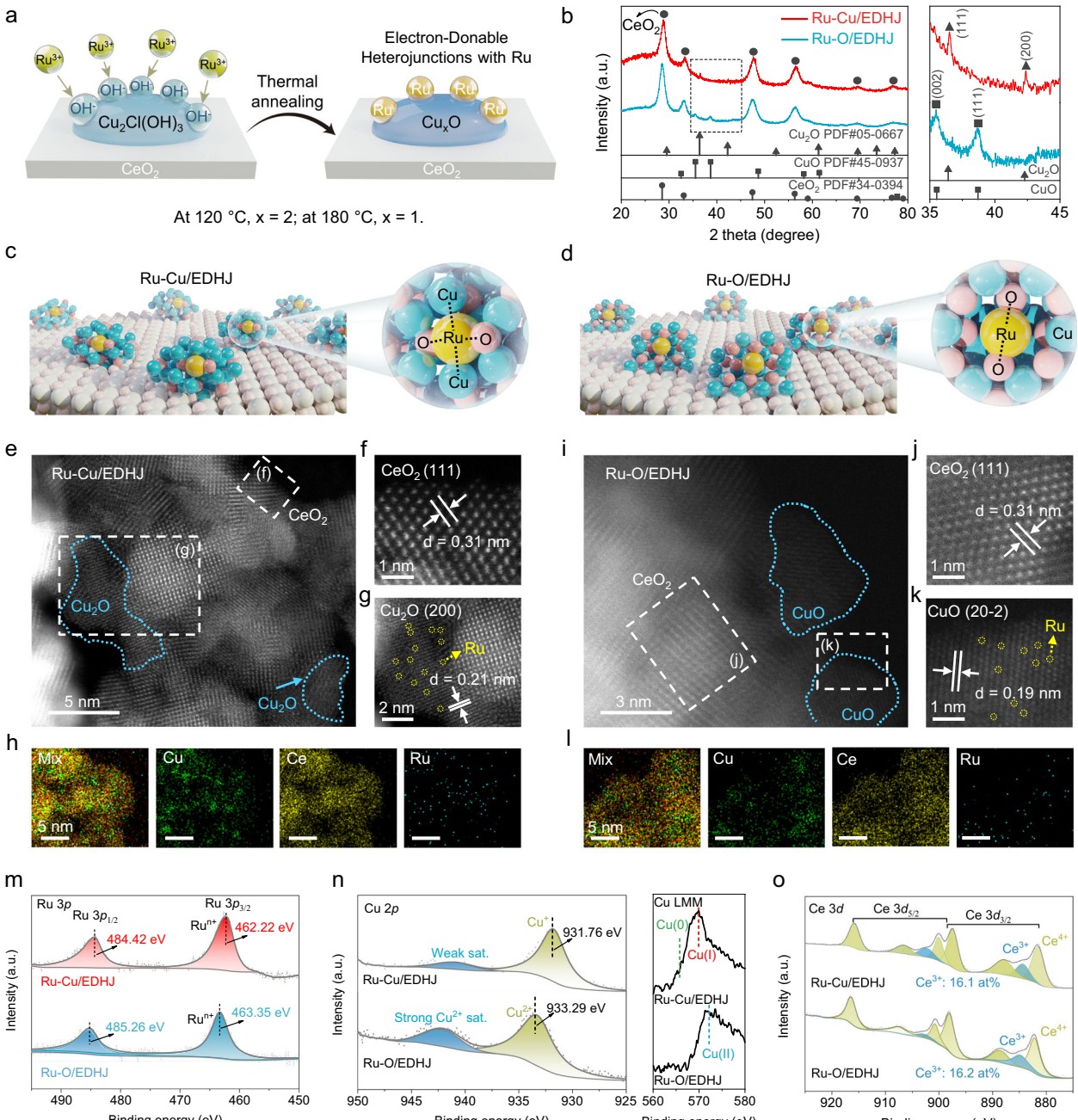

**Fig. 2 | Crystal structure characterizations of biocatalytic heterojunctions.**
**a** Schematic preparation processes for Ru-Cu/EDHJ and Ru-O/EDHJ. **b** XRD spectra of Ru-Cu/EDHJ and Ru-O/EDHJ. The atomic structure difference of (**c**) Ru-Cu/EDHJ and (**d**) Ru-O/EDHJ. Atomic color coding in the structure: Ru, yellow; Cu, blue; Ce, gray; and O, pink. Atomic-scale HAADF-STEM images of (**e**–**g**) Ru-Cu/EDHJ and (**i**–**k**) Ru-O/EDHJ, atomically dispersed Ru atoms are marked in (**g**, **k**) by yellow circles. Elemental mapping of (**h**) Ru-Cu/EDHJ and (**l**) Ru-O/EDHJ. The high-resolution XPS of (**m**) Ru 3*p*, (**n**) Cu 2*p*, Cu LMM, and (**o**) Ce 3*d* of Ru-Cu/EDHJ and Ru-O/EDHJ. Experiments were repeated independently (**e**, **f**, **g**, **h**, **i**, **j**, **k**, **l**) three times with similar results. In (**b**, **m**, **n**, **o**), a.u. indicates the arbitrary units. In (**n**), sat. indicates the satellite peaks. In (**o**), at% indicates the atomic percent. Source data are provided as a Source Data file.

bonds and Ru-Cu bonds, respectively (Fig. 3c), verifying the existence of Ru-Cu pair sites. For Ru-O/EDHJ, there is only the Ru-O peak, indicating that Ru mainly coordinates with O atoms. Moreover, we also compare the results with theoretical radial distances of possible Ru-bonding structures (DFT method, Fig. 3c), including Ru-O-Cu (1.95 Å in model and 1.55 Å in *R* space), Ru-O-Ce (2.15 Å in model and 1.87 Å in *R* space), Ru-Cu (2.53 Å in model and 2.12 Å in *R* space), and Ru-Ce (3.67 Å in model and 3.28 Å in *R* space). After comparison, the Ru-Cu/EDHJ shows both Ru-O-Cu and Ru-Cu moieties, and Ru-O/EDHJ shows only Ru-O-Cu moieties, confirming that there are Ru-Cu pair sites. Furthermore, the EXAFS fitting results and corresponding wavelet-

transform (WT) images are summarized in Fig. 3d, e and Supplementary Table 2, which also disclose the existence of Ru-O and Ru-Cu bonds in Ru-Cu/EDHJ with average coordination numbers of 1.98 and 2.00, respectively.

Then, we use the theoretical calculations to further investigate charge interactions between the Ru-Cu coordination structure (Supplementary Data 1–3). Valence states analysis indicates that Ru atom in Ru-Cu/EDHJ has almost one electron more than that in Ru-O/EDHJ due to the Ru-Cu coordination structure (Fig. 3g, Supplementary Fig. 16). Correspondingly, the partial density of states (PDOS) of Ru 4*d* orbital shows that the extra electron of Ru atom in Ru-Cu/EDHJ occupies

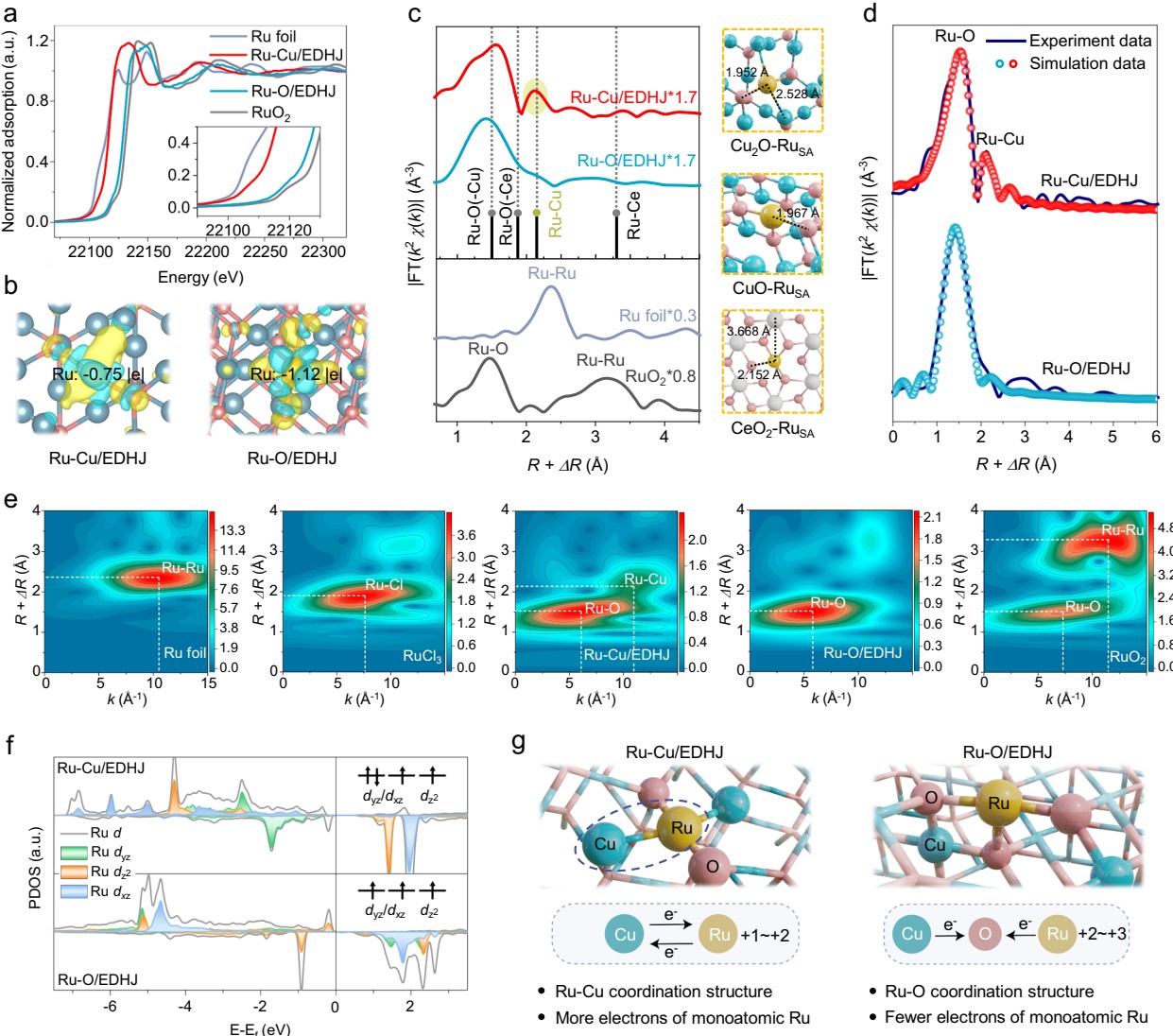

**Fig. 3 | Analysis of precise atomic coordination environments of biocatalytic heterojunctions. a** Ru *K*-edge XANES spectra of Ru-Cu/EDHJ, Ru-O/EDHJ, and references (Ru foil and RuO₂). **b** Differential charge density analysis of Ru centers (yellow and cyan represent charge accumulation and depletion, respectively, with a cutoff value of 0.006 e·Bohr⁻³ for the density-difference isosurface). Atomic color coding in the structure: Ru, yellow; Cu, blue; and O, pink. **c** Fourier-transformed $k^2$-weighted EXAFS spectra. Right: optimized structure models for a Ru atom that anchors on different metal oxide surfaces. Atomic color coding in the structure: Ru,

yellow; Cu, blue; Ce, gray; and O, pink. **d** Experimental and fitting EXAFS results of Ru-Cu/EDHJ and Ru-O/EDHJ. **e** WT analysis at the Ru *K*-edge of different samples. **f** PDOS analysis and molecular orbital diagrams of Ru 4*d* orbital of Ru-Cu/EDHJ and Ru-O/EDHJ. **g** Schematic diagram of the coordination environments and electronic states of Ru centers of Ru-Cu/EDHJ and Ru-O/EDHJ. Atomic color coding in the structure: Ru, yellow; Cu, blue; and O, pink. In (**a**, **f**), a.u. indicates the arbitrary units. Source data are provided as a Source Data file.

$d_{yz}/d_{xz}$ orbital, which can be proved by magnetic moment analysis (Fig. 3f and Supplementary Figs. 17, 18). The above experimental and theoretical results clearly confirm that the synthesized electron-donable heterojunctions with Ru-Cu pair sites are favorable for enhancing the electron density of Ru atoms via direct interaction between Ru and Cu (Fig. 3g).

**Experimental and theoretical study on antioxidase-like activities**
After successfully disclosing the morphology and electronic structures of the synthesized electron-donable heterojunctions, especially its Ru-Cu pair sites, we further tested and compared its antioxidase-mimetic activities. In nature, the SOD and CAT antioxidases can form a cascaded biocatalytic pathway[14]. The SOD transforms the •O₂⁻ to H₂O₂ and O₂, then CAT reacts with H₂O₂ to produce H₂O and O₂. As shown in Fig. 4a, the Ru-Cu/EDHJ also exhibits a cascaded biocatalytic process; it is tested that the Ru-Cu/EDHJ can remove ~62 % of •O₂⁻ within 5 min,

which is much superior to that of Ru-O/EDHJ, EDHJ, and bare CeO₂ (Fig. 4b). For the CAT-like reaction process, the Ru-Cu/EDHJ can dramatically reduce H₂O₂ concentration (~80% at 40 min), which remarkably outperforms the Ru-O/EDHJ, EDHJ, and bare CeO₂ (Fig. 4c, d, and Supplementary Fig. 19). Meanwhile, the generation O₂ in CAT-like reaction gives the similar trends, the Ru-Cu/EDHJ shows significantly increased O₂ concentration from 5.25 to 28.49 mg/mL at 3 min, which is much faster than the Ru-O/EDHJ and other references (Fig. 4e).

For further comparison, the CAT-like reaction kinetics of producing O₂ have also been performed according to the relationship between the substrate doses and velocities. Compared to the Ru-O/EDHJ, the Ru-Cu/EDHJ exhibits a relatively smaller $K_m$, indicating its exceptional substrate affinity (Fig. 4f and Supplementary Table 3). Moreover, the maximal reaction velocity ($V_{max}$, a steady-state kinetic parameter) calculated by Michaelis–Menten fitting (Supplementary

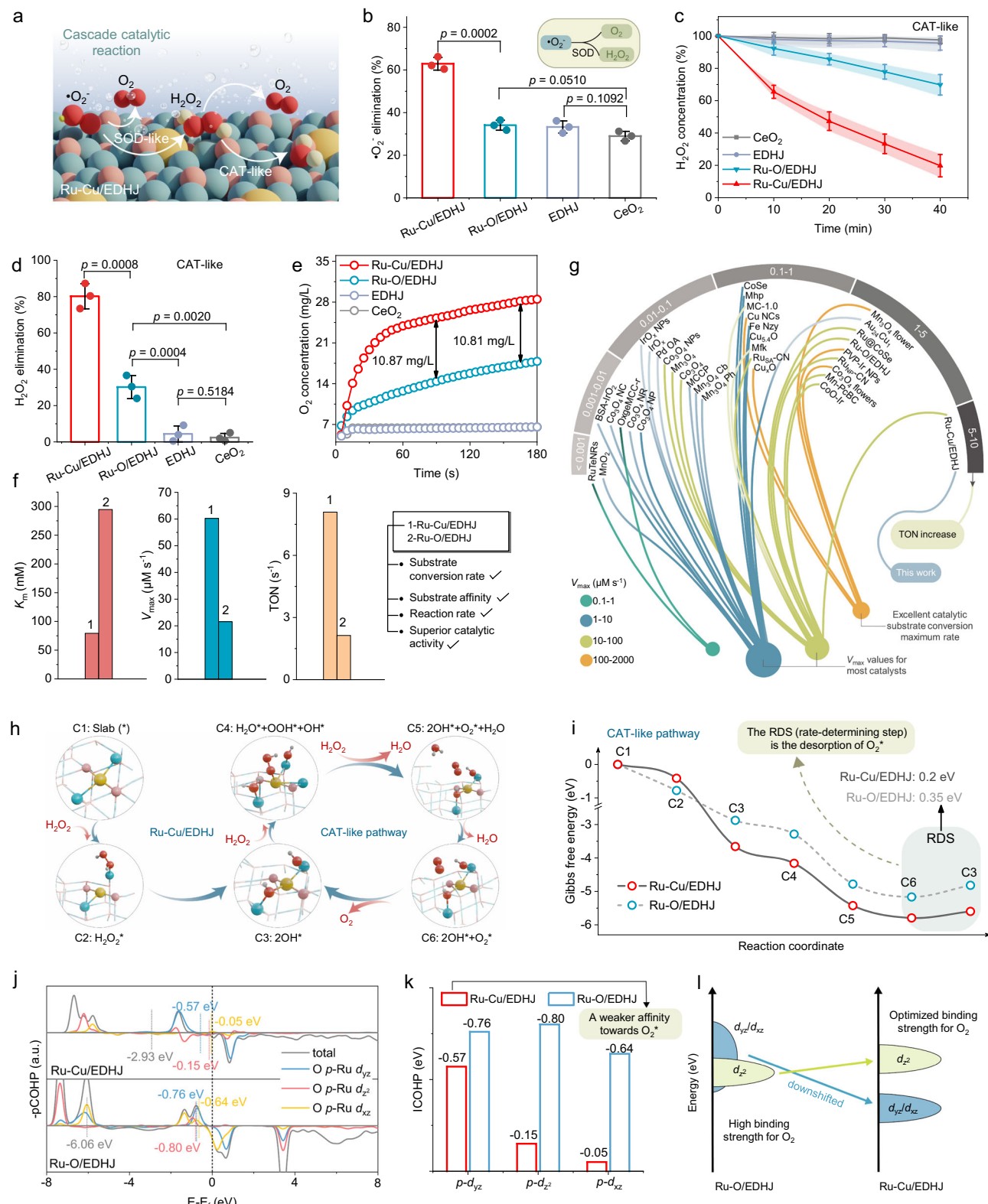

Fig. 20) demonstrates that Ru-Cu/EDHJ presents a much higher $V_{max}$ value (60.24 μM s$^{-1}$), which surpasses that of Ru-O/EDHJ by 2.79-fold. When normalized the amounts of Ru catalytic centers, the turnover number (TON, maximum number of converting substrates per unit catalytic sites) of Ru-Cu/EDHJ is 8.09 s$^{-1}$, which surpasses that of Ru-O/EDHJ by 3.79-fold. Notably, when compared with recently reported antioxidase-mimetic biocatalysts, the Ru-Cu/EDHJ exhibits the highest TON values among these currently established CAT-mimics

(Fig. 4g and Supplementary Data 4). The above results suggest that Ru-Cu/EDHJ with Ru-Cu pair sites show superior CAT-like kinetics than Ru-O sites. Moreover, we also demonstrated that the bare CeO$_2$ nanoparticles, CeO$_2$-Ru$_{SA}$, and EDHJ all display very low antioxidase-like activities (Supplementary Figs. 21, 22 and Table 3), thus demonstrating that the Ru-Cu pair sites are the active centers and EDHJ (CeO$_2$@Cu$_2$O) substrate significantly boost the catalytic activity.

**Fig. 4 | Experimental analysis and theoretical calculation on antioxidase-mimetic activities. a** Schematic diagram of the cascaded catalysis of SOD-like and CAT-like reactions. Atomic color coding in the structure: Ru, yellow; Cu, blue; O in Ru-Cu/EDHJ, pink; O in substrates, red; and H, white. **b** SOD-like activity of different materials (if not specifically stated, EDHJ group refers to $CeO_2@Cu_2O$; $n = 3$ independent experiments, data are presented as mean ± SD). **c** Time-dependent CAT-like performances via $TiSO_4$-based method with the presence of biocatalysts and $H_2O_2$ ($n = 3$ independent experiments, data are presented as mean ± SD). **d** $H_2O_2$ elimination ratio of different materials ($n = 3$ independent experiments, data are presented as mean ± SD). **e** The produced $O_2$ concentration was measured by an oxygen dissolving meter with the presence of biocatalysts and $H_2O_2$. **f** $V_{max}$, $K_m$, and TON values of Ru-Cu/EDHJ and Ru-O/EDHJ. **g** Comparative analysis of the $V_{max}$ and TON with previously reported biocatalysts. **h** CAT-like pathways (atomic color coding in the structure: Ru, yellow; Cu, blue; O in Ru-Cu/EDHJ, pink; O in substrates, red; and H, white) and (**i**) Gibbs free energy diagram of Ru-Cu/EDHJ and Ru-O/EDHJ. **j** Calculated-pCOHP of O $2p$ orbital and Ru $4d$ orbital in $O_2*$ and (**k**) corresponding ICOHP values of Ru-O bonds. **l** Schematic illustration of $d$ orbital energy levels varies with coordination configurations. $V_{max}$ is the maximal reaction velocity, $K_m$ is the Michaelis constant, TON is the turnover number. In (**b**, **d**), statistical significance was calculated using a two-tailed Student's t-test; all tests were two-sided. In (**j**), a.u. indicates the arbitrary units. Source data are provided as a Source Data file.

Thereafter, DFT calculations are performed to reveal the intrinsic correlation between the Ru-Cu coordination structure and CAT-like catalytic activity. First, the catalytic pathways of the CAT-like reaction are revealed in Fig. 4h and Supplementary Fig. 23. The rate-determining steps (RDS) for both CAT-like paths are the desorption of $O_2*$ with reaction energy barriers of about 0.20 eV for Ru-Cu/EDHJ and 0.35 eV for Ru-O/EDHJ (Fig. 4i). The results suggest the presence of Ru-Cu coordination tailors the $d$-orbital energy level of Ru, thus reducing $O_2*$ adsorption and consequently leading to a significantly lower energy barrier. Since $O_2$ is a critical production in antioxidase-like catalysis (Fig. 4a) and the desorption of $O_2*$ is found to be a pivotal step (Fig. 4i), we then take $O_2*$ as an ideal and essential model to deeply investigate the inherent connection between Ru-Cu coordination and catalytic properties.

As presented in Supplementary Fig. 24, when compared to the Ru-O/EDHJ, there is a weaker electronic interaction between Ru in Ru-Cu/EDHJ and $O_2*$, which can be visualized by electron density difference diagrams. Further PDOS analysis of Ru $4d$ and O $2p$ orbitals before the interaction between Ru and $O_2*$ indicates that the orbital components of $O_2*$ involved in the adsorption process are mainly $\pi_{2p}*$ orbitals (Supplementary Figs. 25, 26). Besides, considering the symmetry matching, the interactions between $d_{x2-y2}$ and $d_{xy}$ orbitals of Ru and orbitals of $O_2*$ are negligible. Owing to the extra electron for Ru in Ru-Cu/EDHJ, occupying $d_{yz}/d_{xz}$ orbital, the molecular orbital arrangements suggest that there are more electrons occupying the antibonding orbitals when Ru interacts with $O_2*$, resulting in reduced system stability, which weakens the interaction between Ru and $O_2*$, as displayed in Supplementary Fig. 27. Additionally, the projected Crystal Orbital Hamilton Population (pCOHP, a quantitative analysis of the bonding/antibonding characteristics in the electronic state, Fig. 4j) shows that all the bonding orbitals situated below the Fermi level are completely occupied for Ru-Cu/EDHJ and Ru-O/EDHJ. Notably, more antibonding states are occupied for Ru-Cu/EDHJ than Ru-O/EDHJ. The higher integrated COHP (ICOHP) value of Ru-O bonding reaching the Fermi level in Ru-Cu/EDHJ further suggests a weaker affinity towards $O_2*$ (Fig. 4k). Moreover, we calculate the adsorption energy of different oxygen intermediates on Ru sites (Supplementary Fig. 28), Ru-Cu/EDHJ exhibits weaker interactions with them due to its higher electron density resulting from Ru-Cu coordination. Together, benefiting from the electronic structure regulation of Ru-Cu pair sites, the decreased $d$-orbitals energy level empowers an optimized binding strength of Ru sites for all oxygen species, thus enhancing the antioxidase-like catalytic performances for Ru-Cu/EDHJ (Fig. 4l).

In addition to SOD-like and CAT-like activities, we also investigated the glutathione peroxidase (GPx)-like activity. Our results (Supplementary Fig. 29) demonstrate that Ru-Cu/EDHJ displays the highest GPx-like reaction rate compared to other enzyme-mimetic biocatalysts. Furthermore, an overall evaluation of the antioxidant activities of biocatalysts has been conducted by using the DPPH• (1,1-diphenyl-2-picrylhydrazyl radical) assay, which also validates the Ru-Cu/EDHJ owns rapid elimination capability to all types of free radicals (Supplementary Fig. 30). These findings suggest that Ru-Cu/EDHJ presents multifunctional antioxidase-like properties for exceptional free radical scavenging due to the unique Ru-Cu pair sites.

## In vitro ROS scavenging and stem cell protection of Ru-Cu/EDHJ

After validating that the Ru-Cu/EDHJ reveals remarkable and versatile antioxidase-mimetic activities, we then systematically examine its application potentials for ROS scavenging and stem cell protection via using the mesenchymal stem cells (MSCs) from bone marrow as model systems. The concentration of 100 μM $H_2O_2$ in vitro is chosen according to the estimated physiological concentration during the inflammatory reaction[68,69]. 2,7-dichlorodihydrofluorescein diacetate (DCFH-DA) is applied as the fluorescence probe to explore intracellular ROS levels, as shown in Fig. 5a–c and Supplementary Figs. 31, 32. After treating with $H_2O_2$, apparent green fluorescence can be observed in both human mesenchymal stem cells (hMSCs) and mouse mesenchymal stem cells (mMSCs), demonstrating a high level of intracellular ROS. It can be noted that the green fluorescence signals can be dramatically attenuated after being treated with Ru-Cu/EDHJ, while the Ru-O/EDHJ can only slightly decrease the signal, and the $CeO_2$ or EDHJ ($CeO_2@Cu_2O$) group shows no noticeable effects on decrease ROS level. Then, ROS-triggered stem cell damage is further marked by 4-hydroxynonenal (4-HNE, red signal), which is a biomarker of oxidative/nitrosation stress. Red fluorescence signals increase significantly after being treated with $H_2O_2$ (100 μM), and all of the Ru-O/EDHJ, $CeO_2$, or EDHJ can't effectively prevent the ROS-triggered stem cell damage (Fig. 5d, e). Remarkably, the Ru-Cu/EDHJ can efficiently protect hMSCs from ROS damage, and the red signal is as low as that of the control group.

The cell cytoskeleton and adhesion form a network that plays a crucial role in maintaining cellular structure and essential functions, which are vulnerable to oxidative stress. Evidence indicates that the disruption of cytoskeletal proteins is the initial step in cell damage caused by oxidative stress[70]. To investigate the influence of oxidative stress on the cytoskeleton, F-actin cytoskeleton stained by FITC (fluorescein isothiocyanate)-labeled phalloidin is detected in the high ROS level microenvironment to validate the protection activity of Ru-Cu/EDHJ (Fig. 5f). In the ROS condition, hMSCs display weak-spreading, shrinking morphology, and disordered cytoskeleton. By contrast, after treating with Ru-Cu/EDHJ, hMSCs with $H_2O_2$ can maintain well-spreading and polygonal morphology with highly elongated and well-organized actin filaments (green) surrounding the cell nuclei (blue). This phenomenon of cytoskeleton change is also reported in MCF-7 cells under $H_2O_2$ stimulation[40]. Paxillin is a kind of focal adhesion scaffold protein and participates in recruiting adhesion proteins and transmitting molecule signals[71]. As depicted in Fig. 5f, paxillin (red) shows immature and smaller spotty patterns in high ROS-level conditions ($H_2O_2$ group). However, after adding the Ru-Cu/EDHJ, paxillin shows significantly more mature dotted and diffuse expression along the cell margins. Furthermore, the quantitative analysis of the cell areas (Fig. 5g) and paxillin plaque areas (Fig. 5h) of stem cells also reveal that the Ru-Cu/EDHJ can efficiently protect hMSCs in comparison with the Ru-O/EDHJ and EDHJ group. The Cell Counting Kit-8 (CCK-8) method shows that the half maximal inhibitory concentration

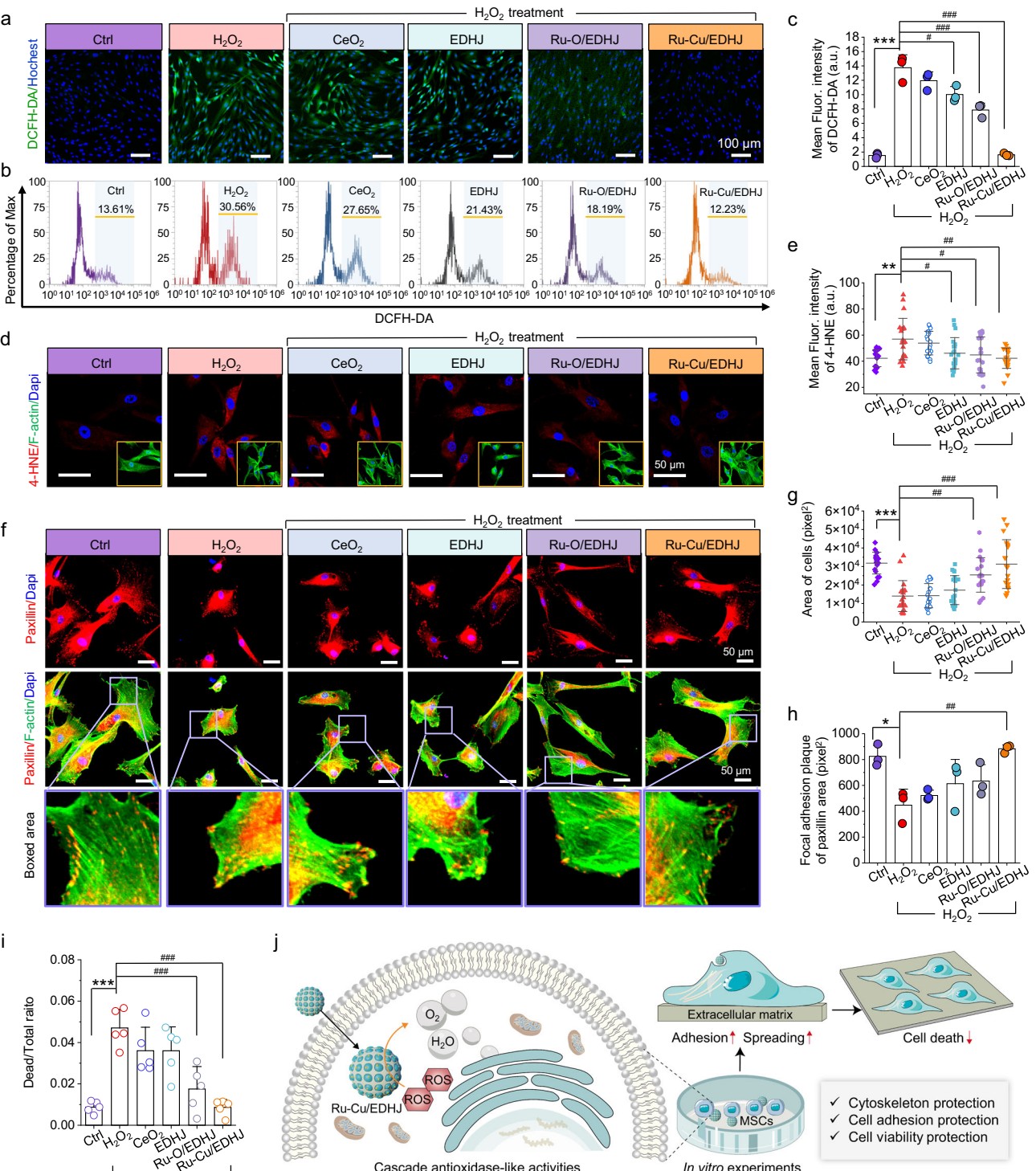

**Fig. 5 | In vitro ROS clearance and stem cell protection by Ru-Cu/EDHJ.**
**a** Fluorescence images, Ctrl (PBS), (**b**) flow cytometry, and (**c**) mean fluorescence intensity of DCFH-DA staining ($n = 3$ independent replicates), $p_{(H2O2)} < 0.0001$, compared with Ctrl group; $p_{(EDHJ-H2O2)} = 0.0118$, $p_{(Ru-O/EDHJ-H2O2)} = 0.0003$, $p_{(Ru-Cu/EDHJ-H2O2)} < 0.0001$, compared with $H_2O_2$ group. Scale bar: 100 μm.
**d** Fluorescence images of 4-HNE (Green: F-actin, Red: 4-HNE, Blue: Dapi). Scale bar: 50 μm. **e** Quantitative analysis of mean fluorescence intensity of 4-HNE staining ($n = 20$ independent replicates), $p_{(H2O2)} = 0.0011$, compared with Ctrl group; $p_{(EDHJ-H2O2)} = 0.0359$, $p_{(Ru-O/EDHJ-H2O2)} = 0.0121$, $p_{(Ru-Cu/EDHJ-H2O2)} = 0.0010$, compared with $H_2O_2$ group. **f** Fluorescence images of paxillin (Green: F-actin, Red: paxillin, Blue: Dapi). Scale bar: 50 μm. **g** Quantitative analysis of the area of cells ($n = 20$ independent replicates), $p_{(H2O2)} < 0.0001$, compared with Ctrl group; $p_{(Ru-O/EDHJ-H2O2)} = 0.0011$, $p_{(Ru-Cu/EDHJ-H2O2)} < 0.0001$, compared with $H_2O_2$ group.

**h** Quantitative analysis of focal adhesion plaque of paxillin area ($n = 3$ independent replicates), $p_{(H2O2)} = 0.0144$, compared with Ctrl group; $p_{(Ru-Cu/EDHJ-H2O2)} = 0.0050$, compared with $H_2O_2$ group. **i** The dead cell ratio counted with the treatment of materials and $H_2O_2$ after live/dead staining ($n = 5$ independent replicates), $p_{(H2O2)} = 0.0001$, compared with Ctrl group; $p_{(Ru-O/EDHJ-H2O2)} = 0.0003$, $p_{(Ru-Cu/EDHJ-H2O2)} < 0.0001$, compared with $H_2O_2$ group. **j** Schematic illustration of cell effects upon the addition of high-concentration $H_2O_2$ and Ru-Cu/EDHJ protection. Data are presented as means ± SD., $*p < 0.05$, $**p < 0.01$, $***p < 0.001$, $\#p < 0.05$, $\#\#p < 0.01$, $\#\#\#p < 0.001$, ns, no significance; statistical significance was calculated using one-way ANOVA followed by Tukey's post-hoc test for multiple comparisons, all tests were two-sided. In (**c**, **e**, **h**), a.u. indicates the arbitrary units. Experiments were repeated independently (**a**, **b**, **d**, **f**) three times with similar results. Source data are provided as a Source Data file.

(IC50) is 426.4 μg/mL and exhibits almost no cytotoxicity to cell viability in low concentration. The concentration value of 100 μg/mL used in this research shows no significant effect on cell viability by CCK-8 and live/dead staining, indicating the excellent biocompatibility of Ru-Cu/EDHJ biocatalyst (Supplementary Figs. 33, 34). After the hMSCs treatment with $H_2O_2$, Fig. 5i, j indicates that the Ru-Cu/EDHJ can maintain stem cell survival with the lowest ratio of dead cells.

Oxidative stress causes DNA damage in MSCs, and extensive DNA damage invariably leads to cell death, either caused by apoptosis or necrosis[72–74]. Thereafter, we systematically explore the catalytic therapeutic effects of Ru-Cu/EDHJ on preventing ROS-related DNA damage and apoptosis. As depicted in Fig. 6a, compared to the Ctrl group, the signals of DNA/RNA damage marker in nuclei significantly increase in the hMSCs+$H_2O_2$ group; the Ru-Cu/EDHJ shows a decreased red signal, which presents a similar fluorescence level as the Ctrl group (Fig. 6b). Then, we further test the phosphorylating of γ-H2A.X, a double-stranded DNA break marker. As the immunofluorescence staining (Fig. 6c) and the spectrum analysis of the nuclei results (Fig. 6d), the $H_2O_2$ group gives obvious γ-H2A.X signals co-express with Dapi, while the Ru-Cu/EDHJ group shows significantly decreased signals, which is as low as the Ctrl group as indicated by quantitative analysis in Fig. 6e, indicating the reduction of DNA damage under oxidative stress by Ru-Cu/EDHJ[75].

Cellular apoptosis occurs under an oxidative stress state, and TUNEL staining is used to detect the breakage of nuclear DNA during cell apoptosis[76]. As depicted in Fig. 6f, h, the Ru-Cu/EDHJ groups show evidently attenuated signal intensities compared to the $H_2O_2$ group, suggesting Ru-Cu/EDHJ protects stem cells from apoptosis. Furthermore, the flow cytometry data are analyzed (Fig. 6g, h and Supplementary Figs. 35, 36), and cell populations are assigned into four stages, including live, early apoptotic, late apoptotic, and necrotic stages. The cells treated with $H_2O_2$ show an obvious apoptosis ratio, while the Ru-Cu/EDHJ group displays a similar ratio as the Ctrl group (Fig. 6i), indicating its efficient protection from oxidative stress-induced cell apoptosis[77].

To explore the protection of Ru-Cu/EDHJ on stem cell function, the in vitro osteogenic differentiation potential of MSCs in high ROS levels is assessed by alizarin red (AR) and alkaline phosphatase (ALP) staining[78]. The group treated with $H_2O_2$ demonstrates a reduced expression of ALP and OCN (osteocalcin) and a decreased formation of calcium nodules by AR staining (Fig. 6j–l and Supplementary Figs. 37, 38). However, the impaired osteogenic phenotype induced by $H_2O_2$ is found to recover with the addition of Ru-Cu/EDHJ. Quantitative analysis of the ALP and AR staining areas indicates that elevated $H_2O_2$ levels do not inhibit both hMSCs and mMSCs osteogenesis when supplemented with Ru-Cu/EDHJ. These findings suggest that treatment with Ru-Cu/EDHJ effectively protects the osteogenesis of MSCs from bone marrow in a high ROS environment, indicating its support to bone regeneration (Fig. 6m)[78]. Furthermore, it is discovered that the $H_2O_2$-treated group demonstrates a weakened expression of COL1A1, an early marker of osteogenesis, which can be rescued by adding Ru-Cu/EDHJ (Supplementary Fig. 39).

To investigate the hMSCs protection mechanisms, the Ctrl (PBS), $H_2O_2$, $CeO_2$ ($H_2O_2$ + $CeO_2$), and Ru-Cu/EDHJ ($H_2O_2$+Ru-Cu/EDHJ) treated hMSCs are collected respectively for transcriptomics analysis. The unguided principal component analysis (PCA) of transcriptomic data suggests good biological repetition (Fig. 7a). It also indicates obvious transcriptome profile differences in Ctrl vs. $H_2O_2$ and Ctrl vs. $CeO_2$, with a significant distance on the principal component 1 (PC1). However, hMSCs in the Ru-Cu/EDHJ group show similar results to the Ctrl, implying its function in protecting stem cells in high ROS-level conditions.

Then, the differentially expressed genes (DEGs) are collected for hierarchical clustering analysis (Fig. 7b). Ru-Cu/EDHJ-treated hMSCs exhibit a gene expression pattern similar to the Ctrl (healthy hMSCs),

which is evidently different from $H_2O_2$-treated hMSCs. Furthermore, the relevant top-enriched Kyoto Encyclopedia of Genes and Genomes (KEGG) pathways in $H_2O_2$ vs. Ru-Cu/EDHJ are analyzed (Fig. 7c). It speculates that the Ru-Cu/EDHJ could control the oxidative stress of hMSC (FoxO signaling pathway)[79] in the ROS-contained microenvironment. The function of Ru-Cu/EDHJ can be reflected in influencing cell cytoskeleton and adhesion (regulation of actin cytoskeleton, focal adhesion, Rap1 signaling pathway[80]), cellular DNA damage and function, cellular activity (ECM-receptor interaction, cellular senescence, cell cycle), and cell apoptosis (apoptosis, p53 signaling pathway[81]). In addition, the Ru-Cu/EDHJ can protect stem cells by changing inflammatory expression (inflammatory mediator regulation of TRP channels, NF-kappa B signaling pathway[82], IL-17 signaling pathway[83], JAK-STAT signaling pathway[84], and TNF signaling pathway[85]), stem cell development and matrix formation (signaling pathways regulating pluripotency of stem cells, ECM-receptor interaction, and MAPK signaling pathway[86]), and subsequently osteogenesis regulation (Wnt signaling pathway[87], PI3K-Akt signaling pathway[88], and TGF-beta signaling pathway[89]). The relevant top-enriched GO (Gene Ontology) terms also speculated similar cellular biological processes (Fig. 7d).

Furthermore, gene set enrichment analysis (GSEA) is carried out to reveal the gene signatures of a series of biological processes (Fig. 7e). The downregulation of "Response to hydrogen peroxide" and upregulation of "Actin Filament Organization" and "Cell Adhesion" are consistent with the observations of stem cell morphology and focal adhesions maintaining by Ru-Cu/EDHJ in Fig. 4. Compared to the $H_2O_2$ group, the Ru-Cu/EDHJ group shows the increase of "DNA replication" and the reduction of "Intrinsic apoptotic signaling pathway in response to DNA damage" and "Positive regulation of apoptotic process", suggesting its stem cell protection activity by reducing DNA damage and apoptosis from oxidative stress. Furthermore, Ru-Cu/EDHJ treatment reduces the "Inflammatory response", promotes "Extracellular matrix organization", and enhances the "Positive regulation of canonical Wnt signaling pathway" for osteogenesis.

## Biocatalytic regeneration of inflammatory mandible defects

Persistent inflammation in infected areas can result in continued inflammatory response and adversely affect the regeneration of bone tissue[90]. Inspired by the efficient and synergistic stem cell protection activities, we further assess the in vivo efficiency of Ru-Cu/EDHJ in treating bone tissue regeneration under inflammatory environments. Initially, we construct a lipopolysaccharide (LPS)-induced inflammatory periapical bone defect (Supplementary Fig. 40)[91], which is a ROS-enhanced mandible defect model (named as ROSup), evidenced by the fluorescence of ROS indicator (dihydroethidium (DHE) and DCFH-DA, Supplementary Fig. 41). This periapical bone defect construction mimics clinic infectious diseases inducing buccal bone wall defect in the tooth apical region, such as periapical inflammation, jaw osteomyelitis, and periodontitis[11].

Next, we investigate the inflammatory elimination and early osteogenesis by Ru-Cu/EDHJ treatment after mandible defect surgery for 1 week as the inflammation and regeneration initiated in this period (Fig. 8c). The fluorescence images (Fig. 8a) and quantitative analysis (Fig. 8d) reveal Ru-Cu/EDHJ significantly reduce proinflammatory cytokines (TNF-α) in the bone defect area. Moreover, we have included an analysis of the CD206 (M2 phenotype) and iNOS (M1 phenotype) markers through tissue section immunofluorescence double staining and quantitative assessment in the mandible defect area (Supplementary Fig. 42). The resulting data indicate that the Ru-Cu/EDHJ group shows a smaller proportion of iNOS relative to CD206 compared to the other experimental groups, suggesting its efficacy in promoting macrophage polarization toward the M2 (anti-inflammatory) phenotype and attenuating proinflammatory responses[92].

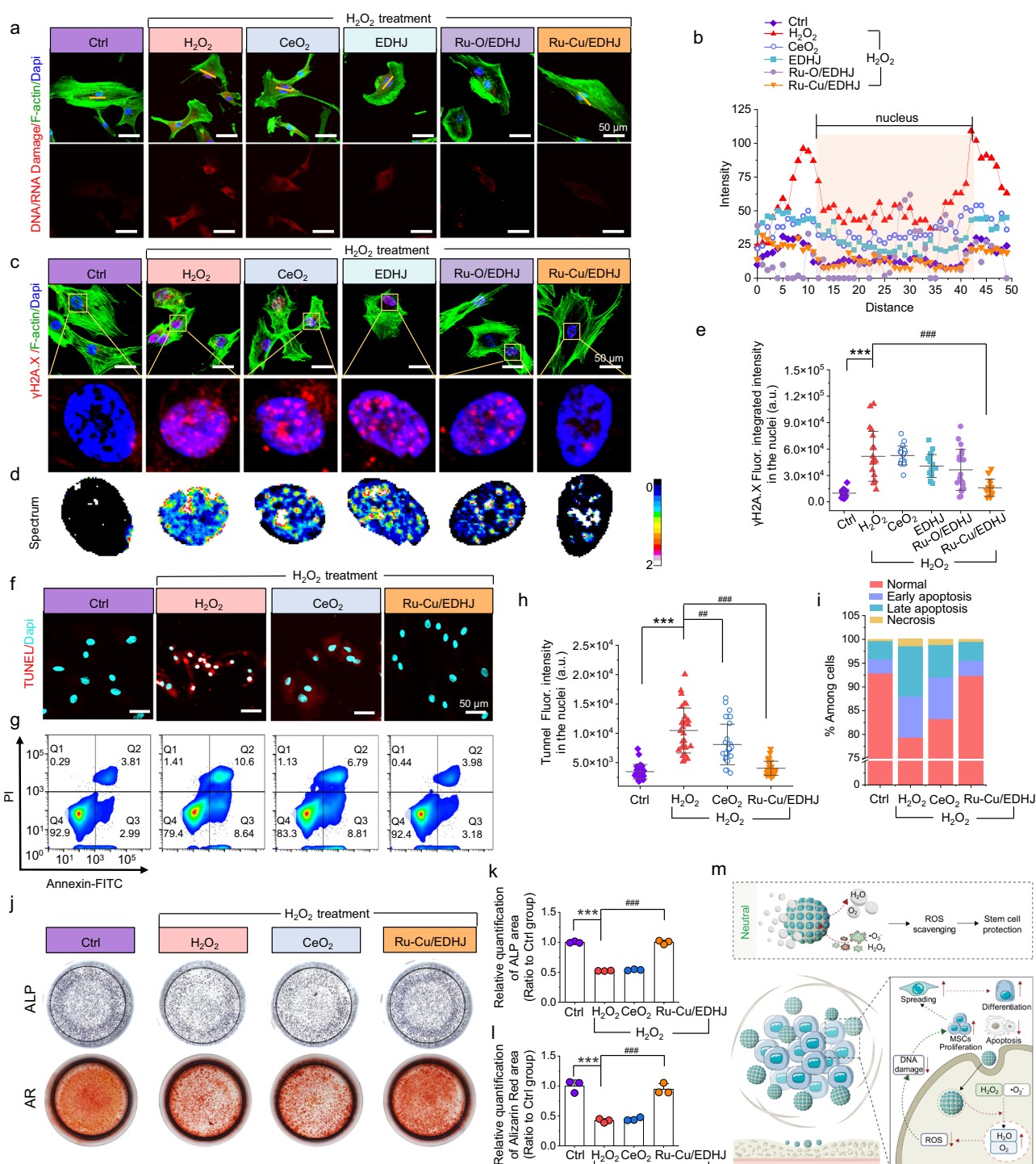

**Fig. 6 | Prevention of ROS-related DNA damage and apoptosis in MSCs by Ru-Cu/EDHJ. a** Fluorescence images and (**b**) linear distribution of fluorescence intensity from DNA/RNA damage staining. Scale bar: 50 μm. **c** Fluorescence images, (**d**) spectrum images, and (**e**) quantitative analysis of fluorescence intensity from γ-H2A.X ($n = 20$ independent replicates), $p_{(H2O2)} < 0.0001$, compared with Ctrl group; $p_{(Ru-Cu/EDHJ-H2O2)} < 0.0001$, compared with $H_2O_2$ group. Scale bar: 50 μm. **f** Confocal scanning images from TUNEL. Scale bar: 50 μm. **g** Apoptosis analysis was performed using flow cytometry of Annexin V-FITC/PI stained hMSCs. **h** Quantitative fluorescence analysis from TUNEL ($n = 30$ independent replicates), $p_{(H2O2)} < 0.0001$, compared with Ctrl group; $p_{(CeO2-H2O2)} = 0.0052$, $p_{(Ru-Cu/EDHJ-H2O2)} < 0.0001$, compared with $H_2O_2$ group. **i** The cell percentages in stages of normal, early apoptosis, late apoptosis, and necrosis. **j** ALP staining after 3-day in vitro osteo-induction and

AR staining after 21-day in vitro osteo-induction using the hMSCs. Quantitative analysis of (**k**) ALP ($n = 3$ independent replicates), $p_{(H2O2)} < 0.0001$, compared with Ctrl group; $p_{(Ru-Cu/EDHJ-H2O2)} < 0.0001$, compared with $H_2O_2$ group. Quantitative analysis of (**l**) AR ($n = 3$ independent replicates) in $H_2O_2$ treated hMSCs, $p_{(H2O2)} < 0.0001$, compared with Ctrl group; $p_{(Ru-Cu/EDHJ-H2O2)} < 0.0001$, compared with $H_2O_2$ group. **m** Schematic illustration of apoptosis and osteogenesis effects with Ru-Cu/EDHJ in high ROS-level microenvironment. Data are presented as means ± SD., *$p < 0.05$, **$p < 0.01$, ***$p < 0.001$, #$p < 0.05$, ##$p < 0.01$, ###$p < 0.001$, ns, no significance; statistical significance was calculated using one-way ANOVA followed by Tukey's post-hoc test for multiple comparisons, all tests were two-sided. Experiments were repeated independently (**a**, **c**, **d**, **f**, **g**, **j**, **l**) three times with similar results. Source data are provided as a Source Data file.

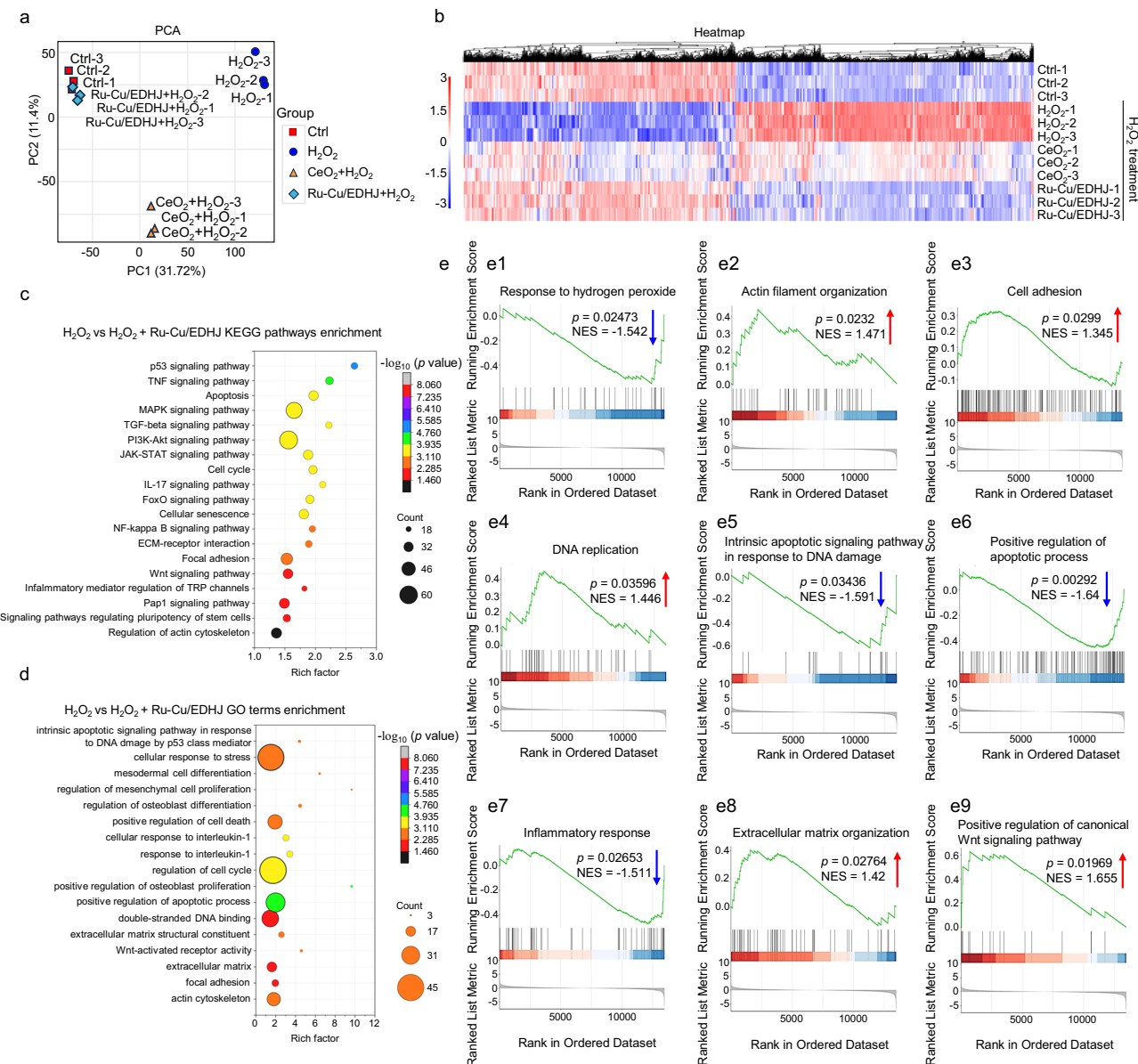

**Fig. 7 | RNA sequencing analysis of hMSCs in high ROS-level microenvironment. a** PCA analysis between samples. **b** Hierarchical clustering of differentially expressed genes from the hMSCs after different treatments. **c** Enriched KEGG pathways of $H_2O_2$ versus Ru-Cu/EDHJ + $H_2O_2$. **d** Enriched GO terms of $H_2O_2$ versus Ru-Cu/EDHJ + $H_2O_2$. **e** (**e1**–**e9**) GSEA analysis of $H_2O_2$ versus Ru-Cu/EDHJ + $H_2O_2$. In (**c**, **d**), *p*-value obtained from one-sided Hypergeometric test without multiple comparisons. In (**e**), *p*-value obtained from one-sided Permutation test without multiple comparisons. In (**e**), NES indicates the normalized enrichment score. Experiments were repeated independently three times with similar results. Source data are provided as a Source Data file.

Then, the early osteogenic differentiation stage is detected by bone morphogenetic protein 2/4 (BMP-2/4) expression, which displays that the Ru-Cu/EDHJ group has the highest production of BMP-2/4 (Fig. 8b, e), thus reflecting its excellent osteoinductive potential. The images by confocal scanning of CD140a and 4-HNE dual staining and quantitative results (Supplementary Fig. 43) obviously demonstrate that more CD140a⁺ stem cells migrate into and adhere to the defect area. In addition, the 4-HNE expression in the Ru-Cu/EDHJ group decreases obviously compared to the ROSup group. We used immu-nofluorescence (IF) analysis with double staining for DNA/RNA damage and CD140a at the defect sites in our in vivo studies (Supplementary Fig. 44). The expression of DNA/RNA damage in the CD140a positive expression area decreased obviously in the Ru-Cu/EDHJ group compared to the ROSup group.

To investigate the in vivo bone formation performance of Ru-Cu/EDHJ, micro-CT analysis is performed[93]. The 3D reconstruction of

micro-CT images displays a larger amount of mineralized bone-like deposition in the Ru-Cu/EDHJ treated one compared to the ROSup group, which can achieve a complete mandible structure after 8 weeks (Fig. 8f). Nevertheless, evident defects are still present in the Ctrl group, the ROSup group and the $CeO_2$ group (Fig. 8f). The formation quantification of new bone was calculated by bone volume/tissue volume fraction (BV/TV), bone mineral density (BMD), trabecular thickness (Tb.Th), and trabecular number (Tb.N), as shown in Fig. 8g–j. The Ru-Cu/EDHJ group performs the highest BV/TV in comparison with other groups (Fig. 8g). A similar result is also discovered in the BMD analysis (Fig. 8h). The elevated BV/TV and complete 3D recon-struction images of the Ru-Cu/EDHJ group show a more compact structure of the newly formed bone. The strength of individual tra-beculae is also detected by studying the average value of Tb.Th and Tb.N. As depicted in Fig. 8i, the Ru-Cu/EDHJ group exhibits the highest Th.N and Th.Th (Fig. 8i, j). Hematoxylin and Eosin (H&E) staining is

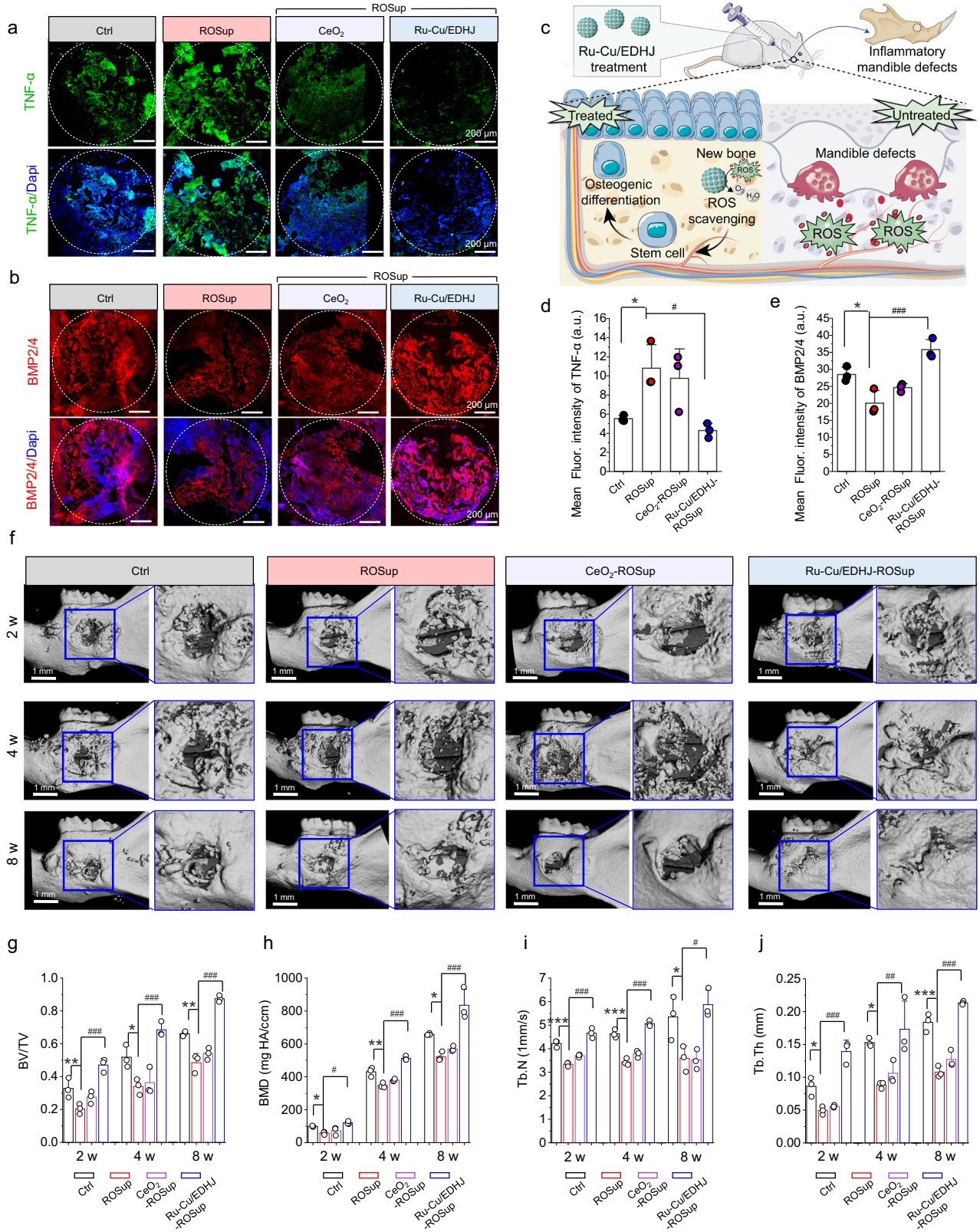

employed to evaluate the potential of the Ru-Cu/EDHJ in facilitating endogenous mandible regeneration[94]. As depicted in Supplementary Fig. 45, the ROSup and CeO₂ groups show evident inflammatory cell infiltration in the defect regions, even after 8 weeks, indicating prolonged inflammation and ongoing damage in the surgical site. The Ctrl group exhibits only a limited amount of newly formed bone matrix in the defect boundary. Conversely, the Ru-Cu/EDHJ group demonstrates

well-structured new bone tissues with minimal inflammatory cell presence. In addition, the expression of osteocalcin (OCN) significantly increases in the Ru-Cu/EDHJ group, highlighting its potential role in promoting osteogenesis[95,96].

We further investigated the degradation and biosafety of Ru-Cu/EDHJ after mandible repair. The inductively coupled plasma mass spectrometry (ICP-MS) analysis demonstrates that Ru-Cu/EDHJ

**Fig. 8 | In vivo mandibular regeneration enhanced by Ru-Cu/EDHJ.** Fluorescence staining images of (**a**) TNF-α and (**b**) BMP2/4 at week 1 after treatments. Scale bar: 200 μm. **c** Schematic illustration of Ru-Cu/EDHJ for ROS scavenging and promoting bone formation. Quantitative results of fluorescence intensity of (**d**) TNF-α ($n = 3$ independent replicates), $p_{(ROSup)} = 0.0499$, compared with Ctrl group; $p_{(Ru-Cu/EDHJ-ROSup)} = 0.0175$, compared with ROSup group. Quantitative results of fluorescence intensity of (**e**) BMP2/4 ($n = 3$ independent replicates), $p_{(ROSup)} = 0.0202$, compared with Ctrl group; $p_{(Ru-Cu/EDHJ-ROSup)} = 0.0004$, compared with ROSup group. **f** 3D reconstruction of micro-CT images. Scale bar: 1 mm. New bone formation quantitative result of (**g**) BV/TV ($n = 3$ independent replicates), $p_{(ROSup, 2w)} = 0.0013$, $p_{(ROSup, 4w)} = 0.0348$, $p_{(ROSup, 8w)} = 0.0013$, compared with Ctrl group; $p_{(Ru-Cu/EDHJ-ROSup, 2w)} < 0.0001$, $p_{(Ru-Cu/EDHJ-ROSup, 4w)} < 0.0001$, $p_{(Ru-Cu/EDHJ-ROSup, 8w)} < 0.0001$, compared with ROSup group. New bone formation quantitative result of (**h**) bone mineral density (BMD) ($n = 3$ independent replicates), $p_{(ROSup, 2w)} = 0.0295$, $p_{(ROSup, 4w)} = 0.0018$, $p_{(ROSup, 8w)} = 0.0461$, compared

with Ctrl group; $p_{(Ru-Cu/EDHJ-ROSup, 2w)} = 0.0112$, $p_{(Ru-Cu/EDHJ-ROSup, 4w)} < 0.0001$, $p_{(Ru-Cu/EDHJ-ROSup, 8w)} = 0.0003$, compared with ROSup group. New bone formation quantitative result of (**i**) Tb.N ($n = 3$ independent replicates), $p_{(ROSup, 2w)} = 0.0003$, $p_{(ROSup, 4w)} < 0.0001$, $p_{(ROSup, 8w)} = 0.0435$, compared with Ctrl group; $p_{(Ru-Cu/EDHJ-ROSup, 2w)} < 0.0001$, $p_{(Ru-Cu/EDHJ-ROSup, 4w)} < 0.0001$, $p_{(Ru-Cu/EDHJ-ROSup, 8w)} = 0.0115$, compared with ROSup group. New bone formation quantitative result of (**j**) Tb.Th ($n = 3$ independent replicates), $p_{(ROSup,2w)} = 0.0206$, $p_{(ROSup,4w)} = 0.0401$, $p_{(ROSup, 8w)} < 0.0001$, compared with Ctrl group; $p_{(Ru-Cu/EDHJ-ROSup, 2w)} < 0.0001$, $p_{(Ru-Cu/EDHJ-ROSup, 4w)} = 0.0096$, $p_{(Ru-Cu/EDHJ-ROSup, 8w)} < 0.0001$, compared with ROSup group. Ctrl (mandible defects with PBS treatment). Data are presented as means ± SD., *$p < 0.05$, **$p < 0.01$, ***$p < 0.001$, #$p < 0.05$, ##$p < 0.01$, ###$p < 0.001$, ns, no significance; statistical significance was calculated using one-way ANOVA followed by Tukey's post-hoc test for multiple comparisons, all tests were two-sided. Experiments were repeated independently (**a, b, f**) three times with similar results. Source data are provided as a Source Data file.

gradually degraded, especially the copper ions (Supplementary Fig. 46). TEM and ICP-MS results indicate a marked reduction of Ru-Cu/EDHJ within the regenerated new bone after 8 weeks, which is evidently different compared to the 1-week time point (Supplementary Fig. 47). Importantly, we found that the Ru-Cu/EDHJ composite could be excreted through urine and feces (Supplementary Fig. 47)[97]. Meanwhile, the Ru-Cu/EDHJ group presents good biocompatibility with no noticeable damage indications to the viscus (heart, liver, spleen, lung, and kidney, Supplementary Fig. 48) and body weight (Supplementary Fig. 49). These findings collectively indicate that Ru-Cu/EDHJ exhibits excellent biosafety.

## pH-controlled antibacterial action of Ru-Cu/EDHJ

Bacterial infection often leads to infectious jaw diseases, such as osteomyelitis[98], apical periodontitis[99], periodontitis[100], peri-implantitis[101], and mandibular fractures with infection[102] and so on, and antibacterial action can promote the healing process at the wound site[103]. Thus, it would be significantly valuable if the Ru-Cu/EDHJ could also exhibit bioadaptive antibacterial activity. POD-like activity of materials enables the local transformation of $H_2O_2$ to generate free radicals, which results in oxidative damage to bacteria by destabilizing and inactivating cytoplasmic proteins and the bacterial cell envelope[104]. We then assess the POD-mimetic activities of materials on converting $H_2O_2$ into ROS (Fig. 9a and Supplementary Fig. 50), the Ru-Cu/EDHJ exhibits the highest POD-mimetic activity in an acidic environment, which surpasses that of $CeO_2$, EDHJ, and Ru-O/EDHJ, highlighting its favorable ROS-generation performance. Further quenching experiments reveal that the ROS produced by Ru-Cu/EDHJ with $H_2O_2$ in an acidic environment primarily consists of $\cdot O_2^-$ and $^1O_2$ (Fig. 9b). Conversely, a neutral environment does not induce any POD-like activity, indicating that the Ru-Cu/EDHJ do not generate ROS or cause cellular damage under normal tissues (neutral conditions, Fig. 9c). Theoretical calculations are utilized to investigate the mechanisms of different catalytic activities triggered by acidic and neutral environments. As shown in the Supplementary Figs. 51, 52, the Ru active site initially adsorbs an $H_2O_2$ molecule, leading to the release of $H_2O$ and the subsequent formation of an *O intermediate. In a neutral environment, where the proton source is $H_2O_2$, *O combines with $H_2O_2$ to release $H_2O$ and $O_2$, thereby exhibiting CAT-like activity. In contrast, within an acidic environment, where the proton sources are $H^+$ and $H_2O_2$, the higher affinity of $H^+$ for *O (with a Gibbs free energy barrier of -1.16 eV) in comparison to $H_2O_2$ (with a Gibbs free energy barrier of -0.17 eV) leads to its preferential surface interaction, generating *OH species. Subsequently, *OH re-adsorbs with another $H_2O_2$ molecule to yield OOH and $H_2O$, showcasing POD-like activity.

Then, we incubated *E. coli* and *S. aureus* (Methicillin-resistant strain) in planktonic conditions with different materials and detected the antibacterial effects. As shown in Fig. 9d, e, the Ru-Cu/EDHJ + $H_2O_2$ group displays the most effective disruption of *E. coli*

and *S. aureus* via ROS damage. Notably, it is a demanding challenge to eradicate a defensive biofilm due to its lower sensibility to antimicrobial drugs. Therefore, we then test the biofilm eradication capability; it is found that the Ru-Cu/EDHJ + $H_2O_2$ treated biofilms display the highest dead bacteria ratios with nearly no live bacteria, while the other groups still have abundant live bacteria (Fig. 9f–i). The morphology of treated bacteria is also investigated by SEM (Fig. 9j); the bacterial membranes are disrupted severely in the Ru-Cu/EDHJ group, which are pointed out by red arrows point, while the other groups exhibit regular spherical shapes of *S. aureus* and the normal rod shape of *E. coli*. Moreover, the crystal violet staining is also used to analyze the treated biofilms. The corresponding OD570 values of the crystal violet solution confirm that Ru-Cu/EDHJ displays the highest elimination properties of *E. coli* and *S. aureus* biofilms (Fig. 9k, l). Thus, we can ensure that Ru-Cu/EDHJ owns excellent pH-controlled antibacterial properties with additional $H_2O_2$ injection (Fig. 9m).

To evaluate whether Ru-Cu/EDHJ exhibits efficient antibacterial capacity under pathological conditions in vivo, we initially examined its antibacterial performance in *S. aureus* and *E. coli*-infected bone defects (Supplementary Fig. 53). After 1 week of treatment, the number of *S. aureus* and *E. coli* colonies in the Ru-Cu/EDHJ group was significantly lower than that in the other groups. The quantification result shows that Ru-Cu/EDHJ exhibits the highest antibacterial efficiency. Furthermore, compared with the $H_2O_2$ treatment, fewer Ly6G⁺ neutrophils and more CD140a⁺ MSCs could be observed from the Ru-Cu/EDHJ group with *S. aureus*-infection via immunofluorescence staining, indicating that the high anti-infective efficacy of Ru-Cu/EDHJ can enhance the survival of endogenous MSCs (Supplementary Fig. 53).

## Discussion

In this work, we have designed an efficient, versatile, and precise biocatalytic heterojunction for treating inflammatory mandible defects. Remarkably, our experimental and theoretical studies have demonstrated that the synthesized Ru-Cu/EDHJ exhibits unique Ru-Cu coordination structures with enhanced electron density and optimized binding strength of oxygen species on Ru sites, which eventually endows the Ru-Cu/EDHJ to present excellent and versatile antioxidase-like activities and pH-controlled ROS catalytic properties. Accordingly, the antioxidase-like Ru-Cu/EDHJ can maintain stem cell survival, proliferation, and osteogenic differentiation in high ROS-level conditions by preventing oxidative stress-induced DNA damage and cell apoptosis, maintaining the metabolism and osteogenesis functions and reducing inflammatory responses. Moreover, the Ru-Cu/EDHJ also displays excellent pH-controlled antibacterial properties via POD-like catalytic ROS production in acidic environments, which allows the Ru-Cu/EDHJ to own bioadaptive anti-infection activity in treating bacteria-induced refractory mandibular diseases. Overall, the synthesized

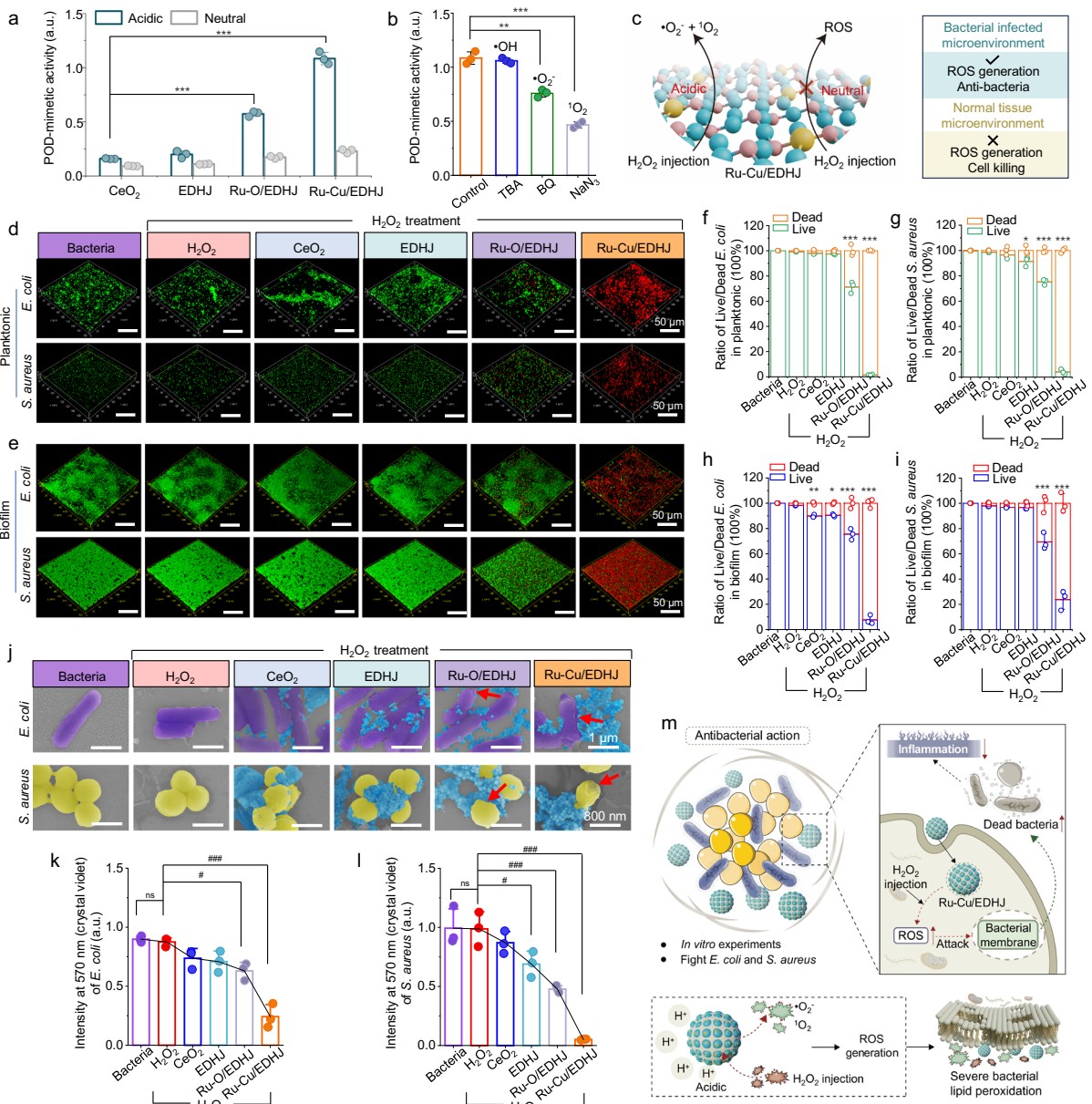

**Fig. 9 | pH-controlled bioadaptive antibacterial action of Ru-Cu/EDHJ. a** POD-like activity of different biocatalysts ($n = 3$ independent replicates), $p_{(Ru-O/EDHJ)}$ <0.0001, $p_{(Ru-Cu/EDHJ)}$ <0.0001, compared with $CeO_2$ group. **b** Tert-butanol (TBA) to quench •OH, benzoquinone (BQ) to quench •$O_2^-$, and $NaN_3$ to quench $^1O_2$ during the biocatalytic process ($n = 3$ independent replicates). $p_{(BQ)} = 0.0011$, $p_{(NaN3)}$ <0.0001, compared with Control group. **c** Schematic illustration of Ru-Cu/EDHJ converting $H_2O_2$ to ROS in acidic conditions but not in neutral environments. Fluorescence images from live/dead staining of (**d**) planktonic *E. coli* and *S. aureus*, and (**e**) *E. coli* and *S. aureus* biofilms. **f** The live/dead bacteria ratios of *E. coli* ($n = 3$ independent replicates, $p_{(Ru-O/EDHJ-H2O2)} < 0.0001$, $p_{(Ru-Cu/EDHJ-H2O2)} < 0.0001$, compared with bacteria in $H_2O_2$ group) and (**g**) *S. aureus* ($n = 3$ independent replicates, $p_{(EDHJ-H2O2)} = 0.0245$, $p_{(Ru-O/EDHJ-H2O2)} < 0.0001$, $p_{(Ru-Cu/EDHJ-H2O2)} < 0.0001$, compared with bacteria in $H_2O_2$ group). **h** The live/dead bacteria ratios from *E. coli* biofilms ($n = 3$ independent replicates, $p_{(CeO2-H2O2)} = 0.0093$, $p_{(EDHJ-H2O2)} = 0.0150$, $p_{(Ru-O/EDHJ-H2O2)} < 0.0001$, $p_{(Ru-Cu/EDHJ-H2O2)} < 0.0001$, compared with bacteria in

$H_2O_2$ group) and (**i**) *S. aureus* biofilms ($n = 3$ independent replicates, $p_{(Ru-O/EDHJ-H2O2)}$ < 0.0001, $p_{(Ru-Cu/EDHJ-H2O2)} < 0.0001$, compared with bacteria in $H_2O_2$ group). **j** SEM images of *E. coli* and *S. aureus* after treatments. Blue: materials, purple: *E. coli*, yellow: *S. aureus*. **k** Quantitative analysis of the crystal violet-stained *E. coli* biofilms ($n = 3$ independent replicates, $p_{(Ru-O/EDHJ-H2O2)} = 0.0147$, $p_{(Ru-Cu/EDHJ-H2O2)} < 0.0001$, compared with $H_2O_2$ group) and (**l**) *S. aureus* biofilms ($n = 3$ independent replicates, $p_{(EDHJ-H2O2)} = 0.0483$, $p_{(Ru-O/EDHJ-H2O2)} = 0.0009$, $p_{(Ru-Cu/EDHJ-H2O2)} < 0.0001$, compared with $H_2O_2$ group). **m** Schematic illustration of the pH-controlled antibacterial action of Ru-Cu/EDHJ in bone infections with added $H_2O_2$. Data are presented as means ± SD., *$p < 0.05$, **$p < 0.01$, ***$p < 0.001$, #$p < 0.05$, ##$p < 0.01$, ###$p < 0.001$, ns, no significance; statistical significance was calculated using one-way ANOVA followed by Tukey's post-hoc test for multiple comparisons, all tests were two-sided. Experiments were repeated independently (**d**, **e**, **j**) three times with similar results. Source data are provided as a Source Data file.

Ru-Cu/EDHJ simultaneously exhibits superior repair function and bioadaptive antibacterial properties in treating mandible defects, which offers a promising pathway to design ROS biocatalytic materials for treating trauma, tumor, or infection-caused maxillofacial bone defects and also provides a new therapeutic avenue to address the oxidative stress-related inflammatory diseases.

## Methods

### Reagents and materials

Copric chloride dihydrate ($CuCl_2·2H_2O$, 99.99%), cerium (III) chloride heptahydrate ($CeCl_3·7H_2O$, 99.99%), and hexamethylenetetramine (HMT, 99.0%) were purchased from Aladdin reagents (Shanghai, China). The ruthenium (III) chloride hydrate ($RuCl_3·xH_2O$) was

obtained from Energy Chemical (Anhui, China). All aqueous solutions were prepared with ultra-pure (UP) water. The rest of the reagents, if not mentioned specifically, were supplied by Aladdin reagents (Shanghai, China). All the reagents were of analytical grade and were used as received.

## Synthesis of EDHJ (CeO$_2$@Cu$_2$O) biocatalyst

In the typical procedure, 186.3 mg of CeCl$_3$·7H$_2$O and 85.3 mg of CuCl$_2$·H$_2$O were dissolved in 50 mL ultrapure water from solution A. Simultaneously, hexamethylenetetramine (HMT, 3.154 g) was dissolved in 50 mL ultrapure water from solution B. Solution A was then introduced into solution B under magnetic stirring for 5 min. The resulting mixture suspension was subjected to a temperature of 75 °C in an oil bath for 4 h. The resulting product was isolated through centrifugation and subsequently washed with deionized water and ethyl alcohol. The collected sample was then subjected to overnight drying under vacuum conditions in an oven maintained at 60 °C. This sample was designated as CeO$_2$@Cu$_2$Cl(OH)$_3$. To acquire the EDHJ biocatalyst, the prepared CeO$_2$@Cu$_2$Cl(OH)$_3$ was introduced into 50 mL of ultrapure water and subjected to a Teflon-lined autoclave maintained at a temperature of 120 °C for 4 h. The resulting solid gray-green precipitate was collected through centrifugation, followed by three subsequent washes with water and ethanol. Subsequently, the collected sample was dried overnight in a vacuum oven at 60 °C.

## Synthesis of Ru-Cu/EDHJ biocatalyst

In the standard procedure, the precursor CeO$_2$@Cu$_2$Cl(OH)$_3$ (50 mg) was dissolved in 50 mL of ultrapure water. Following this, RuCl$_3$·xH$_2$O (5 mg) was added to the solution and stirred at room temperature for 5 min. Subsequently, the resulting solution was introduced into a Teflon-lined autoclave maintained at 120 °C for 4 h. The resulting solid gray precipitate was collected through centrifugation and subsequently subjected to three washes with water and ethanol. The collected sample was then dried under vacuum in an oven set at 60 °C overnight. This sample was designated as Ru-Cu/EDHJ.

## Synthesis of CeO$_2$@CuO biocatalyst

To obtain the CeO$_2$@CuO biocatalyst, the previously prepared CeO$_2$@Cu$_2$Cl(OH)$_3$ was combined with 50 mL ultrapure water and maintained in a Teflon-lined autoclave heated to 180 °C for 10 h. The resulting solid gray-green precipitate was collected through centrifugation and subsequently washed three times with water and ethanol. Finally, the precipitant was dried overnight in a vacuum oven at 60 °C.

## Synthesis of Ru-O/EDHJ biocatalyst

In the standard procedure, the CeO$_2$@Cu$_2$Cl(OH)$_3$ precursor (50 mg) was dissolved in 50 mL ultrapure water. Subsequently, RuCl$_3$·xH$_2$O (5 mg) was added to the solution and stirred at room temperature for 5 min. The resulting solution was then transferred to a Teflon-lined autoclave heated to 180 °C and maintained there for 10 h. The solid gray precipitate was collected by centrifugation and subsequently washed three times with water and ethanol. The collected sample was dried overnight under a vacuum in an oven set at 60 °C. This sample was designated as Ru-O/EDHJ.

## Synthesis of CeO$_2$ biocatalyst

Initially, CeCl$_2$·6H$_2$O (372 mg) was dissolved in 50 mL ultrapure water from solution A. HMT (3.154 g) was dissolved in 50 mL ultrapure water from solution B. Subsequently, solution A was added to solution B while stirring for 5 min at room temperature. The resulting solution was then heated to 75 °C in an oil bath and maintained at that temperature for 4 h. Next, the solution was transferred to a Teflon-lined autoclave heated to 120 °C and kept there for 4 h. The resulting

suspension was centrifuged at 9000 rpm for 5 min, and the obtained CeO$_2$ biocatalyst was washed with water and ethanol, followed by drying in a vacuum oven at 60 °C overnight.

## Synthesis of CeO$_2$-Ru$_{SA}$ biocatalyst

Initially, RuCl$_3$·xH$_2$O (5 mg) was dissolved in 50 mL ultrapure water. The above CeO$_2$ (50 mg) was then added to the solution and stirred at room temperature for 5 min. Subsequently, the resulting solution was transferred to a Teflon-lined autoclave heated to 120 °C and kept there for 4 h. The solid gray precipitate was collected by centrifugation and subsequently washed three times with water and ethanol. The collected sample was dried overnight under a vacuum in an oven at 60 °C, and it was designated as CeO$_2$-Ru$_{SA}$.

## Characterizations

Field emission scanning electron microscopy (FE-SEM) was performed with the Hitachi Regulus8220, Japan. Transmission electron microscopy (TEM) was performed on materials using a FEI Talos F200X operated at 200 kV. For bone defects, TEM was conducted using a Talos F200S microscope. Aberration-corrected high-angle annular dark-field scanning TEM (AC HAADF-STEM) and energy dispersive spectroscopy (EDS) mapping were performed via FEI Titan Cubed Themis G2 300 and Titan Themis 60-300 operated at 200 kV. X–ray diffraction (XRD, DX-2700BH, HaoYuan Instrument, China) was used to analyze crystal structures of catalysts with Cu Kα radiation for a 2θ range of 5–80°. X-ray photoelectron spectroscopy (XPS) spectra were measured on the K-Alpha™ + X-ray Photoelectron Spectrometer System (Thermo Scientific) with a Hemispheric 180° dual-focus analyzer with a 128-channel detector. The X-ray absorption (XAS) spectra and extended X-ray absorption fine structure (EXAFS) were carried out on the sample at 21 A X-ray nanodiffraction beamline of Taiwan Photon Source, National Synchrotron Radiation Research Center (NSRRC). This beamline adopted a 4-bounce channel-cut Si (111) monochromator for mono-beam X-ray nanodiffraction and X-ray absorption spectroscopy. The end station is equipped with three ionization chambers and a Lytle/SDD detector after the focusing position of the KB mirror for transmission and fluorescence mode X-ray absorption spectroscopy. The photon flux on the sample ranges from $1 \times 10^{11}$ to $3 \times 10^9$ photon/s for X-ray energy from 6 to 27 keV. The absorbance was detected by a multifunctional enzyme labeling instrument (ReadMax1900). Fluorescence images were collected via a confocal laser scanning microscope (FV3000 and Nikon, Olympus, Japan). Live/dead staining was counted using a Celigo Image Cytometer (Nexcelom Bioscience LCC., USA). Flow-cytometry data were collected via a flow cytometer (Attune™ NxT, Invitrogen, USA). RNA concentration and purity were measured using NanoDrop 2000 (Thermo Fisher Scientific, Wilmington, DE, USA). RNA integrity was assessed using the RNA Nano 6000 Assay Kit of the Agilent Bioanalyzer 2100 system (Agilent Technologies, CA, USA). Alkaline Phosphatase and Alizarin Red staining images were taken with a stereomicroscope (SZX16, Olympus, Japan). H&E staining and reactive oxygen species determination in vivo were scanned by a full slide scanner (VS200, Olympus, Japan). Micro-CT images were collected using μ-CT Scanner (μ-CT50, Scanco, Bassersdorf, Zurich, Switzerland). The libraries were sequenced on an Illumina NovaSeq 6000 platform. Data were analyzed using GraphPad Prism 8.0, Origin 2022, VASP 5.4.1, MDI Jade 6, Adobe Illustrator 27.0.1, Avantage 5.967, Athena software 0.9.26, Artemis software 0.9.26, and Digital Micrograph 3.7.4. The software Image J 1.8.0 and Image-Pro Plus 6.0 were used for in vitro and in vivo imaging analysis. Flow-cytometry analysis was performed in FlowJo v10.8.1 and in FlowJo v10.8.1 or in the software of Attune™ NxT. Bioinformatic analysis was performed using the free online platform BMKCloud (www.biocloud.net).

## Antioxidase-like activities assays

To assess SOD-like activity, 1 mg $KO_2$ was dissolved into 1 mL dimethyl sulfoxide solution (DMSO, containing 3 mg mL$^{-1}$ 18-crown-6-ether). And then, the biocatalyst was dispersed into the above $KO_2$/DMSO solution with a final concentration of 50 μg mL$^{-1}$. After reaction for 5 min, the remaining •$O_2^-$ will be detected by 10 μL nitro blue tetrazolium (NBT)-DMSO solution (10 mg mL$^{-1}$ NBT). The absorption intensity of the reaction solution at $\lambda_{max}$ = 680 nm was immediately measured and then compared with the original concentration of •$O_2^-$ to estimate the •$O_2^-$ scavenging ability.

The CAT-like activity was performed by evaluating $H_2O_2$ scavenging and $O_2$ generation. The $H_2O_2$ scavenging capacity was evaluated as follows: a total of 10 mM of $H_2O_2$ and 50 μg mL$^{-1}$ of biocatalyst were mixed in PBS (pH = 7.4) to 2 mL. Then, 50 μL of the solution was added to 100 μL of Ti(SO$_4$)$_2$ solution (13.9 mM) at 10, 20, 30, and 40 min. The absorbance of the solution at 405 nm was measured to evaluate the remaining $H_2O_2$ concentration. The $O_2$ generation capability was determined as follows: 200 μL $H_2O_2$ (10 M) was added into 20 mL PBS (pH = 7.4) containing 20 μL Ru-Cu/EDHJ (10 mg mL$^{-1}$), the $O_2$ concentration was detected by a Dissolved Oxygen Meter for 3 min with an interval of 5 s.

The steady-state kinetic assays of catalase-like activities were evaluated by changing the concentration of $H_2O_2$. Briefly, all the assays were conducted in 50 mL centrifuge tube containing 20 mL PBS (pH = 7.4) and 20 μL biocatalysts (10 mg mL$^{-1}$). In a typical essay, different concentrations of $H_2O_2$ (5–400 mM, respectively) were mixed with 20 mL PBS containing biocatalysts to monitor the solubility change of $O_2$ for 3 min using a Dissolved Oxygen Meter. Next, for each $H_2O_2$ concentration, the "Absorbance versus Time" curve was obtained, which can be used to calculate the initial reaction velocity (V). The reaction rates were then plotted against their corresponding $H_2O_2$ concentration (S) and then fitted with the Michaelis–Menten curves (Eq. (1)). Furthermore, the slope and intercept of the linear double-reciprocal plot (Lineweaver–Burk plot, Eq. (2)) were used to determine the maximal reaction velocity ($V_{max}$) and Michaelis–Menten constant ($K_m$). Furthermore, the catalytic efficiency in terms of turnover number (TON) was calculated according to Eq. (3), where [E$_0$] represents the concentration of the catalytic center of catalysts.

$$V = (V_{max} \times [S])/(K_m + [S]) \tag{1}$$

$$1/V = K_m/(V_{max} \times [S]) + (1/V_{max}) \tag{2}$$

$$TON = V_{max}/[E_0] \tag{3}$$

The GPx-like activity was evaluated as follows: 100 μL of glutathione reductase (GR) solution (17 U mL$^{-1}$), 61.5 μL of glutathione (GSH) solution (10 mg mL$^{-1}$), 33.3 μL of nicotinamide adenine dinucleotide phosphate (NADPH) solution (10 mg mL$^{-1}$), 5 μL of catalysts (10 mg mL$^{-1}$), and 24 μL of $H_2O_2$ (0.01 M) were added into 780 μL of PBS (pH = 7.4). Then, 50 μL of the mixed solution was added to a 96-well plate to test the GPx-like catalytic rate at 340 nm.

A total of 50 μg mL$^{-1}$ 2,2-diphenyl-1-picrylhydrazyl (DPPH•, Aladdin, Shanghai, China) ethanol solution and biocatalysts (50 μg mL$^{-1}$) were mixed to form a 2 mL solution. Then, the mixture was placed in the dark for reaction, and the absorbance of the solution at 519 nm was measured after 60 min. The DPPH• radical scavenging abilities were calculated.

## POD-like activity measured by TMB

Peroxidase activities were determined by colorimetric assays. 5 μL of catalysts (10 mg mL$^{-1}$), 25 μL of 3,3',5,5'-Tetramethylbenzidine (TMB, 10 mg mL$^{-1}$), and 25 μL of $H_2O_2$ (0.1 M) were added into a 1945 μL

sodium acetate-acetic acid (NaOAc/HOAc) buffer [100 mM (pH 4.5 or 7.4)]. The catalytic oxidation of TMB (oxTMB) was studied by measuring the absorption changes of the oxidized form of TMB at $\lambda_{max}$ = 652 nm. Unless otherwise stated, POD activities were carried out in an air-saturated buffer.

## Analysis of free radicals by quenching experiments

To probe •OH, 5 μL of catalysts (10 mg mL$^{-1}$), 25 μL of $H_2O_2$ (0.1 M), 200 μL of tert-butanol (TBA), and 25 μL of TMB (10 mg mL$^{-1}$) were added into a 1745 μL of NaOAc/HOAc buffer [100 mM (pH 4.5)]. The absorbance at a wavelength of 652 nm was recorded following a 10-min reaction period.

To probe •$O_2^-$, 5 μL of catalysts (10 mg mL$^{-1}$), 25 μL of $H_2O_2$ (0.1 M), 100 μL of p-benzoquinone (BQ), and 25 μL of TMB (10 mg mL$^{-1}$) were added into an 1845 μL of NaOAc/HOAc buffer [100 mM (pH 4.5)]. The absorbance at a wavelength of 652 nm was recorded following a 10-min reaction period.

To probe $^1O_2$, 5 μL of catalysts (10 mg mL$^{-1}$), 25 μL of $H_2O_2$ (0.1 M), 100 μL of sodium azide (NaN$_3$), and 25 μL of TMB (10 mg mL$^{-1}$) were added into an 1845 μL of NaOAc/HOAc buffer [100 mM (pH 4.5)]. The absorbance at a wavelength of 652 nm was recorded following a 10-min reaction period.

## Theoretical calculations

All theoretical calculations were performed using the DFT method, as implemented in the Vienna ab initio simulation package (VASP)[105–107]. The core electrons were described using the spin-polarized projector augmented wave (PAW) method[108], and the electron exchange and correlation energy were treated within the generalized gradient approximation in the Perdew-Burke-Ernzerhof functional (GGA-PBE)[109]. The valence states of all atoms were expanded in a plane-wave basis set with a cutoff energy of 450 eV. The convergence criteria for the electronic self-consistent iteration and force were set to $10^{-5}$ eV and 0.02 eV/Å with a Gamma centered $2 \times 2 \times 1$ K-points. Denser $5 \times 5 \times 1$ K-points were used for the density of states (DOS) computations. The CeO$_2$ (111) ($2 \times 2$), CuO (111) ($5 \times 2$), and Cu$_2$O (111) ($2 \times 2$) slab was modeled. During the simulation, the bottom layers of the substrates were fixed while the top layers were kept fully relaxed. The slab model was constructed with a vacuum layer of 15 Å in the z direction to avoid the interaction between neighboring images. The charge density differences were evaluated using the formula (4).

$$\Delta\rho = \rho(A + B) - \rho A - \rho B \tag{4}$$

where $\Delta\rho X$ is the electron density of X. Atomic charges were computed using the atom-in-molecule (AIM) scheme proposed by Bader. The isosurface value used for differential charge density is 0.006 e·Bohr$^{-3}$.

To explore the catalytic effect, the change of Gibbs free energy (ΔG) was calculated, as shown in formula (5):

$$\Delta G = \Delta E + \Delta ZPE + \Delta H_0 \rightarrow_{298K} -T\Delta S \tag{5}$$

where ΔE is the energy change obtained from DFT calculations; ΔZPE, ΔH, and ΔS denote the difference in zero-point energy, enthalpy, and entropy due to the reaction, respectively. The enthalpy and entropy of the ideal gas molecules were taken from the standard thermodynamic tables, and some of the calculation results were analyzed by the VASPKIT package.

## Cell culture

Human bone marrow-derived mesenchymal stem cells (hMSCs) were purchased from Cyagen Biosciences (HUXMA-01001, Cyagen, China) and then were cultured in the human mesenchymal stem cell growth medium (HUXMX-90021, Cyagen, China) at 37 °C under 5% $CO_2$,

according to the manufacturer's instructions. mMSCs were isolated from the bone marrow of the femurs and tibias of C57BL/6 mice. Briefly, the femurs and tibias were dissected under sterile conditions, the epiphyses were removed, and bone marrow was flushed out with α-MEM (SH30265.01; Hyclone, Logan, USA) + 1% penicillin-streptomycin. The collected cells were cultured in α-MEM supplemented with 10% fetal calf serum and 1% penicillin-streptomycin at 37 °C in a 5% $CO_2$ incubator. After 24 h of culture, the culture medium was replaced with fresh complete α-MEM medium to remove the nonadherent cells. The adherent cells were cultured for another 3–5 days and passaged until reaching 90% confluence.

### Intracellular ROS detection

Cells among the fourth to sixth passage were seeded onto 96-well plates at the density of 5000 cells/well, and treatment was performed after culturing overnight. DCFH-DA (S0033S, Beyotime, China) was used to detect the residual ROS level, according to manufacturers' instructions. Briefly, probes were preloaded by incubating cells with DCFH-DA working solution for 30 min. After washing with PBS three times, 100 μg/mL of artificial biocatalysts and 100 μM hydrogen peroxide were added successively. After 1 h, the medium was changed to a serum-free medium, and cells were cultured until the testing time point. Intracellular ROS level was determined through fluorescence intensity analysis with flow cytometry (AttuneTM NxT, Invitrogen, USA) and through a confocal laser scanning microscope (FV3000 and Nikon, A1R MP+, Olympus, Japan).

### Immunofluorescence staining

After being treated with different samples for 1 h, the cells are cultured for another 24 h. Cells were washed with PBS twice and fixed in 4% paraformaldehyde for 10 min at room temperature, then permeabilization in PBS containing 0.1% Triton X-100 for 10 min and blocking in PBS with 1% bovine serum albumin (BSA) for 1 h. Primary antibodies were diluted in PBS with 1% BSA and incubated overnight at 4 °C. Then, the cells were incubated by secondary antibodies with the required fluorophore in the dark at room temperature for 2 h. Before imaging, cells were counterstained with 10 μg mL$^{-1}$ 4',6-diamidino-2-phenylindole (Dapi) and mounted in the antifading mounting medium (S2110, Solarbio, China). Samples were washed with PBS three times between each step. A confocal laser scanning microscope (FV3000 and Nikon, A1R MP+, Olympus, Japan) was used to capture images.

Mandible samples from C57BL/6 male mice were fixed with 4% buffered paraformaldehyde overnight. Then they were decalcified in 10% EDTA solution for 4 weeks. After dehydration, samples were embedded in paraffin and sectioned. The paraffin sections were dewaxed in xylene and then rehydrated with gradient alcohol. After the slides were immersed in antigen retrieval solution for 30 min at 95 °C, we followed the process of immunofluorescent staining above.

Primary antibodies and corresponding concentrations used in this study were 4-HNE (MA5-27570, Life Technologies, USA,1:100 dilution), paxillin (ab32084, Abcam, USA, 1:200 dilution), γH2A.X (phospho S139) (ab81299, Abcam, USA, 1:200 dilution), DNA/RNA Damage (ab62623, Abcam, USA, 1:200 dilution), and COL1A1 (R26615, ZEN-BIOSCIENCE, China, 1:50 dilution), TNFα (346654, ZEN-BIOSCIENCE, China, 1:50 dilution), BMP2/4 (sc-137087, Santa Cruz Biotechnology, USA, 1:50 dilution), CD140a (ab203491, Abcam, USA, 1:200 dilution), iNOS (GB11119, Servicebio, China, 1:50), CD206 (sc-58986, Santa Cruz Biotechnology, USA, 1:50 dilution), Ly6g (GB11229, Servicebio, China, 1:50). The cytoskeleton was stained by FITC-conjugated phalloidin (A12379, Invitrogen, USA, 1:200 dilution). The secondary antibodies were Alexa Fluor 647 goat anti-mouse IgG (ab150115, Abcam, USA, 1:200 dilution), Alexa Fluor 647 donkey anti-rabbit IgG (ab150075, Abcam, USA, 1:200 dilution) and Alexa Fluor 488 goat anti-rabbit IgG (ab150077, Abcam, USA, 1:200 dilution), Cy3 goat anti-rabbit IgG

(GB21303, Servicebio, China, 1:50), Cy3 goat anti-mouse IgG (GB21301, Servicebio, China, 1:50).

### Cellular live/dead staining

For detecting hMSC protection by Ru-Cu/EDHJ in a high-level $H_2O_2$ microenvironment test, cells were seeded onto 96-well plates at the same density in different groups, and then 100 μg/mL of artificial biocatalysts and 100 μM hydrogen peroxide were added successively. After 1 h, the medium was changed to a standard culture medium and cultured overnight. For the cytotoxicity test, cells were seeded onto 96-well plates at the same density in groups, and the treatment with 100 μg/mL of artificial biocatalysts was performed for 48 h. Before staining, the plate was centrifuged to make cells settle on the bottom. In addition, the supernatant was sipped gently to avoid sucking up cells. Then, Calcein AM/Propidium Iodide (PI) staining was performed according to the instructions (C2012, Beyotime, China). Briefly, 2 μM Calcein AM solutions in PBS were used to stain live cells, and then 4.5 μM PI solutions were used to stain the dead. The cells were captured by a confocal laser scanning microscope and counted by the Celigo Image Cytometer (Nexcelom Bioscience LCC., America).

### Cell counting kit-8 assay (CCK-8 assay)

hMSCs ($5 \times 10^3$ cells per well) were plated into 96-well plates and incubated in a complete medium containing varying concentrations of Ru-Cu/EDHJ. After a 48-h incubation, the cell viability was assessed using the CCK-8 reagent (K1018, Apexbio Technology, USA), following the protocol of the manufacturer.

### Transcriptome sequencing and data analysis

hMSCs ($5 \times 10^6$ cells/mL) were cultured in a 10 mm plate and were treated with different samples for 1 h. Then, the cells were cultured overnight in a standard culture medium. After being treated with different samples, the hMSCs were lysed by TRIzolTM reagent (15596026, Invitrogen, CA, USA), and cell lysates were stored at −80 °C before sequencing. RNA concentration and purity were measured using NanoDrop 2000 (Thermo Fisher Scientific, Wilmington, DE). RNA integrity was assessed using the RNA Nano 6000 Assay Kit of the Agilent Bioanalyzer 2100 system (Agilent Technologies, CA, USA). The libraries were sequenced on an Illumina NovaSeq platform to generate 150 bp paired-end reads, according to the manufacturer's instructions. Genes with an adjusted Fold Change ≥ 1.5 & FDR < 0 .05 found by DESeq2_edgeR were assigned as differentially expressed. The Heatmap, PCA, GO term enrichment, Kyoto encyclopedia of genes and genomes (KEGG) pathway enrichment, and GSEA pathway enrichment analyses were performed using the free online platform BMKCloud (www.biocloud.net).

### Cell apoptosis

hMSCs were seeded in 6 mm plates and allowed to adhere. After being treated with different samples for 1 h, cells were cultured in a fresh medium for another 24 h[110]. Then, the cells in the supernatant and the digestive cells were centrifuged together to avoid the loss of cells. Then, the Annexin V-FITC/PI Apoptosis Detection Kit (AD10, Dojindo, Japan) was used to stain the cells before the flow cytometry analysis. According to the fluorescence intensity, cell populations were assigned into four quadrants, including live, early apoptotic, late apoptotic, and necrotic cells. The figures were formed using Flowjo software (version 10.8.1). Moreover, the tunnel staining followed the manufacturer's instructions for the One Step TUNEL Apoptosis Assay Kit (C1089, Beyotime Biotechnology, China), and the images were captured by a confocal laser scanning microscope.

### Osteogenic differentiation

After being treated with different samples for 1 h, the medium was changed to a fresh culture medium, and hMSCs were cultured for

another 24 h. Then, an induction medium was applied instead of a normal culture medium. The osteogenic conditional medium containing 50 µg mL$^{-1}$ L-ascorbic acid (Sigma-Aldrich, USA), 10 nM dexamethasone (Sigma-Aldrich, USA), and 10 mM β-Glycerophosphate (Sigma-Aldrich, USA).

## Alkaline phosphatase (ALP) and alizarin red (AR) stainings

Cells were induced in an osteogenesis medium for 3 days, and ALP staining (C3206, Beyotime, China) was performed. On day 21, AR staining (G1452, Solarbio, China) was performed. Images were taken with a stereomicroscope (SZX16, Olympus, Japan).

## In vivo study

The animal experiments and procedures, including euthanasia, were performed using protocols approved by the Institutional Animal Care and Use Committee at Sichuan University (Number: WCHSIRB-D-2020-361). The study was reviewed and approved by the Laboratory Animal Welfare and Ethics Committee of West China Hospital of Stomatology. All experiments involving animal use were performed in accordance with the ARRIVE guidelines. Six-week-old male C57BL/6 male mice (15–25 g) were used in this study. All mice were maintained under a 12 h light-dark cycle (light on from 8:00 a.m. to 8:00 p.m.) with ad libitum access to food and water. All diets were prepared by Jiangsu-Xietong, Inc. (Nanjing, China), catalog number: 1010038. The ambient temperature is 20–26 °C and the humidity is 40–70%. A mouse mandible fenestration defect model was created. All possible efforts were made to minimize the pain and discomfort of the mice during the surgery. The mice were anesthetized with 4% (w/v) isoflurane, followed by an intraperitoneal injection of ketamine (60 mg/kg) and xylazine (12 mg/kg), combined with a subcutaneous injection of buprenorphine for analgesia. The furs surrounding the surgical area were removed, and the skin was disinfected. The area of interest was dissected by an incision until the body of the mandible was reached (buccal plate). After locating the buccal plate, a BR-49 round bur (around 0.8 mm in diameter) was used to initiate access to the buccal bone, followed by the immerging of the end of the bur to create a standard defect. Subsequently, we constructed an inflammatory jaw defect model with C57BL/6 male mice by treating LPS (1 mg/mL, 0.02 mL) dissolved in PBS. Subsequently, the gelatin sponge mixed with PBS or with 100 µg/mL of artificial biocatalysts was injected to cure the defect and promote wound healing.

To evaluate the therapeutic effect of artificial biocatalysts, mice were divided into 4 groups: group 1 was locally treated with PBS; group 2 was locally treated with LPS; group 3 was locally treated with LPS+ CeO$_2$ (100 µg/mL) dissolved in PBS; group 4 was locally treated with LPS+ Ru-Cu/EDHJ (100 µg/mL) dissolved in PBS. The mice were humanely euthanized on 2, 4, and 8 weeks after surgery. For murine sample collections, euthanasia was performed through intraperitoneal injection of 0.5 mg xylazine (16–25 mg/kg for mice weighted 20–30 g) and 1.25 mg Zoletil® (40–60 mg/kg for mice weighted 20–30 g; Virbac, Westlake, TX, USA). Male mice were used for all experiments with data collected from ≥3 mice per experimental condition for all experiments. The samples were harvested for µ-CT assay and histology stainings. The body weight of the mice was monitored every 24 h for 2 weeks.

## Inductively coupled plasma mass spectrometry (ICP-MS) detection

Collected tissues were quantified, mixed with aqua regia, and incubated at 80 °C until completely dissolved. The solution was centrifuged at 12,000 × g for 20 min, and the supernatant was collected. The supernatant was analyzed using an inductively coupled plasma-mass spectrometer (Agilent 7850 spectrometer). To determine the metabolic pathways of nanoparticles, primary metabolites, such as urine and feces, were collected from mice in each group after 4 weeks of administration. To detect the degradation metabolism of nanomaterials in mandible tissue, samples were collected after 1 and 8 weeks of administration. The amounts of nanoparticle elements (Ru, Cu, Ce) were quantified using ICP-MS.

## ROS determination in vivo

Freshly made cryosections of the unfixed mandible were incubated in 10 µM Dihydroethidium (S0063, Beyotime, China) or DCFH-DA for 20 min in the dark. The samples were scanned by a full slide scanner (VS200, Olympus, Japan).

## Micro-computed tomography analysis

Mandibles were fixed in 10% formalin. For µ-CT analysis, the mandibles were scanned using µ-CT Scanner (µ-CT50, Scanco, Bassersdorf, Zurich, Switzerland), operated at 60 kV, 165 µA, 450 ms exposure time, and 10-µm resolution. We used standardized nomenclature for the bone parameters measured according to the µ-CT Scanner protocol, including the microstructural properties of BV/TV, Tb.Th, Tb.N, and bone mineral density (BMD).

## Histological staining assessment

The tissue samples were decalcified, embedded in paraffin, and sectioned for histological analysis using H&E staining. For immunohistochemistry staining, the paraffin sections were dewaxed in xylene and then rehydrated with gradient alcohol. After the sections were immersed in antigen retrieval solution for 30 min at 95 °C, they were treated with 3% hydrogen peroxide for 15 min. After that, the sections were permeabilized with 0.1% Triton X-100 in PBS for 5 min at room temperature and then blocked with 5% goat serum at room temperature for 1 h. We incubated primary antibodies that recognized OCN (GB11233, Servicebio, China, 1:200) overnight at 4 °C. Subsequently, the sections were washed in PBS and incubated with HRP conjugated Goat Anti-Rabbit IgG (GB23303, Servicebio, China, 1:200) for 1 h at room temperature. DAB was used as a chromogen, and hematoxylin was used as a counterstain. The images were observed and collected by (BX53F, Olympus, Japan).

## Antibacterial experiments and live/dead staining

*E. coli* (ATCC 25922, Gram-negative) and *S. aureus* (ATCC 6538, Gram-positive) were used in this research. Regarding *E. coli* and *S. aureus* planktonic culturing, the artificial biocatalysts were first dispersed in a Luria-Bertani medium. Then, the artificial biocatalysts with a final concentration of 100 µg/mL and H$_2$O$_2$ with a final concentration of 100 µM were introduced into 1 mL of ~10$^8$ CFU/mL bacterial suspensions in acidic conditions (PBS: pH 5.6). Similar to earlier reports[111–113], the suspensions were incubated at 37 °C for 2 h and then stained by LIVE/DEAD® Bac Light Bacterial Viability Kits (L7012, Invitrogen, USA). The 3D Z-stack fluorescent and orthogonal-stack images were acquired on the SP8 lighting confocal laser scanning microscope (Leica, Germany).

Regarding *E. coli* and *S. aureus* biofilm culturing, the suspension (50 µL, 10$^8$ CFU/mL) and Luria-Bertani medium (2 mL) were placed in 24-well plates, and then they were cultured at 37 °C for 48 h. After the medium was removed, the unattached bacteria were gently washed away with PBS three times, and the resulting biofilm on 24-well plates was harvested. Similar to earlier reports[111,114], bacteria were incubated separately with six different groups in acidic conditions (PBS: pH 5.6): PBS + H$_2$O$_2$, CeO$_2$ + H$_2$O$_2$, EDHJ + H$_2$O$_2$, Ru-O/EDHJ + H$_2$O$_2$ and Ru-Cu/EDHJ + H$_2$O$_2$ (Concentration: 100 µg/mL of artificial biocatalysts, 100 µM H$_2$O$_2$) at 37 °C for 2 h, and then the biofilms were gently washed with PBS three times and then stained by LIVE/DEAD® Bac Light Bacterial Viability Kits (L7012, Invitrogen, USA). The 3D Z-stack fluorescent and orthogonal-stack images were acquired on the SP8 lighting confocal laser scanning microscope (Leica, Germany).

## Crystal violet staining of bacterial biofilms

The treated biofilms from each group were washed three times with PBS and then fixed with 2.5% glutaraldehyde for 20 min. Then the liquid in each well was discarded. Crystal violet test solution (0.5%) was transferred into each well for 30 min, and excess dye was removed with PBS. Furthermore, the coverslips were allowed to dry. Next, absolute ethyl alcohol was used to dissolve the crystal violet bound to the biofilm. Finally, the value of OD 570 nm of the dissolved solution was examined by a microplate reader (SAF-6801, Bajiu Corporation, Shanghai, China).

## Observation of bacterial morphology

After being treated with different samples, the bacterial suspensions were fixed with 2.5% glutaraldehyde-containing PBS solution for 12 h at 4 °C and dehydrated with a gradient of ethanol/water solution. Then, the SEM images were obtained to observe the morphology of bacteria.

## Antibacterial activity in vivo

A mouse mandible fenestration defect model was established as described above. Subsequently, we added bacterial infection by introducing 1 μL of *S. aureus* or *E. coli* suspension ($1 \times 10^8$ CFU mL$^{-1}$) and incubating for 1 day. After that, materials, $H_2O_2$, and diluted PBS were added to treat the infected bone defect. The final concentrations of materials and $H_2O_2$ are 100 μg mL$^{-1}$ and 100 μM, respectively. The antibacterial experiments are performed in acidic conditions (pH 5.6). PBS and $H_2O_2$ (100 μM) were also used as a contrast. After 1 week of operation, to observe the residual bacteria in the infected bone defect after disinfection treatment, sterile cotton sticks were used to dip the tissue fluid from different bone defects. Then, the cotton sticks were put into 1 mL PBS, and 100 μL PBS was added to the agar plate. After 12 h incubation at 37 °C, the number of colonies on each plate was recorded. The mandible samples were fixed with 10% formaldehyde solution for immunofluorescent staining.

## Statistical analysis

Image J (version 1.8.0) and Image-Pro Plus (version 6.0) were used for in vitro and in vivo imaging analysis. Data were analyzed using Graph-Pad Prism 8.0 software (GraphPad Prism, San Diego, California, USA). Figures were formed using Origin 2022. Sample size (n), probability (P) value, data normalization, and specific statistical tests for each experiment were clarified in the figure legends. All of the data were presented as the mean ± SD. from a minimum of three independent experiments. The comparison of the mean values between multiple groups was performed by one-way analysis of variance (ANOVA) or two-tailed Student's *t*-tests. All tests were two-sided. A value of $p < 0.05$ was considered significant (represented as *$p < 0.05$, **$p < 0.01$, ***$p < 0.001$, #$p < 0.05$, ##$p < 0.01$, ###$p < 0.001$ or not significant (ns)).

## Reporting summary

Further information on research design is available in the Nature Portfolio Reporting Summary linked to this article.

## Data availability

The main data supporting the results of this study are available within the paper and its Supplementary Information. Any other raw data or noncommercial material used in this study are available from the corresponding author. Raw RNA sequencing data generated in this study have been deposited in the NCBI SRA database under accession number GSE253527. Source data are provided with this paper.

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

## Acknowledgements

This work was financially supported by the National Key R&D Program of China (2023YFC3605600 [L.Y.] and 2021YFB3800700 [C.C.]), National Natural Science Foundations of China (82470962 [M.R.B.], 82001020 [M.R.B.], U21A20368 [L.Y.], 52161145402 [C.C.], 52173133 [C.C.], 52373148 [C.C.]), Sichuan Science and Technology Program (2024NSFSC0672 [M.R.B.], 2023YFS0019 [L.Y.], 2023YFH0008 [C.C.], 2021YFG0238 [M.R.B.]), the 1·3·5 Project for Disciplines of Excellence, West China Hospital, Sichuan University (ZYJC21047 [C.C.]), the State Key Laboratory of Polymer Materials Engineering (sklpme2021-4-02 [C.C.]), China Postdoctoral Science Foundation (2019M663525 [M.R.B.]), Research Funding from West China School/Hospital of Stomatology Sichuan University (RCDWJS2023-16 [M.R.B.]), Research and Develop Program, West China Hospital of Stomatology Sichuan University (RD-02-202206 [M.R.B.]), and Med-X Innovation Programme of Med-X Center for Materials, Sichuan University (MCMGD202301 [L.Y.]). Prof. Mohsen Adeli would like to thank Iran Science Elites Federation and Iran National Science Foundation (Project Number 4001281) for the financial support. We gratefully acknowledge Dr. Mi Zhou and Dr. Chao He for their analytical support. We also thank Qiang Guo (State Key Laboratory of Oral Diseases, West China Hospital of Stomatology, Sichuan University) for characterizing micro-CT and Ning Gi (State Key Laboratory of Oral Diseases, West China Hospital of Stomatology, Sichuan University) for animal care.

## Author contributions

M.R.B. and T.W. contributed equally to this work. M.R.B., T.W., Z.Y.X., H.J.H., and X.Z.W. performed the experiments and analyzed the results. M.R.B., T.W., M.A., M.W., and X.L.H. assisted with the figure production and experimental design. M.R.B., T.W., L.Y., and C.C. wrote the manuscript. L.Y. and C.C. designed the experiments, corrected the manuscript, and supervised the whole project. All authors discussed the results and commented on the manuscript.

## Competing interests

The authors declare no competing interests.
