## [Peer Review File · Nature Communications]

REVIEWER COMMENTS

Reviewer #1 (Remarks to the Author):

This is a good piece of work that explores the development of an efficient, versatile, and precise electron-donable heterojunction with synergetic Ru-Cu pair sites (Ru-Cu/EDHJ) for superior biocatalytic regeneration of inflammatory mandible defects and pH-controlled antibacterial therapies. Through very detailed theoretical calculations, the authors elucidated the catalytic mechanism of the unique Ru-Cu pair sites on smart ROS regulation. The authors also comprehensively demonstrated that the biocompatible heterojunction biomaterials (Ru-Cu/EDHJ) could simultaneously exhibit superior repair function and bioadaptive antibacterial properties in treating mandible defects by switching between ROS scavenging (neutral media) and ROS generation (acidic media) capabilities. I think this idea on bioadaptive ROS-catalytic materials is interesting and would be highly potential biomaterials for translational and clinical applications. For the point of the regenerative medicine field, this work provides an important and promising pathway for treating oxidative stress-related inflammatory diseases, such as trauma, bone defects, arthritis, enteritis, and many other ROS-related ones. In general, I think this work shows high novelty, good translational promise, and high clinical value, and the manuscript is well-written and readable. Therefore, I recommend its publication in Nature Communications after addressing the following minor concerns:

1. The authors should include additional AC-STEM images to provide a more detailed and comprehensive observation of the single atom Ru.
2. For the selection of CeO₂, it is reasonable since its wide report for promising clinical applications in bone repair. Are there many earlier studies on the Cu₂O and CuO? I have concerns about the toxicity and degradation of copper oxides, the authors may give more information on these points.
3. For the pH-controlled bioadaptive ROS regulation, this is a quite new concept, the authors need to expound the mechanism of different catalytic activities triggered by acidic and neutral environments further.
4. In the part of "In vitro ROS scavenging and stem cell protection of Ru-Cu/EDHJ", the function of Ru-Cu/EDHJ can be reflected by KEGG pathways. I suggest that adding some relative references after some molecular pathways is indispensable. e.g. in the sentence "the Ru-Cu/EDHJ can protect stem cells by changing inflammatory expression", I would recommend including references to reveal the correlation between inflammation and these five signaling pathways.
5. For antimicrobial therapy, the authors provided abundant proof by using *E. coli* and *S. aureus* at the same time, indicating that Ru-Cu/EDHJ may have excellent application prospects. In the first sentence of the last part, "Bacterial infection often leads to refractory mandibular diseases, such as osteomyelitis and refractory apical periodontitis.", it will be more comprehensive to add some disease examples to show broad medical applications.
6. In the last part of "pH-controlled antibacterial action", the authors may try to express Ru-Cu/EDHJ showing POD-mimetic activities to kill bacteria under acidic environment. It is necessary to provide a more detailed description of it.

Reviewer #2 (Remarks to the Author):

The development of biocompatible materials with biocatalytic properties for the human tissue microenvironment repair is on high demand nowadays. In this work the authors presented a novel material able to both aid bone regeneration (mandible) and address antibacterial properties crucial for oral health applications. The authors successfully synthesized the novel biomaterials and proved their unique bioadaptive properties for controlling the level of ROS-generation or scavenging using in vitro and in vivo models. The authors claim that the material is reported as having dual (but in reality third aspect is shown) function regarding pro-survival of adult stem cells (probably skeletal progenitors) due to modulation of ROS and prevention of apoptosis and DNA damage. At the same time the material is claimed as immunomodulatory and also when exposed to acidic environment it can also possess antibacterial properties. While evidence is shown for all the above mentioned processes and functionalities in my opinion the overall aim i.e. to showcase all these processes instantaneously for bone defect healing are not provided. This weakens the message of the story substantially.

General comments:

In vitro ROS scavenging was assessed by using human MSCs however this is not necessarily reflective of mouse biology – additional results using mouse MSCs should be also performed in order to clearly validate mechanistically link and explain later observations of the in vivo experiments. Moreover more convincing data showing the influence of the materials activity during (both human and mouse) osteogenic differentiation should be provided.

Although the functionality of the material is tested in inflammatory mandible defects little discussion or evidence is provided regarding the functionality and influence of this material in immune regulation (in Fig 7 reduction in proinflammatory TNF α is provided but this is barely minimal).

More attention should be paid to the toxicity of the synthesized biocatalyst by itself. The experiments in vitro included groups of biocatalyst plus H₂O₂, but not biocatalysts alone. The proper IC₅₀ concentrations must be detected and different range of compounds must be applied. The authors should be aware as well, the method for IC₅₀ detection should be chosen wisely and normalized to the seeding rate together with untreated control.

Moreover, short-term incubation 1h was chosen to assess toxicity in vitro, although the long-term toxicity assessment should be considered (at least 48h). The in vitro tests were not consistent in terms of recovery time, so assays were performed immediately, or after different period of times (24, 48, 72h) from 1h incubation with tested compounds. Can authors comment on that? The same comment is regarding antibacterial effect of compounds in the H₂O₂ presence.

In addition, the concentration 100 μ M hydrogen peroxide was used through all experiments. The authors should provide motivation of the chosen concentration, is the expected physiological concentration during first steps of the inflammatory reaction?

The classic histological staining of mandibular defect healing should be provided and the bone restoration should be described.

Did implanted biocatalysts stayed after defect healing and can it be a problem for further biocatalysts applications?

More motivation of the mandibular defect model should be provided and untreated controls should be more characterized.

Why authors did not started with bone defect alone (without LPS-stimulation) and check biocatalyst effect on the 'natural' inflammatory phase of bone healing?

The manuscripts does not contain a literature supported-discussion section. There are large result sections that lacked referencing so a more detailed comparison of provided results with published literature should have been carried out.

Specific comments:

On page 13 (Subsection In vitro ROS scavenging and stem cell protection of Ru-Cu/EDHJ), the authors refer to the Supplementary Fig. 31, but the figure captives are wrong (refer to in vivo work).

In the Cellular live/dead staining Supp methods section, it was described the assessment of cytotoxicity, and results were presented in the supplement figure 32. Regarding to the methodology, the authors calculated live-dead cells after 48h of incubation with compounds. The final evaluation of dead/live cells were based on the cells, which were left attached to the plastic. Talking about hMSC, the population of dead cells is detaching from plastic and consequently, was washed away and excluded from assay. Additionally, 5000 cells were seeded per well and were there the same amount of cells after 5000?

Apoptosis, programmed cell death, occurs very fast as a response on the correspond stimuli (<https://www.ncbi.nlm.nih.gov/pmc/articles/PMC4966906/>). However, the authors exposed hMSC to biocatalysts plus H₂O₂ for 1h, but evaluated the level of apoptosis after 72h. It could be the same problem, that the most responsive cells were detached and washed away and excluded from analysis. For the next 72h, the mere resistant cells survived and proliferate.

Fig5G. The threshold level for PI positive cells is set between 102 – 103. May you provide unstained controls in Suppl data to justify threshold choice.

Page 15: Authors should also provide additional markers to assess in vitro osteogenic differentiation. This is crucial for dissecting the mechanism and effect of the novel material in a relevant assay. Additional immunostaining should be carried out for Osteocalcin, ALP and the alizarin red staining in order to prove osteogenic hMSC differentiation, the synthesis of COL1A1 is not sufficient to claim preserved osteogenic differentiation capacities.

Figure7 is lacking proper immunohistochemistry data of the regenerated defect. There should be additional insights on the regenerative process and outcome. There should be a link between the in vitro tests and the in vivo outcome which is currently not so tightly connected.

Should there be additional data demonstrating antibacterial activity in an in vivo setting ?
How would this affect pro-survival of MSCs?

Regarding bone defect models. May authors add the physical bone defect dimensions in the methods section, currently it is mentioned BR-49 bur. What was the physical dimensions of the gelatin sponge implanted? Did it dissolve after 8 weeks of healing?

Reviewer #3 (Remarks to the Author):

Point-by-point response to the detailed comments by reviewers of “*Electron-Donable Heterojunctions with Synergetic Ru-Cu Pair Sites for Biocatalytic Microenvironment Modulations in Inflammatory Mandible Defects*” with manuscript ID: NCOMMS-24-12604A.

REVIEWER COMMENTS

Reviewer #1 (Remarks to the Author):

“This is a good piece of work that explores the development of an efficient, versatile, and precise electron-donable heterojunction with synergetic Ru-Cu pair sites (Ru-Cu/EDHJ) for superior biocatalytic regeneration of inflammatory mandible defects and pH-controlled antibacterial therapies. Through very detailed theoretical calculations, the authors elucidated the catalytic mechanism of the unique Ru-Cu pair sites on smart ROS regulation. The authors also comprehensively demonstrated that the biocompatible heterojunction biomaterials (Ru-Cu/EDHJ) could simultaneously exhibit superior repair function and bioadaptive antibacterial properties in treating mandible defects by switching between ROS scavenging (neutral media) and ROS generation (acidic media) capabilities. I think this idea on bioadaptive ROS-catalytic materials is interesting and would be highly potential biomaterials for translational and clinical applications. For the point of the regenerative medicine field, this work provides an important and promising pathway for treating oxidative stress-related inflammatory diseases, such as trauma, bone defects, arthritis, enteritis, and many other ROS-related ones. In general, I think this work shows high novelty, good translational promise, and high clinical value, and the manuscript is well-written and readable. Therefore, I recommend its publication in Nature Communications after addressing the following minor concerns.”

Response to the general comment:

We sincerely appreciate your recognition of our bioadaptive ROS-catalytic materials as highly promising biomaterials for translational and clinical applications. Based on your comments and the suggestions from other reviewers, we have conducted more systematic experiments and refined the content throughout the manuscript. All necessary data have been added to support our claims, and we have thoroughly addressed all questions and concerns in the revised manuscript and supplementary

information. Therefore, we believe that the quality of this paper has been significantly enhanced. We hope you will agree with this assessment, and we thank you once again for your considerable efforts.

(1) The authors should include additional AC-STEM images to provide a more detailed and comprehensive observation of the single atom Ru.

Response to comment:

Thank you for your valuable comments. We have provided additional aberration-corrected high-angle annular dark-field scanning transmission electron microscopy (AC-HAADF-STEM) images in Supplementary Fig. 8 to enhance the clarity of our observations regarding the single atom Ru. These images provide a more detailed view of the Ru atoms, which appear as brighter dots due to their higher relative atomic mass compared to the Cu atoms in the Cu₂O supports. Our analysis of AC-HAADF-STEM images has confirmed that the Ru atoms are indeed dispersed as individual entities on the Cu₂O surface, as highlighted by the yellow circles. This observation underscores the successful synthesis of a single-atom Ru catalyst, which is crucial for understanding its unique catalytic properties. Moreover, we have also performed energy-dispersive X-ray spectroscopy (EDS) mapping to validate our findings further. The results confirm that Ru does not aggregate into clusters or nanoparticles, reinforcing the conclusion that it exists in a monoatomic state. We believe that these additional data significantly strengthen our conclusions and provide a more comprehensive understanding of the Ru-Cu/EDHJ. Thank you again for your valuable input, which has helped us to improve the quality of our work.

Supplementary Fig. 8. a-c Atomic-scale HAADF-STEM images and d-e EDS mapping of Ru-Cu/EDHJ.

(2) For the selection of CeO₂, it is reasonable since its wide report for promising clinical applications in bone repair. Are there many earlier studies on the Cu₂O and CuO? I have concerns about the toxicity and degradation of copper oxides, the authors may give more information on these points.

Response to comment:

Thanks for your good comments and helpful suggestions. We acknowledge that the toxicity and degradability of copper oxides are key considerations in bone repair. To ensure the degradability and safety of our materials, it is essential to meticulously monitor copper ion levels and evaluate their effects on normal tissue cells. We will carefully respond to your comments in the following three parts: 1) early literature studies on copper oxides, 2) degradation assessment of materials, and 3) evaluation of cytotoxicity.

1) Early literature studies on copper oxides. In the field of biomedicine, abundant studies have investigated the toxicity and degradation of copper oxides. Copper is an essential trace element that plays a significant role in various enzymatic processes in the human body. It is associated with low long-term toxicity and demonstrates high biosafety both *in vivo* and *in vitro* (*Nat. Commun.*, **2020**, 11, 2788; *Adv. Mater.*, **2020**, 32, 2004647). Additionally, research indicates that small-sized copper oxides can be effectively removed by the kidneys, mitigating the risk of long-term toxicity associated with the retention of nanomaterials in the body (*J. Am. Chem. Soc.*, **2019**, 141, 1091-1099). The material we have designed has a diameter of approximately 50 to 60 nm, a size that facilitates more efficient degradation and metabolic clearance post treatment. This characteristic suggests the significant potential of Ru-Cu/EDHJ for application in treating reactive oxygen species (ROS)-related diseases.

2) Degradation assessment of materials. Firstly, we evaluated the degradation performance of the Ru-Cu/EDHJ through *in vitro* experiments designed to simulate physiological conditions. Specifically, we immersed 1 mg of the Ru-Cu/EDHJ in 10 mL of phosphate-buffered saline (PBS) to simulate the *in vivo* treatment. The inductively coupled plasma mass spectrometry (ICP-MS) analysis demonstrates that Ru-Cu/EDHJ gradually degrades, especially the copper ions reach a 50% degradation ratio after 4 weeks (Supplementary Fig. 44). Thereafter, we evaluated the *in vivo* degradation performance of the Ru-Cu/EDHJ. Transmission electron microscopy (TEM) and ICP-MS results indicate a marked reduction of Ru-Cu/EDHJ within the regenerated new bone after 8 weeks, which is significantly different compared to the 1-week time point (Supplementary Fig. 45). Importantly, we found that the Ru-Cu/EDHJ composite could be excreted through urine and feces (Supplementary Fig. 45).

3) Evaluation of cytotoxicity. Hematoxylin and Eosin (H&E) staining of the major metabolic organs (heart, liver, spleen, lung, and kidney) in rats reveals no significant toxicity of Ru-Cu/EDHJ, indicating a satisfactory biosafety profile (Supplementary Fig. 46).

These findings collectively indicate that Ru-Cu/EDHJ exhibits strong bone induction capabilities while maintaining excellent biosafety, achieving an optimal balance between bone remodeling and material degradation. We believe these results significantly contribute to the understanding of our material's performance and its potential applications in bone repair. The corresponding details have been added in the revised manuscript and revised supplementary information, as also shown below:

Page 22 in the revised manuscript: “We further investigated the degradation and biosafety of Ru-Cu/EDHJ after mandible repair. The inductively coupled plasma mass spectrometry (ICP-MS) analysis demonstrates that Ru-Cu/EDHJ gradually degraded, especially the copper ions (Supplementary Fig. 44). TEM and ICP-MS results indicate a marked reduction of Ru-Cu/EDHJ within the regenerated new bone after 8 weeks, which is significantly different compared to the 1-week time point (Supplementary Fig. 45). Importantly, we found that the Ru-Cu/EDHJ composite could be excreted through urine and feces (Supplementary Fig. 45)⁹⁷. Meanwhile, the Ru-Cu/EDHJ group presents good biocompatibility with no noticeable damage indications to the viscus (heart, liver, spleen, lung, and kidney, Supplementary Fig. 46) and body weight (Supplementary Fig. 47). These findings collectively indicate that Ru-Cu/EDHJ exhibits excellent biosafety.”

Supplementary Fig. 44. Inductively coupled plasma mass spectrometry (ICP-MS) was used to measure the concentration of ions released at different times. 1 mg of the Ru-Cu/EDHJ was immersed in 10 mL PBS to explore the degradation performance of the material over time *in vitro*.

Supplementary Fig. 45. a The TEM images showcase the Ru-Cu/EDHJ materials obtained from the mandibular tissues captured at various time points after the implantation for 1 week and 8 weeks. Mandibular tissue samples were first fixed in osmium tetroxide, then dehydrated in acetone, followed by embedding in epoxy resin, and finally sectioned into ultra-thin slices for TEM imaging. **b** The ICP-MS test measured different metal ion concentrations in mandibular tissue after the implantation of the Ru-Cu/EDHJ for 1 week and 8 weeks. **c** The ICP-MS test measured different metal ion concentrations in mouse urine and feces after the implantation of the Ru-Cu/EDHJ for 4 weeks.

Supplementary Fig. 46. Paraffin-embedded heart, liver, spleen, lung, and kidney of mice on 14 days post-operation were sectioned and stained by H&E (n = 3 independent replicates). Scale bars: 100 μ m.

(3) For the pH-controlled bioadaptive ROS regulation, this is a quite new concept, the authors need to expound the mechanism of different catalytic activities triggered by acidic and neutral environments further.

Response to comment:

We are grateful to the reviewers for this valuable suggestion and positive comment on our concept of pH-controlled bioadaptive ROS regulation. We agree that investigating the underlying reasons for the different catalytic activities triggered by acidic and neutral environments is highly useful for further understanding the catalytic mechanisms and validating the originality of our study. Further details will be provided in the subsequent discussion.

Density functional theory (DFT) calculations indicate that variations in the hydrogen ion (H^+) concentrations, particularly under acidic and neutral conditions, lead to distinct reaction pathways and products when the catalyst and H_2O_2 interact. The Ru-Cu/EDHJ catalyst has demonstrated the ability to spontaneously catalyze H_2O_2 , generating a significant quantity of $*OH$ intermediates (with Gibbs free energy of -3.658 eV) that promote the formation of OH-enriched surfaces (*Adv. Mater.*, **2022**, 34, 2206208). Consequently, we employed $*OH+*OH$ as the reaction initial state (**S1/S1'**) in our study. The detailed reaction pathway (Supplementary Figs. 49, 50) proceeds as follows: initially, the Ru active site adsorbs an H_2O_2 molecule, leading to the release of H_2O and the subsequent formation of an $*O$ intermediate (**S3/S3'**). In a neutral environment, H_2O_2 interacts with $*O$ to result in the creation of $*OO$ and $*H_2O$ (**S4**), thereby exhibiting catalase-like activity. Conversely, in an acidic environment, H^+ exhibits a greater propensity to interact with $*O$ (with a Gibbs free energy barrier of -1.16 eV) compared to H_2O_2 (with a Gibbs free energy barrier of -0.17 eV), giving rise to $*OH$ intermediates (**S4'**). Subsequent re-adsorption of H_2O_2 leads to the generation of $*H_2O$ and $*OOH$ (**S5'**), thereby showing POD-like activity. Notably, OOH is the precursor to produce $\bullet O_2^-$ and 1O_2 (*Nat. Commun.*, **2024**, 15, 1010; *J. Am. Chem. Soc.*, **2023**, 145, 8965-8978).

In summary, the proton source in neutral conditions is solely H_2O_2 , leading the active sites to predominantly interact with H_2O_2 and display catalase-like activity, whereas under acidic conditions, the proton sources include both H^+ and H_2O_2 , and H^+ has a higher affinity for the $*O$ intermediates than H_2O_2 , thus resulting in peroxidase-like activity. The corresponding mechanism of different

catalytic activities triggered by acidic and neutral environments has been added in the revised manuscript and revised supplementary information, as also shown below:

Page 24 in the revised manuscript: “Theoretical calculations are utilized to investigate the mechanisms of different catalytic activities triggered by acidic and neutral environments. As shown in the Supplementary Figs. 49, 50, the Ru active site initially adsorbs an H_2O_2 molecule, leading to the release of H_2O and the subsequent formation of an $^*\text{O}$ intermediate. In a neutral environment, where the proton source is H_2O_2 , $^*\text{O}$ combines with H_2O_2 to release H_2O and O_2 , thereby exhibiting CAT-like activity. In contrast, within an acidic environment, where the proton sources are H^+ and H_2O_2 , the higher affinity of H^+ for $^*\text{O}$ (with a Gibbs free energy barrier of -1.16 eV) in comparison to H_2O_2 (with a Gibbs free energy barrier of -0.17 eV) leads to its preferential surface interaction, generating $^*\text{OH}$ species. Subsequently, $^*\text{OH}$ re-adsorbs with another H_2O_2 molecule to yield OOH and H_2O , showcasing POD-like activity.”

Supplementary Fig. 49. CAT-like and POD-like pathways of Ru-Cu/EDHJ. The Ru-Cu/EDHJ catalyst has demonstrated the ability to spontaneously catalyze H_2O_2 , generating a significant quantity of $^*\text{OH}$ intermediates (with Gibbs free energy of -3.658 eV) that promote the formation of OH-enriched surfaces. Consequently, we employed $^*\text{OH} + ^*\text{OH}$ as the reaction initial state (S1/S1') in this part.

Supplementary Fig. 50. Gibbs free energy diagram of corresponding CAT-like and POD-like pathways for Ru-Cu/EDHJ.

(4) In the part of "In vitro ROS scavenging and stem cell protection of Ru-Cu/EDHJ", the function of Ru-Cu/EDHJ can be reflected by KEGG pathways. I suggest that adding some relative references after some molecular pathways is indispensable. e.g., in the sentence "the Ru-Cu/EDHJ can protect stem cells by changing inflammatory expression", I would recommend including references to reveal the correlation between inflammation and these five signaling pathways.

Response to comment:

Thank you for your good comments and helpful suggestions. We have included references to reveal the correlation between inflammation and these five signaling pathways. The corresponding supporting literature has been added in the revised manuscript, as also shown below:

Page 19 in the revised manuscript: "The function of Ru-Cu/EDHJ can be reflected in influencing cell cytoskeleton and adhesion (regulation of actin cytoskeleton, focal adhesion, Rap1 signaling pathway⁸⁰), cellular DNA damage and function, cellular activity (ECM-receptor interaction, cellular senescence, cell cycle), and cell apoptosis (apoptosis, p53 signaling pathway⁸¹). In addition, the Ru-Cu/EDHJ can protect stem cells by changing inflammatory expression (inflammatory mediator regulation of TRP channels, NF-kappa B signaling pathway⁸², IL-17 signaling pathway⁸³,

JAK-STAT signaling pathway⁸⁴, and TNF signaling pathway⁸⁵), stem cell development and matrix formation (signaling pathways regulating pluripotency of stem cells, ECM-receptor interaction, and MAPK signaling pathway⁸⁶), and subsequently osteogenesis regulation (Wnt signaling pathway⁸⁷, PI3K-Akt signaling pathway⁸⁸, and TGF-beta signaling pathway⁸⁹).”

References:

- 80 Kondo, N., Ueda, Y. & Kinashi, T. Kindlin-3 disrupts an intersubunit association in the integrin LFA1 to trigger positive feedback activation by Rap1 and talin 1. *Sci. Signaling* **14**, eabf2184 (2021).
- 81 Wang, X., Simpson, E. R. & Brown, K. A. p53: Protection against Tumor Growth beyond Effects on Cell Cycle and Apoptosis. *Cancer Res.* **75**, 5001-5007 (2015).
- 82 Yu, H., Lin, L., Zhang, Z., Zhang, H. & Hu, H. Targeting NF-κB pathway for the therapy of diseases: mechanism and clinical study. *Signal Transduction Targeted Ther.* **5**, 77-92 (2020).
- 83 Zhang, Z. *et al.* TAOK1 negatively regulates IL-17-mediated signaling and inflammation. *Cell. Mol. Immunol* **15**, 794-802 (2018).
- 84 Wang, L. *et al.* Targeting JAK/STAT signaling pathways in treatment of inflammatory bowel disease. *Inflammation Res.* **70**, 753-764 (2021).
- 85 Coste, E. *et al.* Identification of small molecule inhibitors of RANKL and TNF signalling as anti-inflammatory and antiresorptive agents in mice. *Ann. Rheum. Dis.* **74**, 220-226 (2015).
- 86 Nam, O. H. *et al.* Ginsenoside Rb1 alleviates lipopolysaccharide-induced inflammation in human dental pulp cells via the PI3K/Akt, NF-κB, and MAPK signalling pathways. *Int. Endod. J.* **57**, 759-768 (2024).
- 87 Leng, Y. *et al.* Osteoblast-derived exosomes promote osteogenic differentiation of osteosarcoma cells via URG4/Wnt signaling pathway. *Bone* **178**, 116933 (2024).
- 88 Tang, L. *et al.* Fgf9 Negatively Regulates Bone Mass by Inhibiting Osteogenesis and Promoting Osteoclastogenesis Via MAPK and PI3K/AKT Signaling. *J. Bone Miner. Res.* **36**, 779-791 (2021).
- 89 Oka, K. *et al.* The role of TGF-β signaling in regulating chondrogenesis and osteogenesis during mandibular development. *Dev. Biol.* **303**, 391-404 (2007).

(5) For antimicrobial therapy, the authors provided abundant proof by using *E. coli* and *S. aureus* at the same time, indicating that Ru-Cu/EDHJ may have excellent application prospects. In the first sentence of the last part, "Bacterial infection often leads to refractory mandibular diseases, such as osteomyelitis and refractory apical periodontitis.", it will be more comprehensive to add some disease examples to show broad medical applications.

Response to comment:

Thank you for your insightful comments and helpful suggestions. We appreciate your recommendation to include examples of refractory mandibular diseases, which will undoubtedly enhance the comprehensiveness of our manuscript and showcase the broader medical implications. In response to your feedback, we have expanded our discussion in the revised manuscript to include additional descriptions of infectious mandibular diseases associated with bacterial infections beyond our original focus on osteomyelitis and apical periodontitis, as also shown below:

Page 24 in the revised manuscript: “Bacterial infection often leads to infectious jaw diseases, such as osteomyelitis⁹⁸, apical periodontitis⁹⁹, periodontitis¹⁰⁰, peri-implantitis¹⁰¹, and mandibular fractures with infection¹⁰² and so on.”

References:

- 98 Sodnom-Ish, B. *et al.* Decompression effects on bone healing in rat mandible osteomyelitis. *Sci. Rep.* **11**, 11673 (2021).
- 99 Dai, X. Z. *et al.* Enterococcus faecalis-Induced Macrophage Necroptosis Promotes Refractory Apical Periodontitis. *Microbiol. Spectrum* **10**, e0104522 (2022).
- 100 Slots, J. Periodontitis: facts, fallacies and the future. *Periodontol. 2000* **75**, 7-23 (2017).
- 101 Wang, C.-W. *et al.* Laser-assisted regenerative surgical therapy for peri-implantitis: A randomized controlled clinical trial. *J. Periodontol.* **92**, 378-388 (2021).
- 102 Oksa, M., Haapanen, A., Marttila, E. & Snall, J. Simple dentate area fractures of the mandible - can we prevent postoperative infections? *Acta Odontol. Scand.* **80**, 494-500 (2022).

(6) In the last part of "pH-controlled antibacterial action ", the authors may try to express Ru-Cu/EDHJ showing POD-mimetic activities to kill bacteria under acidic environment. It is necessary to provide a more detailed description of it.

Response to comment:

We are grateful to your insightful comments. We have included a more comprehensive description of Ru-Cu/EDHJ and its POD-mimetic activities in effectively killing bacteria in acidic environments. We believe this enhancement significantly improves the quality of our manuscript, and the corresponding details can be found in the revised manuscript, as also shown below:

Page 24 in the revised manuscript: “Thus, it would be significantly valuable if the Ru-Cu/EDHJ could also exhibit bioadaptive antibacterial activity. **POD-like activity of materials enables the local**

transformation of H₂O₂ to generate free radicals, which results in oxidative damage to bacteria by destabilizing and inactivating cytoplasmic proteins and the bacterial cell envelope¹⁰⁴. We then assess the POD-mimetic activities of materials on converting H₂O₂ into ROS (Fig. 8a and Supplementary Fig. 48), the Ru-Cu/EDHJ exhibits the highest POD-mimetic activity in an acidic environment, which surpasses that of CeO₂, EDHJ, and Ru-O/EDHJ, highlighting its favorable ROS-generation performance. Further quenching experiments reveal that the ROS produced by Ru-Cu/EDHJ with H₂O₂ in an acidic environment primarily consists of •O₂⁻ and ¹O₂ (Fig. 8b). Conversely, a neutral environment does not induce any POD-like activity, indicating that the Ru-Cu/EDHJ do not generate ROS or cause cellular damage under normal tissues (neutral conditions, Fig. 8c).”

References:

- 104 Ezraty, B., Gennaris, A., Barras, F. & Collet, J.-F. Oxidative stress, protein damage and repair in bacteria. *Nat. Rev. Microbiol.* **15**, 385-396 (2017).

Reviewer #2 (Remarks to the Author):

“The development of biocompatible materials with biocatalytic properties for the human tissue microenvironment repair is on high demand nowadays. In this work the authors presented a novel material able to both aid bone regeneration (mandible) and address antibacterial properties crucial for oral health applications. The authors successfully synthesized the novel biomaterials and proved their unique bioadaptive properties for controlling the level of ROS-generation or scavenging using in vitro and in vivo models. The authors claim that the material is reported as having dual (but in reality third aspect is shown) function regarding pro-survival of adult stem cells (probably skeletal progenitors) due to modulation of ROS and prevention of apoptosis and DNA damage. At the same time the material is claimed as immunomodulatory and also when exposed to acidic environment it can also possess antibacterial properties. While evidence is shown for all the above mentioned processes and functionalities in my opinion the overall aim i.e. to showcase all these processes instantaneously for bone defect healing are not provided. This weakens the message of the story substantially.”

Response to the general comment:

We are grateful for the constructive feedbacks on our manuscript. Your comments and suggestions have been instrumental in enhancing the quality of our work. In response to your insights, we have thoroughly revised the manuscript to provide a clearer exposition of the processes underlying bone defect healing. All queries and concerns have been addressed in the revised manuscript and the accompanying supplementary material. Your guidance has been invaluable in refining our study, and we are confident that these revisions have substantially improved the paper. We extend our thanks once again for your insightful contributions.

(1) In vitro ROS scavenging was assessed by using human MSCs; however, this is not necessarily reflective of mouse biology – additional results using mouse MSCs should also be performed in order to clearly validate mechanistically link and explain later observations of the in vivo experiments. Moreover, more convincing data showing the influence of the materials activity during (both human and mouse) osteogenic differentiation should be provided.

Response to comment:

Thanks for your important and helpful comments on improving the quality of our manuscript. We acknowledge your point that assessing *in vitro* ROS scavenging in human MSCs may not fully represent mouse biology. To address this, we have added mouse bone mesenchymal stem cells (mMSCs) related experiments, which will help to establish a clear mechanistic connection and elucidate our *in vivo* findings.

Initially, we employed 2,7-dichlorodihydrofluorescein diacetate (DCFH-DA) as a fluorescence probe to assess intracellular ROS levels, as shown in Supplementary Fig. 31. Following treatment of mMSCs with H₂O₂ (100 μM), a pronounced green fluorescence is observed, indicating elevated intracellular ROS levels. Notably, the green fluorescence signals are significantly diminished after treatment with Ru-Cu/EDHJ, while Ru-O/EDHJ produces only a modest reduction in fluorescence intensity. Furthermore, neither the CeO₂ nor the EDHJ (CeO₂@Cu₂O) groups exhibit noticeable effects in reducing ROS levels. These findings suggest that Ru-Cu/EDHJ holds promise for

applications in the removal of ROS and the protection of mMSCs from oxidative stress. The corresponding details can be found in the revised manuscript and revised supplementary information, as also shown below:

Page 14 in the revised manuscript: “After validating that the Ru-Cu/EDHJ reveals remarkable and versatile antioxidase-mimetic activities, we then systematically examine its application potentials for ROS scavenging and stem cell protection via using the **mesenchymal stem cells (MSCs) from bone marrow** as model systems. The concentration of 100 μM H_2O_2 *in vitro* is chosen according to the estimated physiological concentration during the inflammatory reaction^{68,69}. 2,7-dichlorodihydrofluorescein diacetate (DCFH-DA) is applied as the fluorescence probe to explore intracellular ROS levels, as shown in Figs. 4a-c and Supplementary Fig. 31. After treating with H_2O_2 , apparent green fluorescence can be observed in both **human mesenchymal stem cells (hMSCs) and mouse mesenchymal stem cells (mMSCs)**, demonstrating a high level of intracellular ROS. It can be noted that the green fluorescence signals can be dramatically attenuated after being treated with Ru-Cu/EDHJ, while the Ru-O/EDHJ can only slightly decrease the signal, and the CeO_2 or EDHJ ($\text{CeO}_2@\text{Cu}_2\text{O}$) group shows no noticeable effects on decrease ROS level.”

Moreover, we conducted experiments using alkaline phosphatase (ALP) and alizarin red (AR) staining to investigate the effects of hMSCs and mMSCs on osteogenic differentiation. The group treated with H_2O_2 demonstrates a reduced expression of ALP and OCN (osteocalcin) and a decreased formation of calcium nodules by AR staining (Figs. 5j-l and Supplementary Figs. 35, 36). However, the impaired osteogenic phenotype induced by H_2O_2 is found to recover with the addition of Ru-Cu/EDHJ. Quantitative analysis of the ALP and AR staining areas indicates that elevated H_2O_2 levels could maintain the osteogenesis of hMSCs and mMSCs when supplemented with Ru-Cu/EDHJ. These findings suggest that treatment with Ru-Cu/EDHJ effectively protects the osteogenesis of hMSCs and mMSCs in high ROS environments and supports bone regeneration. The corresponding details have been added in the revised manuscript and revised supplementary information, as also shown below:

Page 17 in the revised manuscript: “To explore the protection of Ru-Cu/EDHJ on stem cell function, the *in vitro* osteogenic differentiation potential of hMSCs in high ROS levels is assessed by

alizarin red (AR) and alkaline phosphatase (ALP) staining⁷⁸. The group treated with H₂O₂ demonstrates a reduced expression of ALP and OCN (osteocalcin) and a decreased formation of calcium nodules by AR staining (Figs. 5j-l and Supplementary Figs. 35, 36). However, the impaired osteogenic phenotype induced by H₂O₂ is found to recover with the addition of Ru-Cu/EDHJ. Quantitative analysis of the ALP and AR staining areas indicates that elevated H₂O₂ levels do not inhibit both hMSCs and mMSCs osteogenesis when supplemented with Ru-Cu/EDHJ. These findings suggest that treatment with Ru-Cu/EDHJ effectively protects the osteogenesis of MSCs from bone marrow in high ROS environment, suggesting its support to bone regeneration (Fig. 5m)⁷⁸.”

Supplementary Fig. 31. **a** Fluorescence images, Ctrl (PBS), **b** and mean fluorescence intensity of 2,7-dichlorodihydrofluorescein diacetate (DCFH-DA) staining of ROS in mMSCs (n = 3 independent replicates), $p_{(H_2O_2)} < 0.0001$, $p_{(EDHJ-H_2O_2)} = 0.0144$, $p_{(Ru-O/EDHJ-H_2O_2)} < 0.0001$, $p_{(Ru-Cu/EDHJ-H_2O_2)} < 0.0001$. Scale bars: 100 μm. Data are presented as means ± SD., *** $p < 0.001$, # $p < 0.05$, ### $p < 0.001$; one-way ANOVA with multiple comparisons test.

Supplementary Fig. 35. Immunohistochemical staining images of osteocalcin (OCN) in hMSCs, orange arrow indicates the positive expression area (n = 3 independent replicates). Scale bars: 200 μ m.

Supplementary Fig. 36. a Alkaline phosphatase (ALP) staining after 3-day *in vitro* osteo-induction and alizarin red (AR) staining after 21-day *in vitro* osteo-induction using the mMSCs. Quantitative analysis of **b** ALP (n = 3 independent replicates), $p_{(H_2O_2)} < 0.0001$, $p_{(Ru-Cu/EDHJ-H_2O_2)} < 0.0001$, and **c** AR (n = 3 independent replicates) in H₂O₂ treated mMSCs, $p_{(H_2O_2)} < 0.0001$, $p_{(Ru-Cu/EDHJ-H_2O_2)} < 0.0001$. Data are presented as means \pm SD., *** $p < 0.001$, ### $p < 0.001$; one-way ANOVA with multiple comparisons test.

Fig. 5 Prevention of ROS-related DNA damage and apoptosis in MSCs by Ru-Cu/EDHJ. j ALP staining after 3-day *in vitro* osteo-induction and AR staining after 21-day *in vitro* osteo-induction using the hMSCs. Quantitative analysis of **k** ALP (n = 3 independent replicates), $p_{(H_2O_2)} < 0.0001$, $p_{(Ru-Cu/EDHJ-H_2O_2)} < 0.0001$, and **l** AR (n = 3 independent replicates) in H₂O₂ treated hMSCs, $p_{(H_2O_2)} < 0.0001$, $p_{(Ru-Cu/EDHJ-H_2O_2)} < 0.0001$.

(2) Although the functionality of the material is tested in inflammatory mandible defects little discussion or evidence is provided regarding the functionality and influence of this material in immune regulation (in Fig 7 reduction in proinflammatory TNF α is provided but this is barely minimal).

Response to comment:

We sincerely appreciate your insightful and constructive comments aimed at enhancing the quality of our manuscript. We acknowledge the insufficient data in our research concerning the immune regulatory functions of Ru-Cu/EDHJ. In response to your feedback, we have included an analysis of the CD206 (M2 phenotype) and iNOS (M1 phenotype) markers through tissue section immunofluorescence double staining and quantitative assessment, focusing on the mandibular defect area (Supplementary Fig. 40). The resulting data indicate that the Ru-Cu/EDHJ group shows a less proportion of iNOS relative to CD206 compared to the other experimental groups, suggesting its efficacy in promoting macrophage polarization toward the M2 (anti-inflammatory) phenotype and

attenuating proinflammatory responses. These findings offer preliminary insights into the immune regulatory effects of Ru-Cu/EDHJ in the repair of inflammatory bone defects. Moving forward, we are committed to conducting a systematic investigation into the immune responses to assess the clinical applicability of this material further. We appreciate your thoughtful input, which has significantly informed our work. The corresponding details can be found in the revised manuscript and revised supplementary information, as also shown below:

Page 21 in the revised manuscript: “Moreover, we have included an analysis of the CD206 (M2 phenotype) and iNOS (M1 phenotype) markers through tissue section immunofluorescence double staining and quantitative assessment, focusing on the mandible defect area (Supplementary Fig. 40). The resulting data indicate that the Ru-Cu/EDHJ group shows a smaller proportion of iNOS relative to CD206 compared to the other experimental groups, suggesting its efficacy in promoting macrophage polarization toward the M2 (anti-inflammatory) phenotype and attenuating proinflammatory responses⁹².”

Supplementary Fig. 40. **a** Double immunofluorescence staining of tissue sections from mandibular defect area for iNOS and CD206 markers, and **b** corresponding quantitative analysis of the ratio of iNOS/CD206 ($n = 3$ independent replicates), $p_{(ROSup)} = 0.0304$, $p_{(Ru-Cu/EDHJ-ROSup)} = 0.0021$. Ctrl (mandible defects with PBS treatment). Data are presented as means \pm SD., * $p < 0.05$, ## $p < 0.01$, one-way ANOVA with multiple comparisons test. Scale bars: 200 μ m.

(3) *More attention should be paid to the toxicity of the synthesized biocatalyst by itself. The experiments in vitro included groups of biocatalyst plus H₂O₂, but not biocatalysts alone. The proper IC₅₀ concentrations must be detected and different range of compounds must be applied. The authors should be aware as well, the method for IC₅₀ detection should be chosen wisely and normalized to the seeding rate together with untreated control.*

Response to comment:

Thank you for your insightful comments regarding the toxicity of the synthesized biocatalyst. We have taken your advice and conducted cytotoxicity assessments using the Cell Counting Kit-8 (CCK-8) method, specifically examining the biocatalyst in the absence of H₂O₂ (Supplementary Fig. 32). Our results indicate a half maximal inhibitory concentration (IC₅₀) value of 426.4 µg/mL. The 100 µg/mL concentration value used in this research shows no significant effect on cell viability, suggesting that the Ru-Cu/EDHJ biocatalyst exhibits almost no cytotoxicity in low concentrations. Your insights have significantly contributed to refining our research, and we are grateful for your suggestions. The corresponding details can be found in the revised manuscript and revised supplementary information, as also shown below:

Page 15 in the revised manuscript: “The Cell Counting Kit-8 (CCK-8) method shows that the half maximal inhibitory concentration (IC₅₀) is 426.4 µg/mL and exhibits almost no cytotoxicity to cell viability in low concentration. The concentration value of 100 µg/mL used in this research shows no significant effect on cell viability by CCK-8 and Live/Dead staining, indicating the excellent biocompatibility of Ru-Cu/EDHJ biocatalyst (Supplementary Figs. 32, 33).”

Supplementary Fig. 32. Quantitative analysis of cell viability for Ru-Cu/EDHJ without H₂O₂ using the Cell Counting Kit-8 (CCK-8), inhibitory concentration (IC₅₀) = 426.4 µg/mL (n = 3 independent replicates). Data are presented as means ± SD., ***p* < 0.01, ****p* < 0.001; one-way ANOVA with multiple comparisons test.

(4) Moreover, short-term incubation 1h was chosen to assess toxicity *in vitro*, although the long-term toxicity assessment should be considered (at least 48h). The *in vitro* tests were not consistent in terms of recovery time, so assays were performed immediately, or after different periods of time (24, 48, 72h) from 1h incubation with tested compounds. Can authors comment on that? The same comment is regarding antibacterial effect of compounds in the H₂O₂ presence.

Response to comment:

We sincerely appreciate your valuable comments and constructive feedback, which have played an essential role in improving our manuscript. Here, we would like to respond to your comments from four parts comprehensively:

1) For the short-term incubation of 1 h with H₂O₂ or H₂O₂+Ru-Cu/EDHJ: our choice of a 1-hour H₂O₂ incubation aimed to induce a state of oxidative stress, a method of H₂O₂ stimulation in short periods has also been utilized in some literature (*Nat. Commun.*, **2022**, 13, 7739; *Stem Cells Dev.*, **2012**, 21, 1877-1886). Based on the literature and our preliminary experiment, hMSCs experienced reduced adhesion after just 1 hour of exposure to H₂O₂ (as shown in Fig. R1), which is sufficient to lead to certain cellular damage. When the cells were changed to a regular medium and cultured for another 12 h, it could be observed that these hMSCs maintained a damaged state and they couldn't recover to the original state, as shown in Fig. R1. Meanwhile, if the cells are treated with H₂O₂ for more than 6 hours, most of these MSCs can hardly survive, thus leading to very few live cells in the control group, which will affect our detailed observation and comparison. Therefore, we chose 1 hour of exposure to H₂O₂ for better comparison between groups.

Fig. R1. White light photomicrograph of hMSCs after stimulation by H₂O₂.

2) Regarding the 24 h recovery time in our study: for assessing early phenotypic changes post-stimulation, such as cell morphology, adhesion, and apoptosis, we standardized our recovery period to 24 hours, referencing methodologies from earlier studies (*Nat. Commun.*, **2024**, 15, 1643; *J. Biomed. Mater. Res. Part A*, **2021**, 109, 2580-2596; *Nat Commun.*, **2021**, 12, 1436). Additionally, combined with the 13th question, we have supplemented the apoptotic flow cytometry data at the 24-hour time point. In light of your feedback, we have checked the literature and noted that typical detection periods generally range from 12 to 48 hours post-stimulation (*Nat. Commun.*, **2022**, 13, 7739; *Nat. Commun.*, **2021**, 12, 1436; *Nat. Commun.*, **2022**, 13, 7449). Recognizing that our initial 72-hour timeframe was relatively extended, we have revised our approach and reduced the detection period to 24 hours (Figs. 5g, i).

Page 44 in the revised supplementary information manuscript: “**Cell apoptosis.** hMSCs were

seeded in 6 mm plates and allowed to adhere. After being treated with different samples for 1 h, cells were cultured in a fresh medium for another 24 h²⁹. Then, the cells in the supernatant and the digestive cells were centrifuged together to avoid the loss of cells. Then, the Annexin V-FITC/PI Apoptosis Detection Kit (AD10, Dojindo, Japan) was used to stain the cells before the flow cytometry analysis. According to the fluorescence intensity, cell populations were assigned into four quadrants, including live, early apoptotic, late apoptotic, and necrotic cells. The figures were formed by the software of Flowjo (version 10.8.1). Moreover, the tunnel staining followed the manufacturer's instructions for the One Step TUNEL Apoptosis Assay Kit (C1089, Beyotime Biotechnology, China), and the images were captured by a confocal laser scanning microscope.”

3) Regarding the assays on *in vitro* ROS levels that were performed immediately: for our *in vitro* ROS tests, we adhered to the established protocol and conducted 2,7-dichlorodihydrofluorescein diacetate (DCFH-DA) staining immediately after H₂O₂ stimulation to accurately capture the real-time changes in ROS levels, as noted in previous studies (*Chem. Sci.*, **2018**, 9, 2927-2933).

4) Regarding the necessity of a long-term toxicity assessment: as recommended in this comment, here, we have utilized both the CCK-8 method and Live/Dead staining to comprehensively assess the cytotoxicity of Ru-Cu/EDHJ for 48 hours (Supplementary Figs. 32, 33). The corresponding details can be found in the revised manuscript and revised supplementary information.

Regarding your comment on the antibacterial effect of compounds in the H₂O₂ presence: regarding the antibacterial effects of compounds in the presence of H₂O₂, we agree with the reviewer that, indeed, there are different treatment times and protocols in different publications. Here, in this work, we refer to some of the earlier established protocols (*Adv. Mater.*, **2022**, 34, 2108646; *Angew. Chem. Int. Ed.*, **2022**, 61, e202113833; *Adv. Funct. Mater.*, **2023**, 33, 2301986; *Adv. Mater.*, **2021**, 33, 2005477). Therefore, the treatment times for the planktonic bacteria and biofilms are different, and we have cited these previous publications to support the method in the revised manuscript.

(5) In addition, the concentration 100 μM hydrogen peroxide was used through all experiments. The authors should provide motivation of the chosen concentration, is the expected physiological concentration during first steps of the inflammatory reaction?

Response to comment:

We sincerely appreciate your valuable comment regarding the H_2O_2 concentration used in our experiments. We fully concur with the reviewer's suggestion that the rationale for selecting optimal concentrations for *in vitro* experiments should be grounded in *in vivo* microenvironments, as this approach is both more rigorous and compelling. Indeed, before we carry out this study, to ensure our experimental conditions reflect physiological realities, we have already reviewed abundant existing literature. Notably, studies have shown that the abdominal H_2O_2 levels were $85.4 \pm 5.3 \mu\text{M}$ in an LPS-induced abdomen inflammation model (*Adv. Funct. Mater.*, **2020**, 30, 2001771). In an osteoarthritis model, H_2O_2 levels increased to $120 \pm 6 \mu\text{M}$ in inflamed knees (*Adv. Funct. Mater.*, **2020**, 30, 2001771). The studies also established an infection model by subcutaneously inoculating *E. coli* into the left leg of mice, as gram-negative bacterial infections typically trigger acute inflammatory reactions. The infected regions corresponded to an estimated $96.5 \pm 5.1 \mu\text{M}$ H_2O_2 (*Adv. Funct. Mater.*, **2020**, 30, 2001771). Additionally, a dual ratiometric surface-enhanced Raman scattering (SERS) and photoacoustic nanoprobe study found H_2O_2 levels of $91.89 \pm 3.03 \mu\text{M}$ in rabbit models of knee osteoarthritis (*Angew. Chem. Int. Ed.*, **2021**, 60, 7323-7332). These findings suggest that our chosen concentration of $100 \mu\text{M}$ effectively mirrors physiological levels typically observed in the early inflammatory response. We have included this rationale in the revised manuscript to enhance clarity and justify our methodology, as also shown below:

Page 14 in the revised manuscript: “The concentration of $100 \mu\text{M}$ H_2O_2 *in vitro* is chosen according to the estimated physiological concentration during the inflammatory reaction^{68,69}.”

References:

- 68 Ye, J. *et al.* Quantitative Photoacoustic Diagnosis and Precise Treatment of Inflammation In Vivo Using Activatable Theranostic Nanoprobe. *Adv. Funct. Mater.* **30**, 2001771 (2020).
- 69 Li, Q. *et al.* Dual Ratiometric SERS and Photoacoustic Core-Satellite Nanoprobe for Quantitatively Visualizing Hydrogen Peroxide in Inflammation and Cancer. *Angew. Chem. Int.*

Ed. 60, 7323-7332 (2021).

(6) *The classic histological staining of mandibular defect healing should be provided and the bone restoration should be described.*

Response to comment:

We sincerely appreciate your valuable feedback, which has allowed us to improve the quality of this paper. In accordance with your recommendations, we have incorporated traditional histological staining techniques to detail the healing processes of mandibular defects. Hematoxylin and Eosin (H&E) staining demonstrates the efficacy of Ru-Cu/EDHJ in facilitating endogenous jaw bone regeneration. As illustrated in Supplemental Fig. 43, the ROSup and CeO₂ treatment groups exhibit significant inflammatory cell infiltration in the defect regions, persisting even after 8 weeks, which suggests prolonged inflammation and ongoing damage at the surgical site. The Ctrl group (bone defect + PBS) displays only a limited amount of newly formed bone matrix at the defect boundary. Conversely, the Ru-Cu/EDHJ group demonstrates well-structured new bone tissue with minimal inflammatory cell presence. Furthermore, the expression of osteocalcin (OCN) is significantly elevated in the Ru-Cu/EDHJ group, underscoring its potential role in promoting osteogenesis (*Biomaterials*, **2009**, 30, 2252-2258; *Nat. Commun.*, **2022**, 13, 2499). The corresponding details can be found in the revised manuscript and revised supplementary information, as also shown below:

Page 22 in the revised manuscript: “Hematoxylin and Eosin (H&E) staining is employed to evaluate the potential of the Ru-Cu/EDHJ in facilitating endogenous mandible regeneration⁹³. As depicted in Supplementary Fig. 43, the ROSup and CeO₂ groups show evident inflammatory cell infiltration in the defect regions, even after 8 weeks, indicating prolonged inflammation and ongoing damage in the surgical site. The Ctrl group exhibits only a limited amount of newly formed bone matrix in the defect boundary. Conversely, the Ru-Cu/EDHJ group demonstrates well-structured new bone tissues with minimal inflammatory cell presence. In addition, the expression of osteocalcin (OCN) significantly increases in the Ru-Cu/EDHJ group, highlighting its potential role in promoting osteogenesis^{95,96}.”

Supplementary Fig. 43. a Hematoxylin and Eosin (H&E) staining of regenerated bones induced by different scaffolds at week 8 after operation. Ctrl (mandible defects with PBS treatment). (Row 1: Overall observation of the mandible defect repair. Row 2: Magnified view of the center and boundary site of the defects). (N: new bone tissue. V: new blood vessels (black arrow). F: fibrous tissue. B: old bone. I: inflammatory infiltration). **b** Representative immunohistochemistry images of OCN. OCN (brown) and hematoxylin (blue). n = 3 independent replicates.

(7) *Did implanted biocatalysts stayed after defect healing and can it be a problem for further biocatalysts applications?*

Response to comment:

Thanks for your insightful and helpful comments. We agree that understanding whether these biocatalysts remain in the implantation site is crucial for evaluating their applicative value. We will provide a comprehensive discussion on the safety profiles of Ru-Cu/EDHJ, as well as their potential implications for future applications. We believe this additional analysis will enhance the clarity of our manuscript and address your concerns. We will carefully respond to your comments in the following two parts: 1) degradation assessment of materials and 2) evaluation of cytotoxicity.

1) Degradation assessment of materials. Firstly, we evaluated the degradation performance of the Ru-Cu/EDHJ through *in vitro* experiments designed to simulate physiological conditions. Specifically, we immersed 1 mg of the Ru-Cu/EDHJ in 10 mL of phosphate-buffered saline (PBS) to simulate the *in vivo* treatment. The inductively coupled plasma mass spectrometry (ICP-MS) analysis demonstrates that Ru-Cu/EDHJ gradually degrades, especially the copper ions reach a 50% degradation ratio after 4 weeks (Supplementary Fig. 44). Thereafter, we evaluated the *in vivo* degradation performance of the Ru-Cu/EDHJ. Transmission electron microscopy (TEM) and ICP-MS results indicate a marked reduction of Ru-Cu/EDHJ within the regenerated new bone after 8 weeks, which is significantly different compared to the 1-week time point (Supplementary Fig. 45). Importantly, we found that the Ru-Cu/EDHJ composite could be excreted through urine and feces (Supplementary Fig. 45).

2) Evaluation of cytotoxicity. Hematoxylin and Eosin (H&E) staining of the major metabolic organs (heart, liver, spleen, lung, and kidney) in rats reveals no significant toxicity of Ru-Cu/EDHJ, indicating a satisfactory biosafety profile (Supplementary Fig. 46).

These findings collectively indicate that Ru-Cu/EDHJ exhibits strong bone induction capabilities while maintaining excellent biosafety, achieving an optimal balance between bone remodeling and material degradation. We believe these results significantly contribute to the understanding of our material's performance and its potential applications in bone repair. The corresponding details have been added in the revised manuscript and revised supplementary information, as also shown below:

Page 22 in the revised manuscript: “We further investigated the degradation and biosafety of Ru-Cu/EDHJ after mandible repair. The inductively coupled plasma mass spectrometry (ICP-MS) analysis demonstrates that Ru-Cu/EDHJ gradually degraded, especially the copper ions (Supplementary Fig. 44). TEM and ICP-MS results indicate a marked reduction of Ru-Cu/EDHJ within the regenerated new bone after 8 weeks, which is significantly different compared to the 1-week time point (Supplementary Fig. 45). Importantly, we found that the Ru-Cu/EDHJ composite could be excreted through urine and feces (Supplementary Fig. 45)⁹⁷. Meanwhile, the Ru-Cu/EDHJ group presents good biocompatibility with no noticeable damage indications to the viscus (heart,

liver, spleen, lung, and kidney, Supplementary Fig. 46) and body weight (Supplementary Fig. 47). These findings collectively indicate that Ru-Cu/EDHJ exhibits excellent biosafety.”

Supplementary Fig. 44. Inductively coupled plasma mass spectrometry (ICP-MS) was used to measure the concentration of ions released at different times. 1 mg of the Ru-Cu/EDHJ was immersed in 10 mL PBS to explore the degradation performance of the material over time *in vitro*.

Supplementary Fig. 45. a The TEM images showcase the Ru-Cu/EDHJ materials obtained from the mandibular tissues captured at various time points after the implantation for 1 week and 8 weeks. Mandibular tissue samples were first fixed in osmium tetroxide, then dehydrated in acetone, followed by embedding in epoxy resin, and finally sectioned into ultra-thin slices for TEM imaging. **b** The ICP-MS test measured different metal ion concentrations in mandibular tissue after the implantation of the Ru-Cu/EDHJ for 1 week and 8 weeks. **c** The ICP-MS test measured different metal ion concentrations in mouse urine and feces after the implantation of the Ru-Cu/EDHJ for 4 weeks.

Supplementary Fig. 46. Paraffin-embedded heart, liver, spleen, lung, and kidney of mice on 14 days post-operation were sectioned and stained by H&E (n = 3 independent replicates). Scale bars: 100 μ m.

(8) *More motivation of the mandibular defect model should be provided and untreated controls should be more characterized.*

Response to comment:

Thank you for your valuable and constructive comments. Our mandibular defect model possesses unique characteristics, which we intend to highlight more prominently. Jaw lesions primarily involve damage to the bone tissue surrounding the root of the tooth; it should even be considered whether the root of the tooth is adjacent to formulate a treatment strategy for jaw fractures. This specificity distinguishes the repair of jaw injuries from that of ordinary long bones and skulls. Specifically, conditions such as periapical inflammation, jaw osteomyelitis, and periodontitis frequently result in apical defects that can progress to buccal bone wall defects and clinical sinus formations if left unaddressed. Our animal model effectively mimics these clinical realities by creating a bone defect that compromises the buccal bone wall in the periapical region (Supplementary Fig. 38). This approach not only enhances the relevance of our findings but also promises to inform future treatment strategies for bone defects arising from such conditions. According to your comments, we have incorporated the rationale for the mandibular defect model into the revised manuscript, as also outlined below:

Page 21 in the revised manuscript: “Initially, we construct a lipopolysaccharide (LPS)-induced inflammatory periapical bone defect (Supplementary Fig. 38)⁹¹, which is a ROS-enhanced mandible defect model (named as ROSup), evidenced by the fluorescence of ROS indicator (dihydroethidium (DHE) and DCFH-DA, Supplementary Fig. 39). This periapical bone defect construction mimics clinic infectious diseases inducing buccal bone wall defect in tooth apical region, such as periapical inflammation, jaw osteomyelitis, and periodontitis¹¹.”

Supplementary Fig. 38. The coronal plane of the mouse mandible by a micro-CT image to show the jaw defect model.

(9) *Why authors did not started with bone defect alone (without LPS-stimulation) and check biocatalyst effect on the ‘natural’ inflammatory phase of bone healing?*

Response to comment:

Thank you for your insightful comment regarding the choice of starting our study with an LPS-stimulation model. Based on our earlier observation and preliminary experiment, the regeneration of normal mandible bone defect without strong inflammatory microenvironments will not be difficult. Here, our primary aim was to develop a model that mirrors more challenging clinical scenarios where inflammation is exacerbated and healing is impeded. By using an LPS-stimulated model, we believe we can explore the effectiveness of our biocatalyst under more adverse conditions, thereby enhancing its potential clinical relevance. The related contents have also been illustrated in the introduction on page 3 of the revised manuscript: “Especially in inflammatory defects, high ROS-level microenvironments could aggravate localized tissue injury and lead to chronic inflammation^{14,16,17}. Therefore, clearance of excessive ROS to provide an appropriate microenvironment for endogenous stem cells has become an intriguing approach for promoting bone tissue regeneration in the treatments of inflammatory mandible defects¹⁸.”

However, we acknowledge that investigating normal bone defects without LPS stimulation could also yield valuable insights into the natural inflammatory phase of bone healing, especially confirming the effectiveness of our Ru-Cu/EDHJ in biocatalytic modulation and regeneration of mandible defects. Therefore, we have added the corresponding animal experiments by treating the bone defect alone (without LPS stimulation) with PBS and Ru-Cu/EDHJ. It can be found that the healing ability of Ru-Cu/EDHJ-treated normal bone defects is much faster compared to the PBS, thus indicating the Ru-Cu/EDHJ can provide an appropriate microenvironment for endogenous stem cells and promote rapid bone tissue regeneration within four weeks.

Fig. R2. 3D construction of micro-CT after operation for mouse mandible defect at 4 weeks (left: treated with PBS, right: treated with Ru-Cu/EDHJ).

(10) *The manuscript does not contain a literature supported-discussion section. There are large result sections that lacked referencing so a more detailed comparison of provided results with published literature should have been carried out.*

Response to comment:

Thank you for your constructive remarks on our manuscript. We wholeheartedly agree that a literature-supported discussion is essential for lending credibility to our findings. In response to your feedback, we have conducted a thorough review of the relevant literature and have included a more detailed comparison of our results with those previously published literature. The corresponding

literature supported-discussion section has been added to the revised manuscript, as also outlined below:

Page 14 in the revised manuscript: “The cell cytoskeleton and adhesion form a network that plays a crucial role in maintaining cellular structure and essential functions, which are vulnerable to oxidative stress. Evidence indicates that the disruption of cytoskeletal proteins is the initial step in cell damage caused by oxidative stress⁷⁰. To investigate the influence of oxidative stress on the cytoskeleton, F-actin cytoskeleton stained by FITC (fluorescein isothiocyanate)-labeled phalloidin is detected in the high ROS level microenvironment to validate the protection activity of Ru-Cu/EDHJ (Fig. 4f). In the ROS condition, hMSCs display weak-spreading, shrinking morphology, and disordered cytoskeleton. By contrast, after treating with Ru-Cu/EDHJ, hMSCs with H₂O₂ can maintain well-spreading and polygonal morphology with highly elongated and well-organized actin filaments (green) surrounding the cell nuclei (blue). This phenomenon of cytoskeleton change is also reported in MCF-7 cells under H₂O₂ stimulation⁴⁰.”

Page 17 in the revised manuscript: “Oxidative stress causes DNA damage in MSCs and extensive DNA damage leads invariably to cell death either by apoptosis or necrosis⁷²⁻⁷⁴. Thereafter, we systematically explore the catalytic therapeutic effects of Ru-Cu/EDHJ on preventing ROS-related DNA damage and apoptosis.”

Page 17 in the revised manuscript: “As the immunofluorescence staining (Fig. 5c) and the spectrum analysis of the nuclei results (Fig. 5d), the H₂O₂ group gives obvious γ -H2A.X signals co-express with Dapi, while the Ru-Cu/EDHJ group shows significantly decreased signals, which is as low as the Ctrl group as indicated by quantitative analysis in Fig. 5e, indicating the reduction of DNA damage under oxidative stress by Ru-Cu/EDHJ⁷⁵.”

Page 17 in the revised manuscript: “Cellular apoptosis occurs under an oxidative stress state, and TUNEL staining is used to detect the breakage of nuclear DNA during cell apoptosis⁷⁶.”

Page 17 in the revised manuscript: “The cells treated with H₂O₂ show an obvious apoptosis ratio, while the Ru-Cu/EDHJ group displays a similar ratio as the Ctrl group (Fig. 5i), indicating its efficient protection from oxidative stress-induced cell apoptosis⁷⁷.”

Page 17 in the revised manuscript: “To explore the protection of Ru-Cu/EDHJ on stem cell function, the *in vitro* osteogenic differentiation potential of hMSCs in high ROS levels is assessed by alizarin red (AR) and alkaline phosphatase (ALP) staining⁷⁸.”

Page 19 in the revised manuscript: “The function of Ru-Cu/EDHJ can be reflected in influencing cell cytoskeleton and adhesion (regulation of actin cytoskeleton, focal adhesion, Rap1 signaling pathway⁸⁰), cellular DNA damage and function, cellular activity (ECM-receptor interaction, cellular senescence, cell cycle), and cell apoptosis (apoptosis, p53 signaling pathway⁸¹). In addition, the Ru-Cu/EDHJ can protect stem cells by changing inflammatory expression (inflammatory mediator regulation of TRP channels, NF-kappa B signaling pathway⁸², IL-17 signaling pathway⁸³, JAK-STAT signaling pathway⁸⁴, and TNF signaling pathway⁸⁵), stem cell development and matrix formation (signaling pathways regulating pluripotency of stem cells, ECM-receptor interaction, and MAPK signaling pathway⁸⁶), and subsequently osteogenesis regulation (Wnt signaling pathway⁸⁷, PI3K-Akt signaling pathway⁸⁸, and TGF-beta signaling pathway⁸⁹).”

Page 21 in the revised manuscript: “Persistent inflammation in infected areas can result in continued inflammatory response and adversely affect the regeneration of bone tissue⁹⁰. Inspired by the efficient and synergistic stem cell protection activities, we further assess the *in vivo* efficiency of Ru-Cu/EDHJ in treating bone tissue regeneration under inflammatory environments. Initially, we construct a lipopolysaccharide (LPS)-induced inflammatory periapical bone defect (Supplementary Fig. 38)⁹¹, which is a ROS-enhanced mandible defect model (named as ROSup), evidenced by the fluorescence of ROS indicator (dihydroethidium (DHE) and DCFH-DA, Supplementary Fig. 39). This periapical bone defect construction mimics clinic infectious diseases inducing buccal bone wall defect in tooth apical region, such as apical periodontitis, periodontitis, osteomyelitis¹¹.”

Page 21 in the revised manuscript: “Moreover, we have included an analysis of the CD206 (M2 phenotype) and iNOS (M1 phenotype) markers through tissue section immunofluorescence double staining and quantitative assessment in the mandible defect area (Supplementary Fig. 40). The resulting data indicate that the Ru-Cu/EDHJ group shows a smaller proportion of iNOS relative to CD206 compared to the other experimental groups, suggesting its efficacy in promoting macrophage polarization toward the M2 (anti-inflammatory) phenotype and attenuating proinflammatory

responses⁹².”

Page 22 in the revised manuscript: “To investigate *in vivo* bone formation potential of Ru-Cu/EDHJ, micro-CT analysis is performed⁹³.”

Page 24 in the revised manuscript: Bacterial infection often leads to infectious jaw diseases, such as osteomyelitis⁹⁸, apical periodontitis⁹⁹, periodontitis¹⁰⁰, peri-implantitis¹⁰¹, and mandibular fractures with infection¹⁰² and so on, and antibacterial action can promote healing process at the wound site.”

References:

- 70 Zhu, D. H. *et al.* Hydrogen peroxide alters membrane and cytoskeleton properties and increases intercellular connections in astrocytes. *J. Cell Sci.* **118**, 3695-3703 (2005).
- 40 Yao, C. *et al.* Dynamic assembly of DNA-ceria nanocomplex in living cells generates artificial peroxisome. *Nat. Commun.* **13**, 7739 (2022).
- 72 McDonald, K. R., Hernandez-Nichols, A. L., Barnes, J. W. & Patel, R. P. Hydrogen peroxide regulates endothelial surface N-glycoforms to control inflammatory monocyte rolling and adhesion. *Redox Biol.* **34**, 101498 (2020).
- 73 Dhalla, N. S., Shah, A. K., Adameova, A. & Bartekova, M. Role of Oxidative Stress in Cardiac Dysfunction and Subcellular Defects Due to Ischemia-Reperfusion Injury. *Biomedicines* **10**, 1473 (2022).
- 74 Elliott, M. R. & Ravichandran, K. S. The Dynamics of Apoptotic Cell Clearance. *Dev. Cell* **38**, 147-160 (2016).
- 75 Nair, R., Hsu, J., Hobbs, R. & Coulombe, P. A. A novel role for keratin 17 during DNA damage response and tumor initiation. *J. Invest. Dermatol.* **141**, S14-S14 (2021).
- 76 Woo, K. M., Seo, J., Zhang, R. & Ma, P. X. Suppression of apoptosis by enhanced protein adsorption on polymer/hydroxyapatite composite scaffolds. *Biomaterials* **28**, 2622-2630 (2007).
- 77 Ye, T. *et al.* Protective effects of Pt-N-C single-atom nanozymes against myocardial ischemia-reperfusion injury. *Nat. Commun.* **15**, 1682 (2024).
- 78 Sun, H. *et al.* Osteogenic differentiation of human amniotic fluid-derived stem cells induced by bone morphogenetic protein-7 and enhanced by nanofibrous scaffolds. *Biomaterials* **31**, 1133-1139 (2010).
- 80 Kondo, N., Ueda, Y. & Kinashi, T. Kindlin-3 disrupts an intersubunit association in the integrin LFA1 to trigger positive feedback activation by Rap1 and talin 1. *Sci. Signaling* **14**, eabf2184 (2021).
- 81 Wang, X., Simpson, E. R. & Brown, K. A. p53: Protection against Tumor Growth beyond Effects on Cell Cycle and Apoptosis. *Cancer Res.* **75**, 5001-5007 (2015).
- 82 Yu, H., Lin, L., Zhang, Z., Zhang, H. & Hu, H. Targeting NF- κ B pathway for the therapy of diseases: mechanism and clinical study. *Signal Transduction Targeted Ther.* **5**, 77-92 (2020).
- 83 Zhang, Z. *et al.* TAOK1 negatively regulates IL-17-mediated signaling and inflammation.

- Cell. Mol. Immunol* **15**, 794-802 (2018).
- 84 Wang, L. *et al.* Targeting JAK/STAT signaling pathways in treatment of inflammatory bowel disease. *Inflammation Res.* **70**, 753-764 (2021).
- 85 Coste, E. *et al.* Identification of small molecule inhibitors of RANKL and TNF signalling as anti-inflammatory and antiresorptive agents in mice. *Ann. Rheum. Dis.* **74**, 220-226 (2015).
- 86 Nam, O. H. *et al.* Ginsenoside Rb1 alleviates lipopolysaccharide-induced inflammation in human dental pulp cells via the PI3K/Akt, NF- κ B, and MAPK signalling pathways. *Int. Endod. J.* **57**, 759-768 (2024).
- 87 Leng, Y. *et al.* Osteoblast-derived exosomes promote osteogenic differentiation of osteosarcoma cells via URG4/Wnt signaling pathway. *Bone* **178**, 116933 (2024).
- 88 Tang, L. *et al.* Fgf9 Negatively Regulates Bone Mass by Inhibiting Osteogenesis and Promoting Osteoclastogenesis Via MAPK and PI3K/AKT Signaling. *J. Bone Miner. Res.* **36**, 779-791 (2021).
- 89 Oka, K. *et al.* The role of TGF- β signaling in regulating chondrogenesis and osteogenesis during mandibular development. *Dev. Biol.* **303**, 391-404 (2007).
- 90 Thomas, M. V. & Puleo, D. A. Infection, Inflammation, and Bone Regeneration: a Paradoxical Relationship. *J. Dent. Res.* **90**, 1052-1061 (2011).
- 91 Zhang, Y. *et al.* A Zinc Oxide Nanowire-Modified Mineralized Collagen Scaffold Promotes Infectious Bone Regeneration. *Small* **20**, 2309230 (2024).
- 92 Wu, S. *et al.* M2 macrophages independently promote beige adipogenesis via blocking adipocyte Ets1. *Nat. Commun.* **15**, 1646 (2024).
- 93 Holzwarth, J. M. & Ma, P. X. Biomimetic nanofibrous scaffolds for bone tissue engineering. *Biomaterials* **32**, 9622-9629 (2011).
- 98 Sodnom-Ish, B. *et al.* Decompression effects on bone healing in rat mandible osteomyelitis. *Sci. Rep.* **11**, 11673 (2021).
- 99 Dai, X. Z. *et al.* Enterococcus faecalis-Induced Macrophage Necroptosis Promotes Refractory Apical Periodontitis. *Microbiol. Spectrum* **10**, e0104522 (2022).
- 100 Slots, J. Periodontitis: facts, fallacies and the future. *Periodontol. 2000* **75**, 7-23 (2017).
- 101 Wang, C.-W. *et al.* Laser-assisted regenerative surgical therapy for peri-implantitis: A randomized controlled clinical trial. *J. Periodontol.* **92**, 378-388 (2021).
- 102 Oksa, M., Haapanen, A., Marttila, E. & Snall, J. Simple dentate area fractures of the mandible - can we prevent postoperative infections? *Acta Odontol. Scand.* **80**, 494-500 (2022).

Specific comments:

(11) On page 13 (Subsection *In vitro* ROS scavenging and stem cell protection of Ru-Cu/EDHJ), the authors refer to the Supplementary Fig. 31, but the figure captions are wrong (refer to *in vivo* work).

Response to comment:

Thank you for your attentive review and valuable feedback. We apologize for the oversight in referencing Supplementary Fig. 31, as the captions incorrectly pertained to *in vivo* work. We have

rectified this issue in the revised manuscript to enhance the integrity of our presentation. Your careful examination is greatly appreciated, and the updated corrections are included in the revised manuscript and revised supplementary information, as also illustrated below:

Page 14 in the revised manuscript: “2,7-dichlorodihydrofluorescein diacetate (DCFH-DA) is applied as the fluorescence probe to explore intracellular ROS levels, as shown in Figs. 4a-c and Supplementary Fig. 31.”

Supplementary Fig. 31. **a** Fluorescence images, Ctrl (PBS), **b** and mean fluorescence intensity of 2,7-dichlorodihydrofluorescein diacetate (DCFH-DA) staining of ROS in mMSCs (n = 3 independent replicates), $p_{(H_2O_2)} < 0.0001$, $p_{(EDHJ-H_2O_2)} = 0.0144$, $p_{(Ru-O/EDHJ-H_2O_2)} < 0.0001$, $p_{(Ru-Cu/EDHJ-H_2O_2)} < 0.0001$. Scale bar: 100 μ m. Data are presented as means \pm SD., *** $p < 0.001$, # $p < 0.05$, ### $p < 0.001$; one-way ANOVA with multiple comparisons test.

(12) In the Cellular live/dead staining Supp methods section, it was described the assessment of cytotoxicity, and results were presented in the supplement figure 32. Regarding to the methodology,

the authors calculated live-dead cells after 48h of incubation with compounds. The final evaluation of dead/live cells were based on the cells, which were left attached to the plastic. Talking about hMSC, the population of dead cells is detaching from plastic and consequently, was washed away and excluded from assay. Additionally, 5000 cells were seeded per well and were there the same amount of cells after 5000?

Response to comment:

We sincerely appreciate your thoughtful comment regarding the potential limitations of our Live/Dead staining methodology in assessing cytotoxicity. Your observation about the detachment of dead hMSC cells during the washing process is indeed an important consideration. Nevertheless, the Live/Dead staining methodology remains prevalent in the field, as evidenced by studies such as *Adv. Funct. Mater.*, **2020**, 30, 2002234 and *Nat. Commun.*, **2024**, 15, 3131.

However, based on your suggestion, we have improved and repeated the experiment (Supplementary Fig. 33). In this revised experiment, attention was paid to the centrifugation of the plate before staining so that the cells settled on the bottom of the plate. In addition, we sipped supernatant gently to avoid sucking up cells as possible as we could. The results showed that the count of living cells or dead cells did not exhibit significant differences among the groups, suggesting the good biocompatibility of materials. To further enhance the robustness of our findings, we have also integrated results from an additional CCK-8 toxicity assay, specifically focusing on the biocatalyst in the absence of H₂O₂, detailed in Supplementary Fig. 32. Our results, which indicate a half maximal inhibitory concentration (IC₅₀) value of 426.4 µg/mL. The concentration value of 100 µg/mL used in this research shows no significant effect on cell viability, suggesting that the Ru-Cu/EDHJ biocatalyst exhibits almost no cytotoxicity in low concentrations. Your feedback has been invaluable in refining our methodology and enhancing the clarity of our findings. The corresponding details for the CCK-8 method can be found in the revised manuscript and revised supplementary information, as also shown below:

Page 15 in the revised manuscript: “The Cell Counting Kit-8 (CCK-8) method shows that the half maximal inhibitory concentration (IC₅₀) is 426.4 µg/mL and exhibits almost no cytotoxicity to cell viability in low concentration. The concentration value of 100 µg/mL used in this research shows

no significant effect on cell viability by CCK-8 and Live/Dead staining, indicating the excellent biocompatibility of Ru-Cu/EDHJ biocatalyst (Supplementary Figs. 32, 33).”

Page 43 in the revised supplementary information manuscript: “Cellular live/dead staining. For detecting hMSC protection by Ru-Cu/EDHJ in a high-level H₂O₂ microenvironment test, cells were seeded onto 96-well plates at the same density in different groups, and then 100 µg/mL of artificial biocatalysts and 100 µM hydrogen peroxide were added successively. After 1 hour, the medium was changed to a standard culture medium and cultured overnight. For the cytotoxicity test, cells were seeded onto 96-well plates at the same density in groups, and the treatment with 100 µg/mL of artificial biocatalysts was performed for 48 h. Before staining, the plate was centrifuged to make cells settle on the bottom. In addition, the supernatant was sipped gently to avoid sucking up cells. Then, Calcein AM/Propidium Iodide (PI) staining was performed according to the instructions (C2012, Beyotime, China). Briefly, 2 µM Calcein AM solutions in phosphate-buffered saline (pH=7.4, PBS) were used to stain live cells, then 4.5 µM PI solutions were used to stain the dead. The cells were captured by a confocal laser scanning microscope and counted by the Celigo Image Cytometer (Nexcelom Bioscience LCC., America).”

Supplementary Fig. 33. The number of cells with different materials counted after Live/Dead staining (n = 3 independent replicates). Data are presented as means ± SD.; ns indicates no significance; one-way ANOVA with multiple comparisons test.

Supplementary Fig. 32. Quantitative analysis of cell viability for Ru-Cu/EDHJ without H₂O₂ using the Cell Counting Kit-8 (CCK-8), inhibitory concentration (IC₅₀) = 426.4 µg/mL (n = 3 independent replicates). Data are presented as means ± SD., ***p* < 0.01, ****p* < 0.001; one-way ANOVA with multiple comparisons test.

(13) *Apoptosis, programmed cell death, occurs very fast as a response on the corresponding stimuli* (<https://www.ncbi.nlm.nih.gov/pmc/articles/PMC4966906/>). However, the authors exposed hMSC to biocatalysts plus H₂O₂ for 1h, but evaluated the level of apoptosis after 72h. It could be the same problem, that the most responsive cells were detached and washed away and excluded from analysis. For the next 72h, the mere resistant cells survived and proliferate.

Response to comment:

We appreciate your insightful comments regarding the timing of apoptosis detection in our study. In light of your feedback, we have checked the literature and noted that typical detection periods generally range from 12 to 48 hours post-stimulation (*Nat. Commun.*, **2022**, 13, 7739; *Nat. Commun.*, **2021**, 12, 1436; *Nat. Commun.*, **2022**, 13, 7449). Recognizing that our initial 72-hour timeframe was relatively extended, we have revised our approach and reduced the detection period to 24 hours (Figs. 5g, i). Our new findings indicate that, while cell adhesion did weaken after 1-hour exposure to the biocatalysts and H₂O₂, the proportion of floating cells remained minimal. We also recognize that

some detachment occurred after 12 hours, as highlighted by the reviewer. To counteract the potential loss of data, we conducted cell fluid gently once after 1-hour exposure and refrained from conducting cell fluid exchanges before detection, ensuring that we retained all cells for apoptosis detection. Furthermore, when preparing flow cytometry analysis for apoptosis, the cells in the supernatant and the digestive cells were centrifuged together to avoid the loss of cells.

Furthermore, our choice of a 1-hour H₂O₂ incubation aimed to induce a state of oxidative stress, a method of H₂O₂ stimulation in short periods, has also been utilized in some literature (*Nat. Commun.*, **2022**, 13, 7739; *Stem Cells Dev.*, **2012**, 21, 1877-1886). Based on the literature and our preliminary experiment, hMSCs experienced reduced adhesion after just 1 hour of exposure to H₂O₂, it could be observed that these damaged hMSCs maintained an attached state on the plate, which will not severely affect the results of apoptosis detection, as shown in Fig. R1.

Page 44 in the revised supplementary information manuscript: “**Cell apoptosis.** hMSCs were seeded in 6 mm plates and allowed to adhere. After being treated with different samples for 1 h, cells were cultured in a fresh medium for another 24 h²⁹. Then, the cells in the supernatant and the digestive cells were centrifuged together to avoid the loss of cells. Then, the Annexin V-FITC/PI Apoptosis Detection Kit (AD10, Dojindo, Japan) was used to stain the cells before the flow cytometry analysis. According to the fluorescence intensity, cell populations were assigned into four quadrants, including live, early apoptotic, late apoptotic, and necrotic cells. The figures were formed by the software of Flowjo (version 10.8.1). Moreover, the tunnel staining followed the manufacturer’s instructions for the One Step TUNEL Apoptosis Assay Kit (C1089, Beyotime Biotechnology, China), and the images were captured by a confocal laser scanning microscope.”

Fig. 5 g Apoptosis analysis by flow cytometry of Annexin V-FITC/PI stained hMSCs. **i** The cell percentages in stages of normal, early apoptosis, late apoptosis, and necrosis.

Fig. R1. White light photomicrograph of hMSCs after stimulation by H_2O_2 .

(14) *Fig 5G. The threshold level for PI positive cells is set between 10^2 – 10^3 . May you provide unstained controls in Suppl data to justify threshold choice.*

Response to comment:

Thanks for your good comment and helpful suggestion. We regret not providing unstained controls to support our threshold setting. In light of your suggestion, we have now included these controls in the supplementary data to properly justify our threshold choice. Thank you for your meticulous review; the related updates can be found in the revised supplementary information, as also shown below:

Supplementary Fig. 34. Apoptosis analysis by flow cytometry of Annexin V-FITC/PI stained hMSCs. The blank group indicates no staining; FITC and PI groups are single-dye staining with FITC and PI, respectively (n = 3 independent replicates).

(15) Page 15: Authors should also provide additional markers to assess *in vitro* osteogenic differentiation. This is crucial for dissecting the mechanism and effect of the novel material in a relevant assay. Additional immunostaining should be carried out for Osteocalcin, ALP and the alizarin red staining in order to prove osteogenic hMSC differentiation, the synthesis of COL1A1 is not sufficient to claim preserved osteogenic differentiation capacities.

Response to comment:

Thank you for your valuable and constructive comments for enhancing the quality of our manuscript. We wholeheartedly agree that incorporating additional markers to assess *in vitro* osteogenic differentiation is essential for elucidating the mechanisms and effects of the novel material. In response to your suggestions, we have now included alkaline phosphatase (ALP) staining, alizarin red (AR) staining, and osteocalcin (OCN) immunostaining to robustly demonstrate the osteogenic differentiation capabilities of the Ru-Cu/EDHJ-treated group. The results indicate that the group treated with H₂O₂ demonstrates reduced expression of ALP and decreased formation of

calcium nodules, as assessed by AR staining, as illustrated in Figs. 5j-l and Supplementary Figs. 35. However, the impaired osteogenic phenotype induced by H₂O₂ is found to recover with the addition of Ru-Cu/EDHJ. Quantitative analysis of the ALP and AR staining areas indicates that elevated H₂O₂ levels do not inhibit hMSC osteogenesis when supplemented with Ru-Cu/EDHJ. Additionally, increased expression levels of the osteogenesis-associated marker OCN were detected in the Ru-Cu/EDHJ group. These findings suggest that treatment with Ru-Cu/EDHJ effectively enhances the osteogenic differentiation capacities of hMSCs for superior bone regeneration. Your feedback has greatly contributed to the strength of our paper, and we appreciate your thoughtful review. The corresponding details can be found in the revised manuscript and revised supplementary information, as shown below:

Page 17 in the revised manuscript: “To explore the protection of Ru-Cu/EDHJ on stem cell function, the *in vitro* osteogenic differentiation potential of hMSCs in high ROS levels is assessed by alizarin red (AR) and alkaline phosphatase (ALP) staining⁷⁸. The group treated with H₂O₂ demonstrates a reduced expression of ALP and OCN (osteocalcin) and a decreased formation of calcium nodules by AR staining (Figs. 5j-l and Supplementary Figs. 35, 36). However, the impaired osteogenic phenotype induced by H₂O₂ is found to recover with the addition of Ru-Cu/EDHJ. Quantitative analysis of the ALP and AR staining areas indicates that elevated H₂O₂ levels do not inhibit both hMSCs and mMSCs osteogenesis when supplemented with Ru-Cu/EDHJ. These findings suggest that treatment with Ru-Cu/EDHJ effectively protects the osteogenesis of MSCs from bone marrow in a high ROS environment, indicating its support to bone regeneration (Fig. 5m)⁷⁸.”

Fig. 5 Prevention of ROS-related DNA damage and apoptosis in MSCs by Ru-Cu/EDHJ. **j** ALP staining after 3-day *in vitro* osteo-induction and AR staining after 21-day *in vitro* osteo-induction using the hMSCs. Quantitative analysis of **k** ALP (n = 3 independent replicates), $p_{(H_2O_2)} < 0.0001$, $p_{(Ru-Cu/EDHJ-H_2O_2)} < 0.0001$, and **l** AR (n = 3 independent replicates) in H₂O₂ treated hMSCs, $p_{(H_2O_2)} < 0.0001$, $p_{(Ru-Cu/EDHJ-H_2O_2)} < 0.0001$.

Supplementary Fig. 35. Immunohistochemical staining images of osteocalcin (OCN) in hMSCs, orange arrow indicates the positive expression area (n = 3 independent replicates). Scale bars: 200 μm.

(16) Figure 7 is lacking proper immunohistochemistry data of the regenerated defect. There should be additional insights on the regenerative process and outcome. There should be a link between the *in vitro* tests and the *in vivo* outcome which is currently not so tightly connected.

Response to comment:

We sincerely appreciate your constructive suggestions, which have significantly improved our manuscript. Given that MSCs are crucial endogenous cells in bone repair, it is vital to protect them and promote their osteogenic differentiation (*Cell Stem Cell*, **2018**, 22, 824-833; *Nat. Commun.*, **2022**, 13, 2499). Accordingly, we collectively demonstrated that Ru-Cu/EDHJ effectively protects MSCs from DNA damage induced by oxidative stress and promotes osteogenesis based on MSCs through both *in vitro* and *in vivo* experiments to enhance the connection between *in vitro* and *in vivo* findings during mandibular regeneration. Our focus is primarily on the following two aspects:

1) We have shown that Ru-Cu/EDHJ effectively protects MSCs from oxidative stress-induced DNA damage in our *in vitro* experiments (Fig. 5). Then we have newly added immunofluorescence (IF) analysis with double staining for DNA/RNA damage and the endogenous stem cell marker CD140a at the defect sites in our *in vivo* studies (Supplementary Fig. 42). The expression of DNA/RNA damage in the CD140a positive expression area was also quantified. These additions substantiate the protective role of Ru-Cu/EDHJ in CD140a⁺ MSCs from DNA damage in a high ROS environment. We recognize the necessity of establishing stronger connections between our *in vitro* and *in vivo* results, and we believe our revisions reflect this. The corresponding details can be found in the revised manuscript and revised supplementary information, as shown below:

Page 21 in the revised manuscript: “We used immunofluorescence (IF) analysis with double staining for DNA/RNA damage and CD140a at the defect sites in our *in vivo* studies (Supplementary Fig. 42). The expression of DNA/RNA damage in the CD140a positive expression area decreased obviously in the Ru-Cu/EDHJ group compared to the ROSup group.”

2) Additionally, we have enriched our *in vitro* osteogenic mineralization data with alizarin red (AR) and alkaline phosphatase (ALP) staining and complemented the *in vivo* analysis with Hematoxylin and Eosin (H&E) staining to delineate morphological changes in bone repair. The inclusion of osteocalcin (OCN) staining *via* immunohistochemistry (IHC) further elucidates the

mineralization process, reinforcing the connection between our *in vitro* findings and their *in vivo* implications. Specifically, the H₂O₂-treated group showed reduced expression of ALP and OCN and decreased calcium nodule formation, but this impaired osteogenic phenotype can be restored with Ru-Cu/EDHJ supplementation (Figs. 5j-l and Supplementary Figs. 35, 36). H&E staining reveals that while the ROSup groups exhibit prolonged inflammation and limited new bone formation, the Ru-Cu/EDHJ group demonstrates well-structured new bone with minimal inflammation and significantly elevated osteocalcin levels, highlighting its efficacy in promoting osteogenesis (Supplementary Fig. 43). The corresponding details can be found in the revised manuscript and revised supplementary information, as shown below:

Page 17 in the revised manuscript: “To explore the protection of Ru-Cu/EDHJ on stem cell function, the *in vitro* osteogenic differentiation potential of hMSCs in high ROS levels is assessed by alizarin red (AR) and alkaline phosphatase (ALP) staining⁷⁸. The group treated with H₂O₂ demonstrates a reduced expression of ALP and OCN (osteocalcin) and a decreased formation of calcium nodules by AR staining (Figs. 5j-l and Supplementary Figs. 35, 36). However, the impaired osteogenic phenotype induced by H₂O₂ is found to recover with the addition of Ru-Cu/EDHJ. Quantitative analysis of the ALP and AR staining areas indicates that elevated H₂O₂ levels do not inhibit both hMSCs and mMSCs osteogenesis when supplemented with Ru-Cu/EDHJ. These findings suggest that treatment with Ru-Cu/EDHJ effectively protects the osteogenesis of MSCs from bone marrow in a high ROS environment, indicating its support to bone regeneration (Fig. 5m)⁷⁸.”

Page 22 in the revised manuscript: “Hematoxylin and Eosin (H&E) staining is employed to evaluate the potential of the Ru-Cu/EDHJ in facilitating endogenous mandible regeneration⁹³. As depicted in Supplementary Fig. 43, the ROSup and CeO₂ groups show evident inflammatory cell infiltration in the defect regions, even after 8 weeks, indicating prolonged inflammation and ongoing damage in the surgical site. The Ctrl group exhibits only a limited amount of newly formed bone matrix in the defect boundary. Conversely, the Ru-Cu/EDHJ group demonstrates well-structured dense bone tissues with minimal inflammatory cell presence. In addition, the expression of osteocalcin (OCN) significantly increases in the Ru-Cu/EDHJ group, highlighting its potential role in promoting osteogenesis^{95,96}.”

Supplementary Fig. 42. **a** Fluorescence staining images of DNA/RNA damage and CD140a (platelet-derived growth factor receptor alpha), at week 1 after operation. **b** Quantitative results of mean fluorescence intensity of DNA/RNA damage in CD140a⁺ position (n = 3 independent replicates), $p_{(ROSup)} = 0.0469$, $p_{(Ru-Cu/EDHJ-ROSup)} = 0.0110$. Ctrl (mandible defects with PBS treatment). Data are presented as means \pm SD., * $p < 0.05$, # $p < 0.05$, one-way ANOVA with multiple comparisons test. Scale bars: 200 μ m.

Fig. 5 Prevention of ROS-related DNA damage and apoptosis in MSCs by Ru-Cu/EDHJ. **j** ALP staining after 3-day *in vitro* osteo-induction and AR staining after 21-day *in vitro* osteo-induction using the hMSCs. Quantitative analysis of **k** ALP (n = 3 independent replicates), $p_{(H_2O_2)} < 0.0001$, $p_{(Ru-Cu/EDHJ-H_2O_2)} < 0.0001$, and **l** AR (n = 3 independent replicates) in H₂O₂ treated hMSCs, $p_{(H_2O_2)} < 0.0001$, $p_{(Ru-Cu/EDHJ-H_2O_2)} < 0.0001$.

Supplementary Fig. 35. Immunohistochemical staining images of osteocalcin (OCN) in hMSCs, orange arrow indicates the positive expression area (n = 3 independent replicates). Scale bar: 200 μm .

Supplementary Fig. 36. a Alkaline phosphatase (ALP) staining after 3-day *in vitro* osteo-induction and alizarin red (AR) staining after 21-day *in vitro* osteo-induction using the mMSCs. Quantitative analysis of **b** ALP (n = 3 independent replicates), $p_{(\text{H}_2\text{O}_2)} < 0.0001$, $p_{(\text{Ru-Cu/EDHJ-H}_2\text{O}_2)} < 0.0001$, and **c** AR (n = 3 independent replicates) in H₂O₂ treated mMSCs, $p_{(\text{H}_2\text{O}_2)} < 0.0001$, $p_{(\text{Ru-Cu/EDHJ-H}_2\text{O}_2)} < 0.0001$. Data are presented as means \pm SD., *** $p < 0.001$, ### $p < 0.001$; one-way ANOVA with multiple comparisons test.

Supplementary Fig. 43. a Hematoxylin and Eosin (H&E) staining of regenerated bones induced by different scaffolds at week 8 after operation. Ctrl (mandible defects with PBS treatment). (Row 1: Overall observation of the mandible defect repair. Row 2: Magnified view of the center and boundary site of the defects). (N: new bone tissue. V: new blood vessels (black arrow). F: fibrous tissue. B: old bone. I: inflammatory infiltration). **b** Representative immunohistochemistry images of OCN. OCN (brown) and hematoxylin (blue). n = 3 independent replicates.

(17) *Should there be additional data demonstrating antibacterial activity in an in vivo setting? How would this affect pro-survival of MSCs?*

Response to comment:

Thank you for your valuable feedback regarding the need for additional *in vivo* data demonstrating antibacterial activity. We agree that such data are critical to the robustness of our findings. In response, we examined its performance in *S. aureus*- or *E. coli*-infected bone defects to evaluate whether Ru-Cu/EDHJ exhibits efficient antibacterial capacity under pathological conditions *in vivo* (Supplementary Fig. 51). After 1 week of treatment, the number of *S. aureus* and *E. coli* colonies in the Ru-Cu/EDHJ group was significantly lower than that in the other groups. The

quantification result shows the highest antibacterial efficiency of Ru-Cu/EDHJ. Furthermore, compared with the Ru-Cu/EDHJ group, more Ly6G⁺ granulocytes and fewer CD140a⁺ MSCs in the *S. aureus*-infected tissue sites could be observed from the H₂O₂ groups *via* immunofluorescence staining, indicating the high anti-infective efficacy of Ru-Cu/EDHJ can enhance the survival of endogenous MSCs (Supplementary Fig. 51). The corresponding details can be found in the revised manuscript and revised supplementary information, as shown below:

Page 25 in the revised manuscript: “To evaluate whether Ru-Cu/EDHJ exhibits efficient antibacterial capacity under pathological conditions *in vivo*, we initially examined its antibacterial performance in *S. aureus* and *E. coli* infected bone defects (Supplementary Fig. 51). After 1 week of treatment, the number of *S. aureus* and *E. coli* colonies in the Ru-Cu/EDHJ group was significantly lower than that in the other groups. The quantification result shows that Ru-Cu/EDHJ exhibits the highest antibacterial efficiency. Furthermore, compared with the H₂O₂ treatment, fewer Ly6G⁺ neutrophils and more CD140a⁺ MSCs could be observed from the Ru-Cu/EDHJ group with *S. aureus*-infection *via* immunofluorescence staining, indicating that the high anti-infective efficacy of Ru-Cu/EDHJ can enhance the survival of endogenous MSCs (Supplementary Fig. 51).”

Supplementary Fig. 51. **a** Bacterial colonies formed by *S. aureus* and *E. coli* after incubating with PBS, H_2O_2 , $CeO_2 + H_2O_2$, or $Ru-Cu/EDHJ + H_2O_2$. **b** The corresponding antibacterial efficiency of *S. aureus* ($n = 3$ independent replicates), $p_{(Bacteria)} < 0.0001$, $p_{(H_2O_2)} < 0.0001$, $p_{(CeO_2-H_2O_2)} < 0.0001$. Data are presented as means \pm SD., $***p < 0.001$, one-way ANOVA with multiple comparisons test. **c** The corresponding antibacterial efficiency of *E. coli* ($n = 3$ independent replicates), $p_{(Bacteria)} < 0.0001$, $p_{(H_2O_2)} < 0.0001$, $p_{(CeO_2-H_2O_2)} < 0.0001$. Data are presented as means \pm SD., $***p < 0.001$, one-way ANOVA with multiple comparisons test. **d** Fluorescence staining images of Ly6G and CD140a at week 1 after operation. **e** Quantitative results of fluorescence intensity of Ly6G ($n = 3$ independent replicates), $p_{(Ru-Cu/EDHJ)} = 0.0016$. Data are presented as means \pm SD., $**p < 0.01$, p values are assessed by unpaired Student's two-sided t-tests. **f** Quantitative results of fluorescence intensity of CD140a ($n = 3$ independent replicates), $p_{(Ru-Cu/EDHJ)} = 0.0010$. Data are presented as means \pm SD., $***p < 0.001$, p values are assessed by unpaired Student's two-sided t-tests. All antibacterial experiments are performed in acidic conditions (PBS: pH 5.6).

(18) *Regarding bone defect models. May authors add the physical bone defect dimensions in the methods section, currently it is mentioned BR-49 bur. What were the physical dimensions of the gelatin sponge implanted? Did it dissolve after 8 weeks of healing?*

Response to comment:

Thank you for your insightful comments and helpful suggestions regarding the bone defect models described in our manuscript. We appreciate your attention to detail, which has helped us clarify some ambiguities in our methods section. To specify, the BR-49 bur used in our procedures has a nominal diameter of 0.8 mm. The gelatin sponge that was implanted has a cylindrical form, with both diameter and height dimensions measuring around 1 mm. Regarding the degradability of the gelatin sponge, our H&E staining observations indicated that the majority of the sponge material was effectively degraded by 8 weeks post-implantation (Fig. R3). Notably, a minimal amount of the sponge remained visible at that time, primarily in samples that exhibited significant inflammation and inadequate bone tissue formation. This aligns with existing literature, which also indicates the biodegradable nature of gelatin sponges. The similar results are also found in (*Adv. Mater.*, **2022**, 34, e2108300). We appreciate your valuable input and understand the importance of these dimensions for reproducibility and practical application potential. Therefore, we have incorporated this additional information into our revised supplementary information, as also shown below:

Page 45 in the revised supplementary information manuscript: “The mice were anesthetized with 4% (w/v) isoflurane, followed by an intraperitoneal injection of ketamine (60 mg/kg) and xylazine (12 mg/kg), combined with a subcutaneous injection of buprenorphine for analgesia. The furs surrounding the surgical area were removed, and the skin was disinfected. The area of interest was dissected by an incision until the body of the mandible was reached (buccal plate). After locating the buccal plate, a BR-49 round bur (around 0.8 mm in diameter) was used to initiate access to the buccal bone, followed by the immersing of the end of the bur to create a standard defect.”

Fig. R3. Hematoxylin and Eosin (H&E) staining of the mandibular defect was performed after 1 week and 8 weeks, respectively. The yellow arrows indicate the residual gelatin sponges.

Reviewer #3 (Remarks to the Author):

“I co-reviewed this manuscript with one of the reviewers who provided the listed reports. This is part of the Nature Communications initiative to facilitate training in peer review and to provide appropriate recognition for Early Career Researchers who co-review manuscripts.”

Response to the general comment:

Thank you for your invaluable feedback and thoughtful comments on our manuscript. We greatly appreciate the insights provided by you and the other reviewers; these comments have played a crucial role in enhancing the quality of our work. In the revised manuscript, we have made substantial revisions based on the reviewers’ suggestions, and we believe that the revised manuscript has been significantly improved with these helpful comments. Thank you once again for your generous support and the time you dedicated to co-reviewing this paper.

We thank all referees again for their helpful comments and suggestions, and hope that this significantly revised manuscript is now acceptable for publication in *Nature Communications*.

Best Regards,

Yours Sincerely,

Prof. Dr. Chong Cheng (on behalf of the authors)

REVIEWERS' COMMENTS

Reviewer #1 (Remarks to the Author):

The authors have adequately addressed my previous concerns. I recommend the acceptance of the manuscript for publication in Nature Communicatoins.

Reviewer #2 (Remarks to the Author):

The authors have carried out all requested experiments and analysis. I am impressed with the extent of work that was carried out in order to address my comments. I believe that the narrative is very well aligned and all claims are now supported by experimental results. I am therefore in favor of suggesting this manuscript for publication in Nature Comms.

Reviewer #3 (Remarks to the Author):
